# Towards Understanding Link Predictor Generalizability Under Distribution Shifts

## Abstract

State-of-the-art link prediction (LP) models demonstrate impressive benchmark results. However, popular benchmark datasets often assume that training, validation, and testing samples are representative of the overall dataset distribution. In real-world situations, this assumption is often incorrect; since uncontrolled factors lead to the problem where new dataset samples come from different distributions than training samples. The vast majority of recent work focuses on dataset shift affecting node- and graph-level tasks, largely ignoring link-level tasks. To bridge this gap, we introduce a novel splitting strategy, known as LPShift, which utilizes structural properties to induce a controlled distribution shift. We verify the effect of LPShift through empirical evaluation of SOTA LP methods on 16 LPShift generated splits of Open Graph Benchmark (OGB) datasets. When benchmarked with LPShift datasets, GNN4LP methods frequently generalize worse than heuristics or basic GNNs. Furthermore, LP-specific generalization techniques do little to improve performance under LPShift. Finally, further analysis provides insight on why LP models lose much of their architectural advantages under LPShift.

## 1 Introduction

Link Prediction (LP) is concerned with predicting unseen links (i.e., edges) between two nodes in a graph (Liben-Nowell & Kleinberg, 2003). The task has a wide variety of applications including: recommender systems, (Fan et al., 2019), knowledge graph completion (Lin et al., 2015), protein-interaction (Kovács et al., 2019), and drug discovery (Abbas et al., 2021). Traditionally, LP was performed using heuristics that model the pairwise interaction between two nodes (Newman, 2001; Zhou et al., 2009; Adamic & Adar, 2003). The success of Graph Neural Networks (GNNs) (Kipf & Welling, 2017) has prompted their usage in LP (Kipf & Welling, 2016; Zhang & Chen, 2018). However, GNNs are unable to fully-capture representations for node pairs (Zhang et al., 2021; Srinivasan & Ribeiro, 2019). To combat this problem, recent methods (i.e., GNN4LP) empower GNNs with additional information to capture pairwise interactions between nodes (Shomer et al., 2024; Chamberlain et al., 2022; Zhang & Chen, 2018; Wang et al., 2023a) and demonstrate tremendous ability to model LP on real-world datasets (Hu et al., 2020).

While recent methods have shown promise, current benchmarks (Hu et al., 2020) assume that the training and evaluation data are *drawn from the same structural distribution*, with the exception of datasets like ogbl-collab. This assumption often collapses in real-world scenarios, where the structural feature (i.e., covariate) distribution may shift from training to evaluation. Therefore, it's often necessary for models to generalize to samples whose newly-introduced feature distribution differs from the the training dataset (Wiles et al., 2021; Yao et al., 2022a;b; Bevilacqua et al., 2021).

Consider the three graphs in Figure 1 as samples pulled from a larger graph dataset. In order of appearance, each of the three graphs represent samples from the: training, validation, and testing splits of our dataset. Our task is to predict whether new links will form between existing nodes. These predicted links are shown as dotted lines with colors: "green" = training, "blue" = validation, "red" = testing. An optimal LP model trained on the first graph sample effectively learns a representation understanding source and target nodes with 2 Common Neighbors (Zhang & Chen, 2018), making the model effective at predicting new links in structure-rich scenarios. However, such a model is likely to fail when predicting links on the subsequent validation and testing samples; the second and third graph in Figure 1 with 1 CN and 0 CNs, respectively. The learned representation necessary to

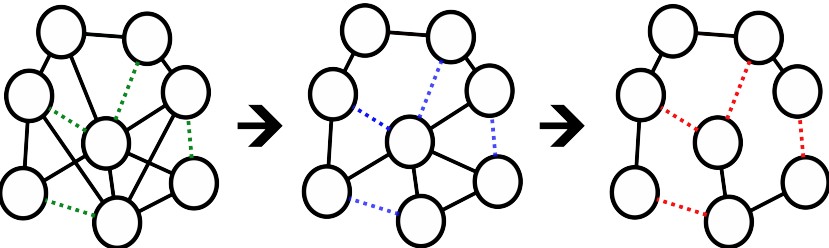

Figure 1: A miniature example of the backward CN LPShift, predicted links between source and target nodes are represented with dotted lines. The loss of edges cause a structural shift, forcing models to generalize on fewer Common Neighbors (CNs). Green = 2 CNs, Blue = 1 CN, Red = 0 CNs.

capture the differing pairwise relations in the second and third graphs requires generalizing to a scenario with fewer CNs, and therefore less structural information. As such, the previously-mentioned scenario represents a case of structural covariate shift, where the training distribution cannot effectively model the testing distribution, defined mathematically as $P^{Train}(X) \neq P^{Test}(Y)$. LPShift, our proposed splitting strategy, provides a controllable means to induce the scenario shown in Figure 1 (labelled in experiments as the CN - 2,1 split), as well as fifteen other tested scenarios. More details on structural shift are included in Section 3.2.

Furthermore, while numerous methods work to account for distribution shifts within graph machine learning (Li et al., 2022b), there remains little work doing so for LP. Specifically, we observe that **(1) No LP Benchmark Datasets**: Current graph benchmark datasets designed with a quantifiable distribution shift are focused solely on the node and graph tasks (Zhou et al., 2022b; Ding et al., 2021), *with no datasets available for LP*. **(2) Absence of Foundational Work**: There is limited existing work for distribution shifts relevant to LP (Zhang et al., 2022). Current methods are primarily focused on detecting and alleviating anomalies within node- and graph-level tasks (Jin et al., 2022; Bevilacqua et al., 2021; Gao et al., 2023; Wu et al., 2024; Guo et al., 2023; Wu et al., 2023; Sui et al., 2024; Li et al., 2022b). Additionally, few methods are designed for aiding LP generalization in any setting (Dong et al., 2022; Zhao et al., 2022; Zhang et al., 2022; Zhou et al., 2022b). Also, other LP generalization methods which are theorized to improve performance in shifted scenarios remain crucially untested (Singh et al., 2021; Wang et al., 2023b).

To tackle these problems, this work proposes the following contributions:

- **Creating Datasets with Meaningful Distribution Shifts**. LP requires pairwise structural considerations (Liben-Nowell & Kleinberg, 2003; Mao et al., 2024). Additionally, when considering realistic settings (Li et al., 2024) or distribution shift (Zhu et al., 2024), GNN4LP models perform poorly relative to models used in graph (Wei et al., 2021; Yuan et al., 2021) and node classification (Shi et al., 2023; Zhao et al., 2023). To better understand distribution shifts, we use key structural LP heuristics to split the links into train/validation/test splits via LPShift. By applying LPShift to generate dataset splits, we induce shifts in the underlying feature distribution of the links which are relevant to the link's formation (Mao et al., 2024). Further justification is provided in Appendix Section L.

- **Benchmarking Current LP Methods**. To our surprise, GNN4LP models struggle more than simpler methods when generalizing to data split by LPShift. Despite the existence of LP generalization methods, such as FakeEdge (Dong et al., 2022) and Edge Proposal Sets (Singh et al., 2021), there remains little work benchmarking link-prediction models under distribution shifts (Ding et al., 2021; Dong et al., 2022; Zhu et al., 2024). This lack of benchmarking contributes to a gap in understanding, impeding the capabilities of LP models to generalize. This work quantifies the performance of current SOTA LP models under 16 unique LPShift scenarios and provides analysis as a foundation for improving LP model generalization. We further quantify the effects of LP and graph-specific generalization methods, finding that they also struggle to generalize with differing structural shifts.

The remainder of this paper is structured as follows. In Section 2, we provide background on the heuristics, models, and generalization methods used in LP. In Section 3, we detail how the heuristics relate to our proposed splitting strategy and formally introduce LPShift. Lastly, in Section 4, we benchmark a selection of LP models and generalization methods on LPShift, followed by analysis with the intent of understanding the effects of this new strategy.

## 2 RELATED WORK

**LP Heuristics**: Classically, neighborhood heuristics, which measure characteristics between source and target edges, functioned as the primary means of predicting links. These heuristics show limited effectiveness with a relatively-high variability in results, largely due to the complicated irregularity within graph datasets which only grows worse with larger datasets (Liben-Nowell & Kleinberg, 2003). Regardless of this, state-of-the-art GNN4LP models have integrated these neighborhood heuristics into neural architectures to elevate link prediction capabilities (Wang et al., 2023a; Chamberlain et al., 2022).

For a given heuristic function, $u$ and $v$ represent the source and target nodes in a potential link, $(u, v)$. $\mathcal{N}(v)$ is the set of all edges, or neighbors, connected to node $v$. $f(v_{i,i+1}, u)$ is a function that considers all paths of length $i$ that start at $v$ and connect to $u$. The three tested heuristics are as follows:

*Common Neighbors* (Newman, 2001): The number of neighbors shared by two nodes $u$ and $v$,

$$\text{CN}(u, v) = |\mathcal{N}(u) \cap \mathcal{N}(v)|. \tag{1}$$

*Preferential Attachment* (Liben-Nowell & Kleinberg, 2003): The product of the number of neighbors (i.e., the degree) for nodes $u$ and $v$,

$$\text{PA}(u, v) = |\mathcal{N}(u)| \times |\mathcal{N}(v)|. \tag{2}$$

*Shortest Path Length* (Liben-Nowell & Kleinberg, 2003): The path between $u$ and $v$ which considers the smallest possible number of nodes, denoted as length $n$,

$$\text{SP}(u, v) = \arg\min_{\Sigma}(\Sigma_{i=1}^{n-1} f(v_{i,i+1}, u)). \tag{3}$$

**GNNs for Link Prediction (GNN4LP)**: LP's current SOTA methods rely on GNNs for a given model's backbone. The most common choice is the Graph Convolutional Network (GCN) (Kipf & Welling, 2017), integrating a simplified convolution operator to consider a node's multi-hop neighborhood. The final score (i.e., probability) of a link existing considers the representation between both nodes of interest. However, (Zhang et al., 2021) show that such methods aren't suitably expressive for LP, as they ignore vital pairwise information that exists between both nodes. To account for this, SEAL (Zhang & Chen, 2018) conditions the message passing on both nodes in the target link by applying a node-labelling trick to the enclosed k-hop neighborhood. They demonstrate that this can result in a suitably expressive GNN for LP. NBFNet (Zhu et al., 2021) conditions the message passing on a single node in the target link by parameterizing the generalized Bellman-Ford algorithm. In practice, it's been shown that conditional message passing is prohibitively expensive to run on many LP datasets (Chamberlain et al., 2022). Instead, recent methods pass both the standard GNN representations and an additional pairwise encoding into the scoring function for prediction. For the pairwise encoding, Neo-GNN (Yun et al., 2021) considers the higher-order overlap between neighborhoods. BUDDY (Chamberlain et al., 2022) estimates subgraph counts via sketching to infer information surrounding a target link. Neural Common-Neighbors with Completion (NCNC) (Wang et al., 2023a) encodes the enclosed 1-hop neighborhood of both nodes. Lastly, LPFormer (Shomer et al., 2024) adapts a transformer to learn the pairwise information between two nodes.

**Generalization in Link Prediction**: Generalization methods for LP rely on a mix of link and node features in order to improve LP model performance. DropEdge (Rong et al., 2020) randomly removes edges with increasing probability from the training adjacency matrix, allowing for different views of the graph. Edge Proposal Sets (EPS) (Singh et al., 2021) considers two models – a filter and rank model. The filter model is used to augment the graph with top-k relevant common neighbors, while the rank method scores the final prediction. (Wang et al., 2023b) built Topological Concentration (TC), which considers the overlap in subgraph features for a given node with each connected

---

**Algorithm 1** Dataset Splitting Strategy (LPShift)

---

**Require:**
  $G$ = Initial Graph,
  $\Psi(.,.)$ = Heuristic function
  $i_{train}, i_{valid}$ = Heuristic score thresholds
  $Train, Valid, Test = \emptyset, \emptyset, \emptyset$

1: **while** edge, $(u,v)$ not visited in $G$ **do**
2:   $\Psi(u,v) = h(u,v)$                               ▷ Score edge with neighborhood heuristic
3:   **if** $h(u,v) \leq i_{train}$ **then**                               ▷ Train Split
4:     $Train \leftarrow (u,v)$
5:   **else if** $h(u,v) > i_{train}$ and $h(u,v) \leq i_{valid}$ **then**     ▷ Valid split
6:     $Valid \leftarrow (u,v)$
7:   **else**                                          ▷ $h(u,v) > i_{valid}$, Test Split
8:     $Test \leftarrow (u,v)$
9:   **end if**
10: **end while**
11: **return** $Train, Valid, Test$                            ▷ Return Final Splits

---

neighbor, correlating well with LP performance for individual links. To improve the performance of links with a low TC, a re-weighting strategy applies more emphasis on links with a lower TC. Counter-Factual Link Prediction (CFLP) (Zhao et al., 2022) conditions a pre-trained model with edges that contain information counter to the original adjacency matrix, allowing models to generalize on information not present in a provided dataset.

## 3 BENCHMARK DATASET CONSTRUCTION

In this section, we explain how LPShift induces a shift in each dataset's structure; clarifying the importance of each structural measure and their application to splitting graph data.

### 3.1 TYPES OF DISTRIBUTION SHIFTS

We induce distribution shifts by splitting the links based on key structural properties affecting link formation and thereby LP. We consider three type of metrics: Local structural information, Global structural information, and Preferential Attachment. Recent work by (Mao et al., 2023) has shown the importance of local and global structural information for LP. Furthermore, due to the scale-free nature of many real-world graphs relates to link formation (Barabási & Albert, 1999), we also consider Preferential Attachment. A representative metric is then chosen for each of the three types, shown as follows:

**(1) Common Neighbors (CNs)**: CNs measure *local structural information* by considering only those nodes connected to the target and source nodes. A real-world case for CNs is whether you share mutual friends with a random person, thus determining if they are your "friend-of-a-friend" (Adamic & Adar, 2003). CNs plays a large role in GNN4LP, given that NCNC (Wang et al., 2023a) and EPS (Singh et al., 2021) integrate CNs into their framework and achieve SOTA performance. Furthermore even on complex real-world datasets, CNs achieves competitive performance against more advanced neural models (Hu et al., 2020). To control for the effect of CNs on shifted performance, the relevant splits will consider thresholds which include more CNs.

**(2) Shortest Path (SP)**: SP captures a graph's *global structural information*, thanks to the shortest-path between a given target and source node representing the most efficient path for reaching the target (Russell & Norvig, 2009). The shift in global structure caused by splitting data with SP can induce a scenario where a model must learn how two dissimilar nodes form a link with one another (Evtushenko & Kleinberg, 2021), which is comparable to the real-world scenario where two opponents choose to co-operate with one another (Schelling, 1978; Granovetter, 1978).

**(3) Preferential Attachment (PA)**: PA captures the *scale-free property* of larger graphs by multiplying the degrees between two given nodes (Barabási & Albert, 1999). When applied to graph generation, PA produces synthetic Barabasi-Albert (BA) graphs which retain the scale-free property

to effectively simulate the formation of new links in real-world graphs, such as the World Wide Web (Barabási & Albert, 1999; Albert & Barabási, 2002). Similar to CNs, the relevant PA splits will consider thresholds that integrate higher PA values.

## 3.2 DATASET SPLITTING STRATEGY

In the last subsection we described the different types of metrics to induce distribution shifts for LP. The metrics cover fundamental structural properties that influence the formation of new links. We now describe how we use these measures to split the dataset into train/validation/test splits to induce such shifts.

In order to build datasets with structural shift, we apply a given neighborhood heuristic to score each link. This score is then compared to a threshold ($i_{train}, i_{valid}$) to categorize a link as a different sample. As denoted in Alg. 1, the heuristic score of the link $(u, v)$ is $h(u, v)$. The link falls into: training when $h(u, v) < i_{train}$, validation when $i_{train} < h(u, v) \leq i_{valid}$, and testing when $h(u, v) > i_{valid}$. The new training graph is constructed from the original OGB dataset (Hu et al., 2020). Validation and testing samples are removed from the new training graph to prevent test-leakage and limited to 100-thousand edges maximum. The full algorithm is detailed in Algorithm 1 with additional details in Appendix C.

With Figure 2, we provide a small example of how splits are produced by our proposed splitting strategy. Specifically, Figure 2(a) demonstrates an outcome of the CN split labelled "CN - 1,2" where sampled edges pulled from the: black-dotted line = training (no CNs), red-dotted line = validation, (1 CN), and blue-dotted line = testing ($\geq$2 CNs). See Appendix C for information on Figure 2(b) and Figure 2(c).

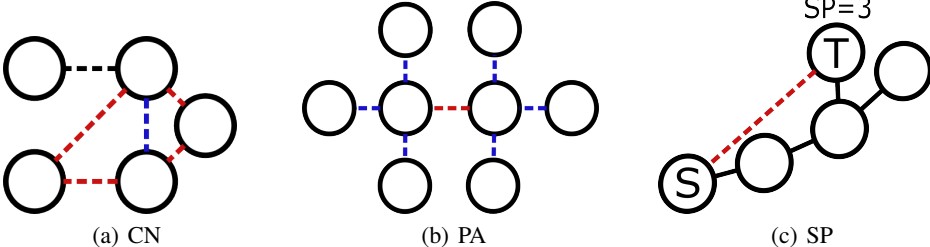

(a) CN         (b) PA         (c) SP

Figure 2: An example of the three splitting strategies: (a) Common Neighbors, (b) Preferential-Attachment, (c) Shortest-Path. The dashed lines represent a cut on the source and target node that forms a given edge for our splitting strategy. The color of the line distinguishes the score assigned by the heuristic.

To test how different LPShift thresholds impact performance, we adjust the $i_{train}$ and $i_{valid}$ thresholds to produce 3 varied CN and PA splits; as well as 2 varied SP splits. The variations in split thresholds were chosen based on two conditions. **1)** Structural information within splits varies due to user-defined thresholds. In the "Forward" scenario (i.e. CN - 1,2), splits are given increasingly more structural information between training to validation, giving the model an easier time generalizing under testing. For the "Backward" scenario (i.e, CN - 2,1), stricter thresholds mean less structural information between validation to testing; making generalization difficult. **2)** The final dataset split contains a sufficient number of samples. Each LPShift split requires enough split samples to allow model generalization. Given the limited number of SP split samples, the SP splits were limited to 2 variants. See Appendix C for more algorithmic details, and Appendix E for more information on LPShift's efficiency and usefulness

## 4 EXPERIMENTS

To bridge the gap for GNN4LP generalizing under distribution shifts, this work addresses the following questions: **(RQ1)** Is the distribution shift induced by LPShift significant? **(RQ2)** Can SOTA GNN4LPs generalize under our proposed distribution shifts? **(RQ3)** Can current LP generaliza-

tion methods further boost the performance of current methods? **(RQ4)** What components of the proposed distribution shift are affecting the LP model's performance?

### 4.1 EXPERIMENTAL SETUP

**Datasets**: We consider 16 "Forward" and "Backward" LPShift splits of the ogbl-collab and ogbl-ppa datasets (Hu et al., 2020), for a total of 32 tested splits. The resulting datasets represent tasks in two separate domains and three shifted scenarios, allowing a comprehensive study of LP generalization under distribution shift. For both datasets, we create multiple splits corresponding to each structural property detailed in Section 3.2. For the "Forward" split, denoted as $(X, Y, Z)$, an increase in $Y$ and $Z$ indicates more structural information available to the training adjacency matrix. The "Backward" split swaps the training and testing splits from their counterpart in the "Forward" split, resulting in the training adjacency matrix losing access to structural information as $X$ and $Y$ increase. See Appendix K for more details.

**GNN4LP Methods**: We test multiple SOTA GNN4LP methods including: NCNC (Wang et al., 2023a), BUDDY (Chamberlain et al., 2022), LPFormer (Shomer et al., 2024), SEAL (Zhang et al., 2021) and Neo-GNN (Yun et al., 2021). We further consider GCN (Kipf & Welling, 2017) as a simpler GNN baseline, along with the Resource Allocation (RA) (Zhou et al., 2009) heuristic. All models were selected based on their benchmark performance with the original OGB datsets (Hu et al., 2020) and their architectural differences detailed in Section 2.

**Generalization Methods**: We also test the performance of different LP models with multiple generalization techniques. This includes DropEdge (Rong et al., 2020), which randomly removes a portion of edges from the training adjacency matrix. Edge Proposal Sets (EPS) (Singh et al., 2021), which utilizes one LP model to filter edges based on common neighbors and another method to rank the top-k filtered edges in the training adjacency matrix. Lastly, we consider Topological Concentration (TC) (Wang et al., 2023b), which re-weights the edges within the training adjacency matrix based on the structural information captured by the TC metric.

**Evaluation Setting**: We consider the standard evaluation procedure in LP, in which every positive validation/test sample is compared against $M$ negative samples. The goal is that the model should output a higher score (i.e., probability) for positive sample than the negatives. To create the negatives, we make use of the HeaRT evaluation setting (Li et al., 2024) which generates $M$ negatives samples *per positive sample* according to a set of common LP heuristics. In our study, we set $M = 250$ and use CNs as the heuristic in HeaRT.

**Evaluation Metrics**: We evaluate all methods using multiple ranking metrics as a standard practice in LP literature (Li et al., 2024). This includes the mean reciprocal rank (MRR) and Hits@20.

**Hyperparameters**: All methods were tuned on permutations of learning rates in $\{1e^{-2}, 1e^{-3}\}$ and dropout in $\{0.1, 0.3\}$. Each model was trained and tested over five seeds to obtain the mean and standard deviations of their results. Given the significant time complexity of training and testing on the customized ogbl-ppa datasets, NCNC and LPFormer were tuned on a single seed, followed by an evaluation of the tuned model on five separate seeds. We include additional hyperparameter details within Appendix Section K

### 4.2 RESULTS FOR GNN4LP

In order to provide a unified perspective on how distribution shift affects link prediction models, each GNN4LP method was trained and tested across five seeded runs on versions of ogbl-collab and ogbl-ppa split by: Common Neighbors, Shortest-Path, and Preferential-Attachment. Examining the results, we have the following three key observations.

**Observation 1: Poor Performance of GNN4LP**

As shown in Table 1, RA and GCN consistently out-perform or remain competitive with GNN4LP models, with more detailed results included in Appendix G. In Table 7 and 8, RA is overwhelmingly the best-performing, achieving scores at least 6 MRR and 13 Hits@20 higher than the next best model. However the results for the ogbl-ppa forward split, as shown in Table 9, indicate LPFormer as the best-performing model on the PA split and NeoGNN on the CN - 3,5 split, albeit with a much lower average score than demonstrated within the ogbl-collab forward split.

Given ogbl-ppa's reduction in performance and the superiority of simpler methods with ogbl-collab, the structural shift induced by LPShift makes it difficult for GNN4LP to generalize. A key consideration for this result is LPShift's direct effect on graph structure, which applies a covariate shift to features (Koh et al., 2021), especially where structure is correlated to the feature distribution.

To further quantify the extent of this correlation with LPShift, we measure the cosine similarity of each split's feature distribution, as shown in Appendix D. Additional analysis shown in Figure 5 and 6 indicate the performance effects of LPShift.

### Observation 2: Performance Differs By Both Split Type and Thresholds.

As shown in Figure 3, regardless of whether a model is tested on a "Forward" or "Backward" split; the change in structural information for each subsequent split gradually changes a model's performance. We note that the results for ogbl-ppa and ogbl-collab nearly mirror one another for any given "Forward" split; where an ogbl-ppa split increases, the respective ogbl-collab decreases. On the "Backward" split, a stark increase is seen across most splits, indicating that more structural information between training and validation improves LP performance (Wang et al., 2023a). The fact that these results include splits produced by Preferential-Attachment, Global Structural Information (SP), and Local Structural Information (CN) indicates the effect of *any* change in structural information when training LP models (Mao et al., 2023).

Table 1: Mean LP model rank for every tested 'forward' and 'backward' LPShift split. Rankings determined by Hits@20.

| Split Type | Forward | | | Backward | | |
|---|---|---|---|---|---|---|
| | CN | SP | PA | CN | SP | PA |
| RA | 3.17 | **1** | 3 | 5.33 | 7 | **2.33** |
| GCN | **2.67** | 3.75 | **2.8** | 2.5 | **1.25** | 6 |
| BUDDY | 3.67 | 3.25 | 4.8 | 4 | 1.75 | 4.33 |
| NCNC | 4.33 | 3.0 | 5.8 | **2** | 4 | 3.67 |
| LPFormer | 5.5 | 4.25 | 4 | 3.5 | 4 | **2.33** |
| NeoGNN | 4.83 | 6 | 3.8 | 5.17 | 4 | 6.5 |
| SEAL | 3.83 | 6.5 | 3.8 | 5.5 | 6 | 2.83 |

**Observation 3: Impacts on Performance Vary by Model.** All raw model scores are stored in Appendix Section G within Tables 7 to 14. **Common Neighbors:** Most models fail to generalize on the "Backward" CN splits. However, once more Common Neighbors are made available in the CN - 4,2 and CN - 5,3 splits; NCNC performs 2 to 3 times better than other GNN4LP models. Therefore, indicating that it is possible to generalize with limited local information. **Shortest-Path:** GNN4LP Models which rely more on local structural information (i.e. NCNC, LPFormer, and SEAL) typically suffer more under the "Backward" SP splits, resulting in the models performing 2x to 4x worse than BUDDY or GCN. Therefore, indicating the necessity for models to adapt in scenarios with an absence of local structural information. **Preferential-Attachment:** Model performance on the PA split is often 2 times higher than the original ogbl-collab (Hu et al., 2020), but reduces drastically with LPShift's ogbl-ppa. Therefore, indicating the impact that structural shift incurs on larger datasets.

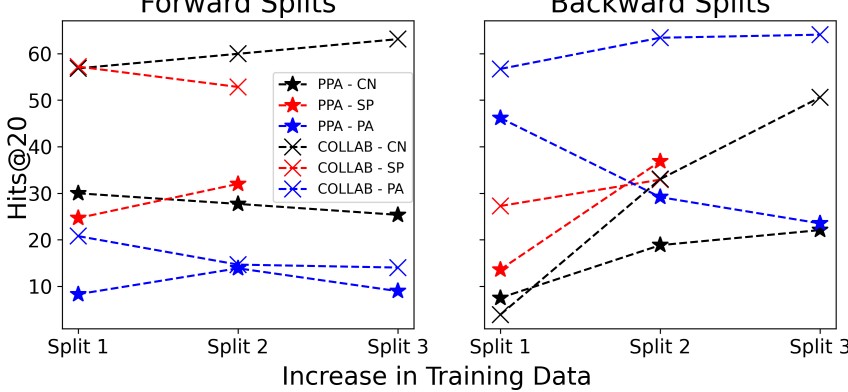

Figure 3: The mean scores of the best-performing GNN4LP models tested with our proposed splitting strategy. Each line represents a given dataset and split, arranged uniformly between figures. In the case of decreasing performance, the model with the highest average values was selected.

## 4.3 RESULTS FOR GENERALIZATION METHODS

In this section, we apply DropEdge (Rong et al., 2020), EPS (Singh et al., 2021), and TC (Zhao et al., 2023) on the previously benchmarked GCN (Kipf & Welling, 2017) and BUDDY (Chamberlain et al., 2022) to determine the feasibility of improving the LP models' generalization under our LPShift.

Table 2: The mean change in performance for all splits with a given generalization method applied to GCN.

| Methods | ogbl-collab | | | ogbl-ppa | | |
| --- | --- | --- | --- | --- | --- | --- |
| | Forward | backward | Overall | Forward | backward | Overall |
| DropEdge | **+4%** | **+2%** | **+3%** | **-1%** | 0% | **-0.5%** |
| EPS | **-39%** | **-40%** | **-40%** | **-35%** | **-15%** | **-25%** |
| TC | **-9%** | **-35%** | **-22%** | 0% | 0% | 0% |

**Observation 1: LP-Specific Generalization Methods Struggle**. As demonstrated in Table 2, the two generalization methods specific to LP: TC (Wang et al., 2023b) and EPS (Singh et al., 2021) fail to improve performance under LPShift. EPS always results in a decrease of performance from our baseline, indicating a failure to adjust for structural changes induced by LPShift. To validate this, we calculate Earth Mover's Distance (EMD) (Rubner et al., 1998) between the heuristic scores of the training and testing splits before and after applying the generalization methods. EPS injects CNs into the training adjacency matrix, significantly altering the training and testing distributions. This drastic change is indicated in Figures 4 and 7 with the difference between 'EPS' and 'LPShift' EMD scores. Such a change in EMD and the Table 2 results, demonstrate that generalizing under LPShift requires more than simply updating the training graph's structure.

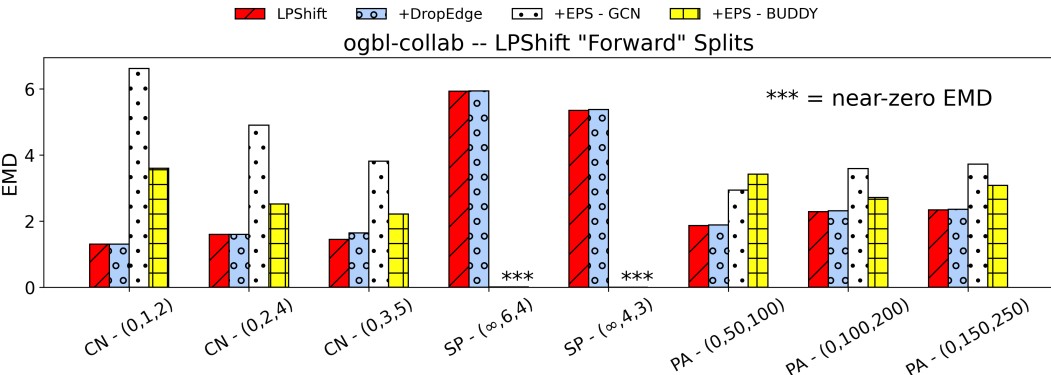

Figure 4: The EMD values calculated between the heuristic scores of training and testing samples. *Note:* The tested heuristics correspond to their labelled splits, so as to simulate the dataset splitting.

TC decreases ogbl-collab performance, with little effect on ogbl-ppa performance. This is likely due to LPShift's distinct split thresholds; meaning there is limited structural overlap between sample distributions. As such, TC can't re-weight the training adjacency matrix for improved generalization to neighborhood information (Wang et al., 2023b; Li et al., 2022a). This result runs contrary to current work, where re-weighting is effective for handling distribution shifts in other graph tasks (Zhou et al., 2022a) and computer vision (Fang et al., 2020).

**Observation 2: DropEdge Occasionally Works.** As demonstrated in Table 2, DropEdge (Rong et al., 2020) consistently improves performance on LPShift's ogbl-collab; with a small detrimental effect on ogbl-ppa. Figure 4 indicates that DropEdge causes little change in EMD between training and testing samples. Given that EPS significantly affects EMD scores, DropEdge improving performance is due to minor structural changes in the training adjacency matrix and limited effect on sample distributions. Additional EMD results are provided in Appendix Section I.

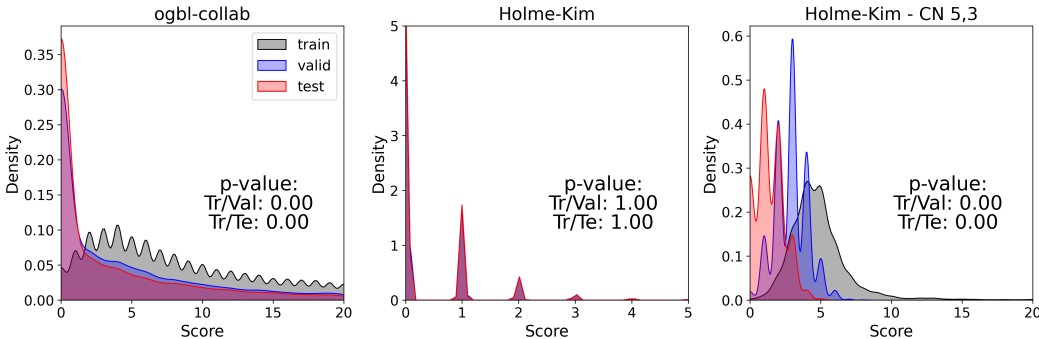

Figure 5: Three subplots detailing CN distributions for: 1.) the unaltered ogbl-collab dataset 2.) a Holme-Kim (HK) graph with a random split 3.) the HK graph from 2. split with LPShift's CN - 5,3 strategy

## 4.4 DISCUSSION

**How effective is LPShift at inducing distribution shifts?** The following section will explore the capability of LPShift to induce a measurably-significant distribution shift in the structure of the ogbl-collab dataset. We apply the 2-sample Kolmgorov-Smirnov (KS) test (Hodges Jr, 1958) to compare if training, validation, and testing distributions can be sampled from one another, both before and after applying LPShift. As a controlled baseline to test LPShift, we generate a Holme-Kim (HK) graph (Holme & Kim, 2002) with a 40% chance to close a triangle, allowing the HK graph to contain numerous Common Neighbors without becoming fully-connected. HK graph generation parameters are included in Appendix D.

The first subplot in Figure 5 extends the reasoning introduced with NCNC (Wang et al., 2023a). As such, this subplot indicates there is a natural shift for CNs within the original ogbl-collab dataset (Hu et al., 2020). The p-value of 0, measured across both split permutations, indicates that the training distribution of CNs is shifted from the validation and testing distributions. The second subplot depicts a randomly-split HK graph (Holme & Kim, 2002), where CN distributions for each split match one another, further indicated by the p-values of 1. The third subplot depicts the HK graph from the second subplot split with LPShift's CN - 5,3 strategy, resulting in a distinct shift between all dataset splits, as confirmed by the 0 p-values. As such, LPShift induces structural shift that is as measurably dissimilar as the structural shift present in the original ogbl-collab dataset, even when the initial dataset splits are measurably identical. Additionally, the CN - 5,3 split causes the shape of the HK graph's CN distributions to become more similar to the shift observed within the original ogbl-collab dataset, indicating that the "Backward" LPShift strategy can function like a real-world distribution shift.

**Does GNN4LP generalize and do generalization methods work?** As detailed in Section 4, basic GNN and Heuristic methods perform competitively compared to GNN4LP models. This observation coupled with the limited success of LP generalization methods, indicates the challenging problem LPShift poses. As shown in Appendix Section J, the "Backward" CN and SP splits on ogbl-collab and all ogbl-ppa splits result in a $30-90\%$ performance decrease from the HeaRT standard (Li et al., 2024). This is especially notable given the difficulty HeaRT imposes on the original benchmark setting (Hu et al., 2020). We also note that the majority of PA and forward splits for ogbl-collab result in a 2-3 times performance increase. This result quantifies the prevailing assumption that GNN4LP generalize well in scenarios with increasing structural information (Mao et al., 2024; Wang et al., 2023a). However, LPShift's ogbl-ppa suffers from the inverse scenario, where the majority of splits result in a $60-90\%$ decrease in performance. Therefore indicating that LPShift has stronger effects across dataset domains, particularly when the dataset is larger. DropEdge is the only method to (Rong et al., 2020) consistently improve LP performance when handling LPShift for the ogbl-collab dataset. Traditional generalization methods (Arjovsky et al., 2019; Sagawa et al., 2019; Ganin et al., 2016; Sun & Saenko, 2016), such as VREx (Krueger et al., 2021), achieve some promising results. However, these gains are typically marginal, especially on the "Backward" ogbl-collab split; as indicated in Appendix Section H. Given LPShift's impact on GNN4LP performance as well it's

resistance to current generalization methods, further study is required to eliminate the sensitivity current GNN4LP models have to structural shift.

**How is the proposed distribution shift affecting performance?** The EMD calculations for the training and testing samples show that simply adjusting training adjacency matrix structure is ineffective for improving performance under LPShift. To quantify how dataset structure affects performance, we follow a similar analysis conducted in (Wang et al., 2023a), as shown in Figure 6. In which, CN's predictive performance is measured under LPShift. to provide insight on targeted ways to impact dataset structure for better generalization.

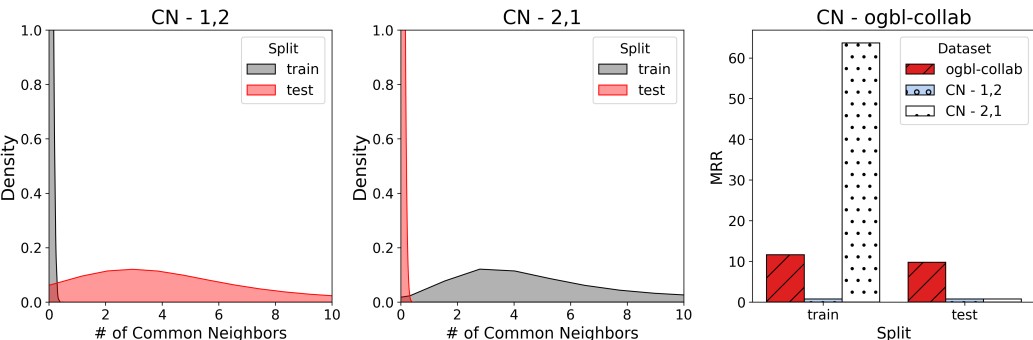

Figure 6: Three subplots corresponding to: 1.) the 'CN - 1,2' LPShift on ogbl-collab 2.) the 'CN - 2,1' LPShift on ogbl-collab 3.) CN predictor performance for the orginal ogbl-collab, 'CN - 1,2', and 'CN - 2,1' splits.

The first two subplots in Figure 6 depict the density estimates of the CN distribution for the CN - 1,2 and CN - 2,1 LPShift splits of the ogbl-collab dataset. The third subplot depicts the performance of CN on the datasets depicted in the first two subplots.

Given that LPShift only induces structural shift in affected datasets, it follows that the reduced performance (1% MRR) of the CN ranking, as indicated in Figure 6, on the CN - 1,2 split is due to the lack of structural information in the training split (CN = 0). After which, CN is required to rank predictions on valid and test splits with 1 and >2 CNs, respectively. The correlation between structure and performance becomes especially clear in the CN - 2,1 scenario. The training split contains 2 or more Common Neighbors, granting the CN predictor access to high amounts of structural information, achieving an MRR of roughly 60%. However, since the testing split consists of samples with zero Common Neighbors, the success on the training split does not transfer, resulting in an MRR of 1%. Given how many GNN4LP models incorporate neighborhood/structural information into their architectures (Wang et al., 2023a), it follows that GNN4LP's reduced performance on LPShift's forward splits is due to the absence of structural information induced by LPShift.

When one considers GNN4LP's performance under LPShift, along with results presented in Figure 4 and 6, then augmenting the original graph structure with additional structural information, may require a finer-scale focus than top-k CNs (Singh et al., 2021) to "repair" the missing structural components in the adjacency matrix and enhance generalization under LPShift. As such, it would be pertinent for LP models to extrapolate pairwise information from more varieties of structural information in order to improve performance in shifted scenarios.

## 5 CONCLUSION

This work proposes LPShift, a simple dataset splitting strategy for inducing structural shift relevant for link prediction. The effect of this structural shift was then benchmarked on 16 shifted versions of ogbl-collab and ogbl-ppa, posing a unique challenge for SOTA GNN4LP models and generalization methods. Further analysis with EMD calculations and CN distributions indicate that current generalization methods do not improve performance under structural shift. As such, LPShift provides a challenging problem requiring new considerations about how structure functions within link prediction architectures.

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

# A  REALISTIC SCENARIOS FOR STRUCTURAL SHIFT IN LINK PREDICTION

- **Adversarial Recommender Systems = Shortest-Path:** A company may want to understand which products to *avoid* showing to a potential customer without need to hear the user's preferences directly. In this scenario, global information, as captured by Shortest-Path, becomes the most valuable for the specific use-case.

- **Social Recommender Systems = Preferential-Attachment:** A video-streaming platform working with independent content creators may wish to understand what drives users to engage with the platform's content creators, so that engagement can increase for less-popular creators. As a starting point, the content creators with the most followers may have different characteristics that increase engagement versus less-popular creators. So, the streaming platform may wish to tune their dataset to determine if their recommendation system can generalize from more to less-popular creators.

- **Movie Recommender Systems = Common-Neighbors:** A movie-streaming platform wants to provide suggestions to users that are the most relevant to the user's current interests. So, the movie platform sorts possible movie recommendations by how much overlap the movies share with one another. In order to enhance exposure to new movies that overlap with a user's interests, the streaming platform can apply the Common Neighbor LPShift to their dataset to force the algorithm to generalize in a scenario where movies may not fully-overlap.

# B  HEURISTIC CHOICE

Resource Allocation and Adamic-Adar Index were not considered for splitting strategies given that they build upon the original Common Neighbor formulation. Their inclusion is redundant given our intentions to induce distinctive structural shifts based on varying structural information, as described for each heuristic in Section 3.

# C  SPLITTING STRATEGY – ADDITIONAL ALGORITHMIC DETAILS

This section provides additional details about the way data was formatted before being used as input for Algorithm 1 of our proposed splitting strategy and the intuition behind how Preferential-Attachment and Shortest-Path work within the splitting strategy. The details on the algorithm includes:

- Validation and Testing Edges are limited to 100k edges total.

- PPA Training Edges are limited to 3 million edges total.

- Negative Testing and Validation edges are produced via HeaRT (Li et al., 2024).

- Validation and testing edges that are duplicated with training edges are removed from the edge index.

- In order to provide overlap within a given dataset, validation and testing edges that do not connect to training nodes are removed from the edge index.

- After sampling the necessary training edges, the adjacency matrix is extracted from the edge index, converted to an undirected graph and has any edge weights standardized to 1.

Common Neighbors, Preferential-Attachment and Shortest-Path, as shown in Figure 2(a), 2(b), and 2(c) respectively, are interchangeable within the dataset splitting strategy. Details about how Common Neighbors functions within the strategy are included in Section 3.2. Figure 2(b) and Figure 2(c) serve as toy examples and do not correspond directly to any dataset splits tested within our study. However, the examples illustrated within Figure 2(b) and Figure 2(c) do correspond to how their given heuristic functions within our splitting strategy.

For Figure 2(b) or Preferential-Attachment, it determines the degrees between a given source and target node and then multiples the two to produce the score, based on that score, the sample is then sorted into a new dataset split.

For Figure 2(c) or Shortest-Path, the heuristic determines the score by determining the minimum number of nodes necessary to reach the target node from the source node. If there is a link between the two nodes, we remove the link and then re-add to the adjacency matrix after the score calculation. The final Shortest-Path score applies the calculated shortest-path length, $SP(u, v)$ as the denominator in a ratio of $\frac{1}{SP(u,v)}$, which is then used to sort the sample into it's respective dataset split.

## D  ADDITIONAL ANALYSIS DETAILS

Anonymous Code is available at the following link:Anonymous Link

*Note:* We were unable to test CFLP (Zhao et al., 2022) and EERM (Wu et al., 2022) after adapting their current implementations to our evaluation settings. CFLP experienced an out-of-memory error on all tested dataset splits. EERM experienced an out-of-memory error on every LPShift split of ogbl-ppa and exceeded 48 hours per run on ogbl-collab before converging.

The Holme-Kim graph used for analysis in Figure 5 was generated with the following parameters:

- $n = 235868$, $m = 5$, $p_c = 0.4$, seed = 42

Table 3: The cosine-similarity of ogbl-collab's original feature distribution and it's LPShift versions. Test permutations include: Train and Test, Train and Valid, Valid and Test. *Note:* ogbl-ppa is not measured given it's use of one-hot encoded feature vectors (Hu et al., 2020)

| Split Type | | CN Splits | | | SP Splits | | | PA Splits | |
|---|---|---|---|---|---|---|---|---|---|
| 'Forward' | Original | (0, 1, 2) | (0, 2, 4) | (0, 3, 5) | $(\infty, 6, 4)$ | $(\infty, 4, 3)$ | (0, 50, 100) | (0, 100, 200) | (0, 150, 250) |
| Train/Test | 83.50 ± 7.33 | 87.73 ± 6.12 | 87.06 ± 6.08 | 86.45 ± 6.12 | 84.11 ± 6.94 | 84.35 ± 6.93 | 83.92 ± 7.22 | 87.48 ± 6.16 | 87.61 ± 6.10 |
| Train/Valid | 83.40 ± 7.33 | 87.42 ± 6.17 | 86.72 ± 6.14 | 86.19 ± 6.12 | 82.12 ± 7.34 | 82.66 ± 7.45 | 82.59 ± 7.63 | 87.55 ± 6.1 | 87.81 ± 6.17 |
| Valid/Test | 86.91 ± 6.58 | 91.78 ± 4.01 | 91.01 ± 4.29 | 90.42 ± 4.41 | 85.60 ± 6.75 | 85.97 ± 6.75 | 85.50 ± 7.14 | 92.52 ± 3.70 | 92.45 ± 3.84 |
| 'Backward' | Original | (2, 1, 0) | (4, 2, 0) | (5, 3, 0) | $(4, 6, \infty)$ | $(3, 4, \infty)$ | (100, 50, 0) | (200, 100, 0) | (250, 150, 0) |
| Train/Test | 83.50 ± 7.33 | 85.53 ± 6.74 | 85.97 ± 7.09 | 86.08 ± 7.29 | 81.10 ± 7.74 | 81.99 ± 7.84 | 82.28 ± 7.96 | 84.75 ± 6.65 | 85.08 ± 6.65 |
| Train/Valid | 83.40 ± 7.33 | 85.22 ± 6.79 | 85.55 ± 7.15 | 85.62 ± 7.34 | 82.15 ± 7.79 | 82.81 ± 7.98 | 83.22 ± 8.00 | 85.32 ± 6.64 | 85.51 ± 6.64 |
| Valid/Test | 86.91 ± 6.58 | 90.45 ± 4.49 | 90.71 ± 4.43 | 90.74 ± 4.53 | 81.86 ± 7.48 | 82.76 ± 7.72 | 83.17 ± 7.85 | 89.94 ± 4.52 | 90.62 ± 4.46 |

## E  TIME TO SPLIT TESTED DATASET SAMPLES

A key consideration for LPShift's application as a splitting strategy is to alleviate the burden of gathering new datasets, allowing researchers to control for and then induce a distribution shift in link-prediction datasets quickly; without requiring an expensive and time-consuming project to build a new dataset. This consideration is inspired by current graph and node-classification benchmark datasets, all of which induce distribution shifts in pre-existing benchmark datasets (Gui et al., 2022),(Ji et al., 2022),(Koh et al., 2021). LPShift is not meant to replace high-quality benchmark datasets, especially for distribution shifts, but to serve as a supplement for current datasets and enhance understanding of LP generalization. Results demonstrating LPShift's time-efficiency on tested dataset splits are included below in Table 4.

Table 4: The average time in seconds (s) across 10 runs to generate each 'Forward' and 'Backward' split for ogbl-ppa and ogbl-collab.

| 'Forward' | CN Splits | | | SP Splits | | PA Splits | | |
|---|---|---|---|---|---|---|---|---|
| | (0, 1, 2) | (0, 2, 4) | (0, 3, 5) | $(\infty, 6, 4)$ | $(\infty, 4, 3)$ | (0, 50, 100) | (0, 100, 200) | (0, 150, 250) |
| ogbl-collab | 7.48 s | 7.49 s | 7.63 s | 53.12 s | 52.24 s | 19.25 s | 19.05 s | 19.35 s |
| ogbl-ppa | 177.89 s | 177.09 s | 178.23 s | 2748.64 s | 2705.91 s | 406.5 s | 408.04 s | 407.81 s |
| 'Backward' | (2, 1, 0) | (4, 2, 0) | (5, 3, 0) | $(4, 6, \infty)$ | $(3, 4, \infty)$ | (100, 50, 0) | (200, 100, 0) | (250, 150, 0) |
| ogbl-collab | 7.66 s | 7.63 s | 7.86 s | 53.93 s | 53.7 s | 19.65 s | 19.34 s | 19.37 s |
| ogbl-ppa | 184.98 s | 186.49 s | 185.95 s | 2715.34 s | 2751.56 s | 425.3 s | 409.93 s | 403.55 s |

# F  SIZE OF DATASET SAMPLES

In this section we detail the number of training, validation, and test edges for all of the newly created splits detailed in Section 3. There are in Tables 5 and 6 for ogbl-collab and ogbl-ppa, respectively.

Table 5: Number of samples in the ogbl-collab dataset for the forward and backward heuristic splits, separated into columns allocated for training, validation, and testing splits.

| Heuristic | Split | Train | Valid | Test |
|---|---|---|---|---|
| CN | (0, 1, 2) | 57638 | 6920 | 4326 |
|  | (0, 2, 4) | 237928 | 20045 | 14143 |
|  | (0, 3, 5) | 493790 | 31676 | 21555 |
|  | (2, 1, 0) | 1697336 | 23669 | 9048 |
|  | (4, 2, 0) | 1193456 | 24097 | 11551 |
|  | (5, 3, 0) | 985820 | 25261 | 11760 |
| SP | ($\infty$, 6, 4) | 46880 | 1026 | 2759 |
|  | ($\infty$, 4, 3) | 52872 | 1238 | 3457 |
|  | (4, 6, $\infty$) | 1882392 | 5222 | 2626 |
|  | (3, 4, $\infty$) | 1877626 | 4384 | 7828 |
| PA | (0, 50, 100) | 210465 | 46626 | 9492 |
|  | (0, 100, 200) | 329383 | 62868 | 25527 |
|  | (0, 150, 250) | 409729 | 75980 | 39429 |
|  | (100, 50, 0) | 1882392 | 64729 | 41381 |
|  | (200, 100, 0) | 1877626 | 64202 | 30983 |
|  | (250, 150, 0) | 457372 | 65323 | 30999 |

Table 6: Number of samples in the ogbl-ppa dataset for the forward and backward heuristic splits, separated into columns allocated for training, validation, and testing splits

| Heuristic | Split | Train | Valid | Test |
|---|---|---|---|---|
| CN | (0, 1, 2) | 2325936 | 87880 | 67176 |
|  | (0, 2, 4) | 3000000 | 95679 | 83198 |
|  | (0, 3, 5) | 3000000 | 98081 | 88778 |
|  | (2, 1, 0) | 3000000 | 96765 | 92798 |
|  | (4, 2, 0) | 3000000 | 93210 | 85448 |
|  | (5, 3, 0) | 3000000 | 92403 | 81887 |
| SP | ($\infty$, 6, 4) | 17464 | 149 | 20 |
|  | ($\infty$, 4, 3) | 134728 | 4196 | 1180 |
|  | (4, 6, $\infty$) | 3000000 | 90511 | 458 |
|  | (3, 4, $\infty$) | 3000000 | 97121 | 74068 |
| PA | (0, 5k, 10k) | 95671 | 95671 | 45251 |
|  | (0, 10k, 20k) | 98562 | 98562 | 63178 |
|  | (0, 15k, 25k) | 99352 | 99352 | 72382 |
|  | (10k, 5k, 0) | 3000000 | 90623 | 44593 |
|  | (20k, 10k, 0) | 3000000 | 89671 | 34321 |
|  | (25k, 15k, 0) | 3000000 | 91995 | 35088 |

## G DATASET RESULTS

In this section, we include all of the results for each experiment conducted on the generalization methods and EMD calculations. Results from Tables 15, 16, 17, and 18 were used for the calculations demonstrated in Figure 2. Figure 4 was constructed from results within Table 21.

Table 7: **ogbl-collab** results on the **Forward** splits. Results reported in MRR with the best **bolded** and the second best underlined.

| Models | CN Splits | | | SP Splits | | PA Splits | | |
|---|---|---|---|---|---|---|---|---|
| | (0, 1, 2) | (0, 2, 4) | (0, 3, 5) | ($\infty$, 6, 4) | ($\infty$, 4, 3) | (0, 50, 100) | (0, 100, 200) | (0, 150, 250) |
| RA | **32.22** | **29.74** | **29.86** | **33.87** | **33.91** | **36.87** | **26.78** | **24.07** |
| GCN | 12.92 ± 0.31 | 15.20 ± 0.16 | 17.54 ± 0.19 | 10.29 ± 0.52 | 12.94 ± 0.59 | 20.78 ± 0.25 | 14.66 ± 0.20 | 14.03 ± 0.15 |
| BUDDY | 17.48 ± 1.19 | 15.47 ± 0.57 | 16.60 ± 0.89 | 16.20 ± 1.40 | 16.42 ± 2.30 | 21.27 ± 0.74 | 14.04 ± 0.76 | 13.06 ± 0.53 |
| NCNC | 9.00 ± 1.02 | 13.99 ± 1.35 | 15.04 ± 1.25 | 14.43 ± 1.36 | 18.33 ± 1.24 | 12.76 ± 1.60 | 6.66 ± 1.24 | 6.48 ± 1.53 |
| LPFormer | 4.27 ± 1.17 | 13.7 ± 1.48 | 25.36 ± 2.04 | 4.6 ± 3.15 | 15.7 ± 2.87 | 25.31 ± 5.67 | 11.98 ± 3.12 | 12.43 ± 6.62 |
| NeoGNN | 5.76 ± 1.69 | 16.10 ± 0.82 | 18.22 ± 0.61 | 6.11 ± 0.73 | 6.67 ± 1.16 | 16.65 ± 0.26 | 11.64 ± 0.49 | 12.25 ± 0.95 |
| SEAL | 7.06 ± 1.67 | 20.60 ± 6.23 | 19.78 ± 1.24 | 2.08 ± 1.50 | 2.10 ± 1.24 | 29.06 ± 1.57 | 20.69 ± 1.51 | 16.23 ± 3.69 |

Table 8: **ogbl-collab** results on the **Forward** splits. Results reported in Hits@20 with the best **bolded** and the second best underlined.

| Models | CN Splits | | | SP Splits | | PA Splits | | |
|---|---|---|---|---|---|---|---|---|
| | (0, 1, 2) | (0, 2, 4) | (0, 3, 5) | ($\infty$, 6, 4) | ($\infty$, 4, 3) | (0, 50, 100) | (0, 100, 200) | (0, 150, 250) |
| RA | **79.84** | **79.12** | **79.38** | **80.65** | **79.98** | **75.94** | **67.13** | **62.19** |
| GCN | 40.36 ± 1.58 | 50.49 ± 0.40 | 57.09 ± 0.69 | 38.11 ± 2.48 | 45.52 ± 1.62 | 62.95 ± 0.37 | 48.51 ± 0.46 | 45.92 ± 0.44 |
| BUDDY | 56.81 ± 1.99 | 59.49 ± 0.86 | 63.09 ± 0.63 | 52.82 ± 3.76 | 57.19 ± 3.46 | 60.19 ± 1.66 | 46.52 ± 1.55 | 43.10 ± 0.94 |
| NCNC | 37.03±1.34 | 46.05±1.46 | 49.76±1.13 | 48.31±2.95 | 56.52±3.10 | 47.22 ± 3.18 | 28.13 ± 4.06 | 24.53 ± 3.48 |
| LPFormer | 10.93±3.74 | 54.11±3.85 | 57.43±3.92 | 22.15±4.83 | 50.16±2.96 | 58.83 ± 8.89 | 33.39 ± 5.84 | 34.45 ± 11.73 |
| NeoGNN | 27.51 ± 11.67 | 42.50 ± 8.68 | 53.85 ± 0.96 | 17.69 ± 7.32 | 31.02 ± 10.15 | 53.23 ± 1.03 | 38.49 ± 0.83 | 35.11 ± 1.26 |
| SEAL | 25.90 ± 5.29 | 53.66 ± 12.69 | 56.48 ± 4.47 | 4.41 ± 3.16 | 6.37 ± 2.45 | 63.25 ± 2.49 | 53.80 ± 2.72 | 45.27 ± 7.50 |

Table 9: **ogbl-ppa** results on the **Forward** splits. Results reported in MRR with the best **bolded** and the second best underlined.

| Models | CN Splits | | | SP Splits | | PA Splits | | |
|---|---|---|---|---|---|---|---|---|
| | (0, 1, 2) | (0, 2, 4) | (0, 3, 5) | ($\infty$, 6, 4) | ($\infty$, 4, 3) | (0, 5k, 10k) | (0, 10k, 20k) | (0, 15k, 25k) |
| RA | 4.71 | 4.45 | 4.38 | **32.57** | **19.84** | 3.9 | 3.14 | 2.72 |
| GCN | 8.13 ± 0.38 | **7.51 ± 0.32** | 7.12 ± 1.05 | 5.40 ± 0.57 | 5.56 ± 0.21 | 4.19 ± 0.43 | 4.98 ± 0.52 | 5.95 ± 0.28 |
| BUDDY | 7.90 ± 0.32 | 3.83 ± 0.24 | 3.06 ± 0.06 | 1.24 ± 0.02 | 5.87 ± 0.16 | 3.93 ± 0.98 | 6.38 ± 3.48 | 2.48 ± 0.03 |
| NCNC | 4.26 ± 0.45 | 6.87 ± 0.36 | 6.32 ± 0.57 | 8.91 ± 7.46 | 5.55 ± 0.45 | 8.00 ± 0.60 | 6.90 ± 1.46 | 8.00 ± 0.78 |
| LPFormer | 3.28 ± 0.63 | 2.46 ± 0.51 | 4.84 ± 0.73 | 9.83 ± 5.92 | 4.94 ± 0.62 | **9.27 ± 1.78** | **9.03 ± 1.6** | **9.07 ± 2.43** |
| NeoGNN | 4.50 ± 0.45 | 5.86 ± 2.87 | **10.60 ± 3.54** | 3.13 ± 0.38 | 3.58 ± 0.45 | 4.92 ± 0.58 | 6.29 ± 0.87 | 8.98 ± 1.10 |
| SEAL | **11.91 ± 1.85** | 4.84 ± 0.10 | 5.15 ± 0.10 | 11.14 ± 12.06 | 2.96 ± 4.58 | 4.22 ± 0.63 | 3.43 ± 0.19 | 3.57 ± 0.74 |

Table 10: **ogbl-ppa** results on the **Forward** splits. Results reported in Hits@20 with the best **bolded** and the second best underlined.

| Models | CN Splits | | | SP Splits | | PA Splits | | |
|---|---|---|---|---|---|---|---|---|
| | (0, 1, 2) | (0, 2, 4) | (0, 3, 5) | ($\infty$, 6, 4) | ($\infty$, 4, 3) | (0, 5k, 10k) | (0, 10k, 20k) | (0, 15k, 25k) |
| RA | 18.09 | 17.01 | 16.71 | **90** | 63.56 | 14.93 | 11.16 | 9.15 |
| GCN | 29.98 ± 1.37 | **27.70 ± 1.16** | **25.35 ± 2.38** | 26.00 ± 6.52 | 27.00 ± 0.57 | 14.80 ± 1.08 | 16.49 ± 0.83 | 21.06 ± 0.57 |
| BUDDY | 26.42 ± 1.21 | 15.00 ± 1.38 | 11.38 ± 0.32 | 23.00 ± 5.70 | 23.95 ± 0.23 | 10.87 ± 1.57 | 14.37 ± 6.07 | 7.38 ± 0.15 |
| NCNC | 19.94±1.43 | 25.51±0.96 | 23.23±2.01 | 32.00 ± 17.54 | 24.66±2.15 | 21.00 ± 1.21 | 16.31 ± 2.03 | 17.76 ± 1.45 |
| LPFormer | 8.12±1.27 | 9.85±1.49 | 15.06±1.84 | 35±14.83 | 16.36±2.74 | **22.35 ± 5.09** | **20.37 ± 3.15** | 22.03 ± 5.45 |
| NeoGNN | 21.80 ± 2.66 | 18.24 ± 3.22 | 19.74 ± 3.93 | 20.00 ± 6.12 | 16.76 ± 2.96 | 15.50 ± 0.19 | 18.26 ± 1.54 | **22.22 ± 0.78** |
| SEAL | **30.76 ± 5.02** | 18.79 ± 0.25 | 19.28 ± 0.47 | 23.00 ± 18.91 | 10.85 ± 17.73 | 13.19 ± 1.43 | 10.62 ± 1.96 | 10.93 ± 2.08 |

Table 11: **ogbl-collab** results on the **Backward** splits. Results reported in MRR with the best **bolded** and the second best underlined.

| Models | CN Splits | | | SP Splits | | PA Splits | | |
|---|---|---|---|---|---|---|---|---|
| | (2, 1, 0) | (4, 2, 0) | (5, 3, 0) | (4, 6, ∞) | (3, 4, ∞) | (100, 50, 0) | (200, 100, 0) | (250, 150, 0) |
| RA | 0.6 | 4.79 | 15.9 | 0.69 | 0.63 | **33.09** | 42.28 | **44.14** |
| GCN | **5.78 ± 0.19** | 5.91 ± 0.05 | 7.07 ± 0.11 | **8.69 ± 0.47** | **6.47 ± 0.37** | 21.38 ± 0.27 | 13.36 ± 0.24 | 11.95 ± 0.35 |
| BUDDY | 3.70 ± 0.13 | 3.55 ± 0.09 | 5.40 ± 0.04 | 6.73 ± 0.32 | 3.71 ± 0.39 | 24.95 ± 0.92 | 15.52 ± 0.73 | 13.36 ± 0.83 |
| NCNC | 1.89 ± 1.27 | **16.48 ± 1.30** | **19.69 ± 1.52** | 1.62 ± 0.72 | 1.08 ± 0.73 | 14.86 ± 1.59 | 18.67 ± 2.75 | 17.39 ± 2.12 |
| LPFormer | 2.01 ± 0.95 | 3.87 ± 0.74 | 8.36 ± 0.63 | 3.16 ± 0.62 | 1.86 ± 0.48 | 17.76 ± 2.01 | 27.56 ± 9.10 | 24.04 ± 11.35 |
| NeoGNN | 2.14 ± 0.05 | 3.48 ± 0.10 | 9.44 ± 0.38 | 4.58 ± 0.30 | 2.55 ± 0.10 | 13.58 ± 1.12 | 7.65 ± 0.45 | 7.86 ± 1.52 |
| SEAL | 1.01 ± 0.02 | 3.38 ± 0.62 | 8.08 ± 1.96 | 0.93 ± 0.07 | 0.80 ± 0.01 | 31.83 ± 6.44 | 31.96 ± 6.65 | 39.58 ± 4.81 |

Table 12: **ogbl-collab** results on the **Backward** splits. Results reported in Hits@20 with the best **bolded** and the second best underlined.

| Models | CN Splits | | | SP Splits | | PA Splits | | |
|---|---|---|---|---|---|---|---|---|
| | (2, 1, 0) | (4, 2, 0) | (5, 3, 0) | (4, 6, ∞) | (3, 4, ∞) | (100, 50, 0) | (200, 100, 0) | (250, 150, 0) |
| RA | 0 | 11.21 | 34.95 | 0 | 0 | 66.81 | **74.92** | 78.81 |
| GCN | **24.14 ± 0.72** | 22.85 ± 0.41 | 26.98 ± 0.20 | **32.91 ± 0.79** | **27.25 ± 0.38** | 69.43 ± 0.25 | 56.29 ± 0.71 | 54.56 ± 0.67 |
| BUDDY | 12.90 ± 0.29 | 11.69 ± 0.41 | 19.32 ± 0.51 | 25.05 ± 1.00 | 13.46 ± 1.22 | **73.41 ± 1.53** | 58.93 ± 1.61 | 59.17 ± 2.77 |
| NCNC | 3.91±1.94 | **33.02±2.05** | **50.6±2.69** | 4.11±0.85 | 1.33±0.83 | 53.49 ± 2.87 | 62.82 ± 5.22 | 62.57 ± 4.78 |
| LPFormer | 4.8±1.84 | 11.06±1.67 | 25.81±1.47 | 11.5±1.47 | 5.16±0.73 | 56.69 ± 2.57 | 63.39 ± 10.27 | 64.05 ± 8.47 |
| NeoGNN | 6.73 ± 0.31 | 11.42 ± 0.34 | 29.28 ± 0.88 | 18.32 ± 1.11 | 9.16 ± 0.62 | 43.72 ± 1.71 | 29.93 ± 2.17 | 30.89 ± 2.46 |
| SEAL | 1.88 ± 1.69 | 11.55 ± 2.71 | 23.21 ± 3.40 | 1.14 ± 0.61 | 0.00 ± 0.00 | 72.39 ± 13.93 | 73.82 ± 4.29 | **82.46 ± 3.08** |

Table 13: **ogbl-ppa** results on the **Backward** splits. Results reported in MRR with the best **bolded** and the second best underlined.

| Models | CN Splits | | | SP Splits | | PA Splits | | |
|---|---|---|---|---|---|---|---|---|
| | (2, 1, 0) | (4, 2, 0) | (5, 3, 0) | (4, 6, ∞) | (3, 4, ∞) | (10k, 5k, 0) | (20k, 10k, 0) | (25k, 15k, 0) |
| RA | 0.53 | 0.92 | 1.17 | 0.65 | 0.54 | 7.4 | 5.81 | 5.08 |
| GCN | 3.52 ± 0.09 | 3.02 ± 0.09 | 2.94 ± 0.05 | **10.53 ± 0.48** | **3.38 ± 0.11** | 1.55 ± 0.07 | 1.29 ± 0.02 | 1.28 ± 0.03 |
| BUDDY | 1.60 ± 0.05 | 2.47 ± 0.07 | 2.56 ± 0.08 | 9.91 ± 0.32 | 3.03 ± 0.06 | 3.15 ± 0.16 | 2.55 ± 0.16 | 2.37 ± 0.02 |
| NCNC | 2.37 ± 0.15 | **8.54 ± 0.74** | **9.04 ± 0.92** | 5.56 ± 1.02 | 1.34 ± 0.56 | 7.33 ± 0.74 | 6.02 ± 0.85 | 5.55 ± 0.77 |
| LPFormer | **6.04 ± 0.41** | 4.23 ± 0.46 | 3.87 ± 0.1 | 5.9 ± 1.76 | 1.38 ± 0.46 | **14.43 ± 4.45** | **8.43 ± 3.46** | **6.27 ± 3.87** |
| NeoGNN | 0.76 ± 0.02 | 0.79 ± 0.00 | 0.86 ± 0.02 | 4.89 ± 0.13 | 0.83 ± 0.01 | 1.52 ± 0.05 | 1.38 ± 0.04 | 1.39 ± 0.06 |
| SEAL | 1.03 ± 0.54 | 0.95 ± 0.09 | 1.35 ± 0.56 | 1.51 ± 0.72 | 0.51 ± 0.02 | 4.88 ± 0.90 | 4.50 ± 1.10 | 2.38 ± 0.73 |

Table 14: **ogbl-ppa** results on the **Backward** splits. Results reported in Hits@20 with the best **bolded** and the second best underlined.

| Models | CN Splits | | | SP Splits | | PA Splits | | |
|---|---|---|---|---|---|---|---|---|
| | (2, 1, 0) | (4, 2, 0) | (5, 3, 0) | (4, 6, ∞) | (3, 4, ∞) | (10k, 5k, 0) | (20k, 10k, 0) | (25k, 15k, 0) |
| RA | 0 | 1.03 | 1.89 | 0 | 0 | 28.74 | 23.36 | 20.28 |
| GCN | 13.87 ± 0.40 | 0.76 ± 0.40 | 9.87 ± 0.09 | 32.75 ± 0.86 | **15.45 ± 0.46** | 3.01 ± 0.41 | 1.58 ± 0.18 | 1.57 ± 0.18 |
| BUDDY | 3.22 ± 0.25 | 8.05 ± 0.42 | 8.06 ± 0.45 | **36.86 ± 1.32** | 13.60 ± 0.43 | 12.83 ± 0.79 | 8.84 ± 0.95 | 8.14 ± 0.15 |
| NCNC | 7.52±0.46 | **18.84±1.83** | **22.08±1.47** | 30.35±2.84 | 2.95±0.83 | 29.52 ± 2.86 | 22.41 ± 2.09 | 19.65 ± 2.61 |
| LPFormer | **15.17±0.84** | 12.12±0.73 | 11.73±0.79 | 24.24±3.02 | 2.45±0.72 | **46.16 ± 10.94** | **29.15 ± 14.20** | **23.51 ± 11.70** |
| NeoGNN | 0.01 ± 0.00 | 0.08 ± 0.01 | 0.37 ± 0.06 | 18.25 ± 0.81 | 0.50 ± 0.03 | 3.15 ± 0.30 | 2.13 ± 0.25 | 2.33 ± 0.33 |
| SEAL | 0.95 ± 1.47 | 0.98 ± 0.27 | 1.69 ± 1.16 | 5.15 ± 5.22 | 0.00 ± 0.00 | 15.78 ± 4.48 | 17.43 ± 3.42 | 8.52 ± 3.15 |

Table 15: ogbl-collab results on the forward and backward splits when using DropEdge and TC.

| | | DropEdge | | TC | |
|---|---|---|---|---|---|
| Heuristic | Split | GCN | BUDDY | GCN | BUDDY |
| CN | (0, 1, 2) | 13.92 ± 0.78 | 15.54 ± 0.98 | 12.26± 0.28 | 11.27 ± 2.03 |
| | (0, 2, 4) | 15.85 ± 0.25 | 16.16 ± 0.17 | 12.62± 0.57 | 12.39 ± 1.43 |
| | (0, 3, 5) | 17.75 ± 0.11 | 16.34 ± 0.17 | 13.3± 0.47 | 14.99 ± 1.69 |
| | (2, 1, 0) | 5.96 ± 0.17 | 2.61 ± 0.09 | 4.95 ±0.16 | 5.28 ± 0.05 |
| | (4, 2, 0) | 6.14 ± 0.065 | 2.88 ± 0.11 | 5.37 ±0.13 | 5.23 ± 0.05 |
| | (5, 3, 0) | 7.20 ± 0.15 | 4.99 ± 0.08 | 6.04± 0.07 | 6.03 ± 0.10 |
| SP | (∞, 6, 4) | 11.94 ± 0.46 | 12.31 ± 0.51 | 11.42 ±0.43 | 7.25 ± 0.81 |
| | (∞, 4, 3) | 13.87 ± 0.43 | 17.11 ± 1.02 | 12.88± 0.43 | 9.33 ± 1.66 |
| | (4, 6, ∞) | 9.18 ± 0.52 | 5.34 ± 0.43 | 9.09± 4.74 | 6.88 ± 0.30 |
| | (3, 4, ∞) | 6.71 ± 0.13 | 2.93 ± 0.18 | 3.57± 2.30 | 6.24 ± 0.13 |
| PA | (0, 50, 100) | 20.76 ± 0.19 | 21.35 ± 0.36 | 17.55±0.57 | 18.82 ± 1.35 |
| | (0, 100, 200) | 14.57 ± 0.20 | 13.84 ± 0.64 | 13.22±1.1 | 12.13±1.04 |
| | (0, 150, 250) | 13.78 ± 0.28 | 12.85 ± 0.78 | 13.03±0.24 | 10.63±0.55 |
| | (100, 50, 0) | 21.34 ± 0.65 | 26.09 ± 0.62 | 6.4 ± 0.2 | 15.84 ± 1.13 |
| | (200, 100, 0) | 12.89 ± 0.59 | 15.68 ± 0.85 | 4.3 ± 0.14 | 9.15 ± 0.39 |
| | (250, 150, 0) | 11.68 ± 0.37 | 13.13 ± 0.94 | 4.4 ± 0.16 | 6.7 ± 0.2 |

Table 16: ogbl-ppa results on the forward and backward splits when using DropEdge and TC.

| | | DropEdge | | TC | |
|---|---|---|---|---|---|
| Heuristic | Split | GCN | BUDDY | GCN | BUDDY |
| CN | (0, 1, 2) | 8.20 ± 0.34 | 7.83 ± 0.27 | OOM | 5.27 ± 0.34 |
| | (0, 2, 4) | 7.39 ± 0.33 | 3.83 ± 0.25 | OOM | 2.91 ± 0.06 |
| | (0, 3, 5) | 6.04 ± 0.32 | 3.06 ± 0.06 | OOM | 2.67 ± 0.13 |
| | (2, 1, 0) | 3.50 ± 0.16 | 1.61 ± 0.04 | OOM | 3.44 ± 0.08 |
| | (4, 2, 0) | 3.01 ± 0.07 | 2.47 ± 0.07 | OOM | 3.45 ± 0.1 |
| | (5, 3, 0) | 2.97 ± 0.06 | 2.56 ± 0.08 | OOM | 3.55 ± 0.13 |
| SP | (∞, 6, 4) | 6.17 ± 0.76 | 3.86 ± 0.39 | OOM | 4.0 ± 0.29 |
| | (∞, 4, 3) | 5.55 ± 0.22 | 5.87 ± 0.16 | OOM | 4.82 ± 0.39 |
| | (4, 6, ∞) | 3.44 ± 0.17 | 3.86 ± 0.39 | OOM | 13.2 ± 0.45 |
| | (3, 4, ∞) | 15.69 ± 0.54 | 5.87 ± 0.16 | OOM | 2.89 ± 0.09 |
| PA | (0, 50, 100) | 4.19 ± 0.43 | 3.93 ± 0.98 | OOM | 3.62 ± 0.21 |
| | (0, 10k, 20k) | 4.98 ± 0.52 | 6.38 ± 3.48 | OOM | 3.13 ± 0.12 |
| | (0, 15k, 25k) | 5.95 ± 0.28 | 2.49 ± 0.01 | OOM | 3.33 ± 1.38 |
| | (10k, 5k, 0) | 1.51 ± 0.02 | 3.13 ± 0.1 | OOM | 1.78 ± 0.16 |
| | (20k, 10k, 0) | 1.25 ± 0.07 | 2.56 ± 0.19 | OOM | 1.5 ± 0.0008 |
| | (25k, 15k, 0) | 1.28 ± 0.03 | 2.40 ± 0.03 | OOM | 1.53 ± 0.04 |

Table 17: ogbl-collab results on the forward and backward splits when using EPS for each given Filter + Rank model configuration.

| Heuristic | Split | GCN + BUDDY | BUDDY + GCN | RA + GCN | RA + BUDDY |
|---|---|---|---|---|---|
| CN | (0, 1, 2) | 8.50 ± 1.10 | 5.81 ± 0.11 | 7.42 ± 0.12 | 3.94 ± 0.51 |
| | (0, 2, 4) | 12.85 ± 0.83 | 5.31 ± 0.22 | 7.07 ± 0.16 | 6.61 ± 0.30 |
| | (0, 3, 5) | 15.35 ± 0.96 | 5.88 ± 0.16 | 7.52 ± 0.13 | 7.16 ± 0.09 |
| | (2, 1, 0) | 5.46 ± 0.11 | 4.90 ± 0.08 | 5.24 ± 0.14 | 4.32 ± 0.19 |
| | (4, 2, 0) | 5.36 ± 0.12 | 4.62 ± 0.16 | 5.27 ± 0.09 | 4.45 ± 0.12 |
| | (5, 3, 0) | 5.96 ± 0.06 | 5.17 ± 0.09 | 5.15 ± 0.26 | 4.93 ± 0.11 |
| SP | (∞, 6, 4) | 7.38 ± 0.82 | 7.10 ± 0.43 | 7.38 ± 0.42 | 3.84 ± 0.59 |
| | (∞, 4, 3) | 9.60 ± 0.39 | 6.24 ± 0.52 | 7.07 ± 0.46 | 7.63 ± 0.25 |
| | (4, 6, ∞) | 6.86 ± 1.13 | 6.51 ± 0.24 | 6.11 ± 0.48 | 6.86 ± 1.13 |
| | (3, 4, ∞) | 6.47 ± 0.24 | 4.86 ± 0.38 | 5.23 ± 0.23 | 6.47 ± 0.24 |
| PA | (0, 50, 100) | 15.92 ± 1.01 | 13.84 ± 0.14 | 13.23 ± 0.21 | 15.92 ± 1.01 |
| | (0, 100, 200) | 9.47 ± 0.31 | 10.85 ± 0.11 | 10.71 ± 0.10 | 9.47 ± 0.31 |
| | (0, 150, 250) | 9.60 ± 0.41 | 10.33 ± 0.15 | 9.96 ± 0.06 | 9.60 ± 0.41 |
| | (100, 50, 0) | 14.34 ± 1.05 | 5.23 ± 0.28 | 5.07 ± 0.21 | 14.34 ± 1.05 |
| | (200, 100, 0) | 8.35 ± 0.34 | 3.06 ± 0.08 | 2.93 ± 0.06 | 8.35 ± 0.34 |
| | (250, 150, 0) | 5.50 ± 0.33 | 3.14 ± 0.14 | 2.79 ± 0.07 | 5.50 ± 0.33 |

Table 18: ogbl-ppa results on the forward and backward splits when using EPS for each given Filter + Rank model configuration.

| Heuristic | Split | GCN + BUDDY | BUDDY + GCN | RA + GCN | RA + BUDDY |
|---|---|---|---|---|---|
| CN | (0, 1, 2) | 4.48 ± 0.33 | OOM | 3.53 ± 0.03 | 4.04 ± 0.26 |
| | (0, 2, 4) | 3.79 ± 0.28 | OOM | 3.35 ± 0.03 | 3.42 ± 0.20 |
| | (0, 3, 5) | 3.16 ± 0.10 | OOM | 3.22 ± 0.04 | 2.95 ± 0.13 |
| | (2, 1, 0) | 3.19 ± 0.08 | OOM | 2.79 ± 0.11 | 2.58 ± 0.09 |
| | (4, 2, 0) | 3.25 ± 0.09 | OOM | 2.64 ± 0.02 | 3.04 ± 0.05 |
| | (5, 3, 0) | 3.36 ± 0.13 | OOM | 2.55 ± 0.09 | 3.10 ± 0.15 |
| SP | (∞, 6, 4) | 4.00 ± 0.20 | OOM | 2.09 ± 0.17 | 3.89 ± 0.23 |
| | (∞, 4, 3) | 5.53 ± 0.92 | OOM | 5.18 ± 0.51 | 5.53 ± 0.94 |
| | (4, 6, ∞) | 14.41 ± 0.67 | OOM | 6.46 ± 0.84 | 13.63 ± 0.97 |
| | (3, 4, ∞) | 2.93 ± 0.15 | OOM | 2.48 ± 0.06 | 2.51 ± 0.14 |
| PA | (0, 50, 100) | 3.66 ± 0.38 | OOM | 4.34 ± 0.06 | 3.66 ± 0.38 |
| | (0, 100, 200) | 3.19 ± 0.22 | OOM | 4.23 ± 0.03 | 3.19 ± 0.22 |
| | (0, 150, 250) | 2.88 ± 0.06 | OOM | 3.78 ± 0.07 | 2.88 ± 0.06 |
| | (100, 50, 0) | 2.05 ± 0.05 | OOM | 1.43 ± 0.05 | 2.05 ± 0.04 |
| | (200, 100, 0) | 1.67 ± 0.03 | OOM | 1.26 ± 0.02 | 1.65 ± 0.05 |
| | (250, 150, 0) | 1.66 ± 0.02 | OOM | 1.30 ± 0.02 | 1.64 ± 0.03 |

# H    TRADITIONAL OOD GENERALIZATIOM METHOD RESULTS

Table 19: MRR Results for traditional generalization methods applied to GCN and BUDDY on the **Forward** splits for **ogbl-collab**. Generalization methods that significantly improve performance are **bolded**.

| Models | CN Splits | | | SP Splits | | PA Splits | | |
|---|---|---|---|---|---|---|---|---|
| | (0, 1, 2) | (0, 2, 4) | (0, 3, 5) | ($\infty$, 6, 4) | ($\infty$, 4, 3) | (0, 50, 100) | (0, 100, 200) | (0, 150, 250) |
| BUDDY | 17.48 ± 1.19 | 15.47 ± 0.57 | 16.60 ± 0.89 | 16.20 ± 1.40 | 16.42 ± 2.30 | 21.27 ± 0.74 | 14.04 ± 0.76 | 13.06 ± 0.53 |
| +IRM | 1.94 ± 0.16 | 13.42 ± 2.15 | 14.18 ± 0.97 | 4.92 ± 2.55 | 5.05 ± 0.91 | 18.17 ± 3.87 | 9.40 ± 2.22 | 9.40 ± 1.92 |
| +VREx | 14.76 ± 0.58 | **29.81 ± 0.06** | 16.32 ± 0.51 | 11.36 ± 0.92 | **17.56 ± 0.71** | **22.36 ± 0.48** | 14.30 ± 0.26 | 13.30 ± 0.27 |
| GCN | 12.92 ± 0.31 | 15.20 ± 0.16 | 17.54 ± 0.19 | 10.29 ± 0.52 | 12.94 ± 0.59 | 20.78 ± 0.25 | 14.66 ± 0.20 | 14.03 ± 0.15 |
| +IRM | 9.29 ± 0.25 | 12.15 ± 0.38 | 12.00 ± 0.61 | 9.93 ± 0.48 | 10.15 ± 0.72 | 18.77 ± 0.52 | 13.75 ± 0.13 | 12.08 ± 0.74 |
| +VREx | **14.03 ± 0.43** | **17.58 ± 0.39** | 17.56 ± 0.16 | **11.47 ± 0.45** | **13.96 ± 0.76** | **21.13 ± 0.35** | 14.68 ± 0.08 | 13.81 ± 0.07 |
| +GroupDRO | 5.59 ± 0.52 | 5.03 ± 0.34 | 5.68 ± 0.32 | 5.74 ± 0.45 | 6.61 ± 0.65 | 9.33 ± 1.28 | 7.46 ± 0.95 | 6.10 ± 0.77 |
| +DANN | 9.91 ± 0.24 | 10.43 ± 0.39 | 12.50 ± 0.37 | **10.69 ± 0.48** | 11.21 ± 0.85 | **21.51 ± 0.15** | **14.88 ± 0.18** | 12.97 ± 0.15 |
| +Deep CORAL | 9.52 ± 0.36 | 10.08 ± 0.62 | 12.40 ± 0.24 | **10.62 ± 0.51** | 10.99 ± 0.25 | **21.50 ± 0.11** | 14.65 ± 0.22 | 12.80 ± 0.24 |

Table 20: MRR Results for traditional generalization methods applied to GCN and BUDDY on the **Backward** splits for **ogbl-collab**. Generalization methods that significantly improve performance are **bolded**.

| Models | CN Splits | | | SP Splits | | PA Splits | | |
|---|---|---|---|---|---|---|---|---|
| | (2, 1, 0) | (4, 2, 0) | (5, 3, 0) | (4, 6, $\infty$) | (3, 4, $\infty$) | (100, 50, 0) | (200, 100, 0) | (250, 150, 0) |
| BUDDY | 3.70 ± 0.13 | 3.55 ± 0.09 | 5.40 ± 0.04 | 6.73 ± 0.32 | 3.71 ± 0.39 | 24.95 ± 0.92 | 15.52 ± 0.73 | 13.36 ± 0.83 |
| +IRM | 1.81 ± 0.29 | 2.63 ± 0.16 | 3.82 ± 0.17 | 4.93 ± 3.73 | 1.72 ± 0.21 | 16.99 ± 3.53 | 15.76 ± 3.08 | 12.29 ± 4.12 |
| +VREx | 2.65 ± 0.08 | 3.00 ± 0.11 | 4.99 ± 0.13 | 5.10 ± 1.01 | **18.12 ± 1.41** | 22.51 ± 0.76 | 16.01 ± 1.10 | **17.01 ± 0.65** |
| GCN | 5.78 ± 0.19 | 5.91 ± 0.05 | 7.07 ± 0.11 | 8.69 ± 0.47 | 6.47 ± 0.37 | 21.38 ± 0.27 | 13.36 ± 0.24 | 11.95 ± 0.35 |
| +IRM | 4.54 ± 0.54 | 4.87 ± 0.20 | 5.18 ± 0.17 | 6.41 ± 0.35 | 4.65 ± 0.96 | 15.15 ± 0.69 | 10.01 ± 0.57 | 8.63 ± 0.57 |
| +VREx | 5.84 ± 0.31 | **6.04 ± 0.08** | **7.22 ± 0.14** | 8.55 ± 0.42 | 6.49 ± 0.19 | 21.30 ± 0.31 | 13.28 ± 0.28 | 11.94 ± 0.26 |
| +GroupDRO | 2.49 ± 0.07 | 2.93 ± 0.18 | 3.04 ± 0.08 | 4.04 ± 1.24 | 2.93 ± 1.06 | 2.63 ± 0.73 | 2.60 ± 0.32 | 2.35 ± 0.26 |
| +DANN | 6.01 ± 0.15 | 5.95 ± 0.14 | 6.49 ± 0.07 | 14.68 ± 0.38 | 9.38 ± 0.20 | 8.62 ± 0.24 | 8.73 ± 0.54 | 6.67 ± 0.19 |
| +Deep CORAL | 5.98 ± 0.12 | 6.01 ± 0.15 | 6.48 ± 0.16 | 14.73 ± 0.44 | 9.42 ± 0.09 | 8.42 ± 0.45 | 8.57 ± 0.37 | 6.54 ± 0.17 |

# I    EARTH MOVER'S DISTANCE (EMD) RESULTS

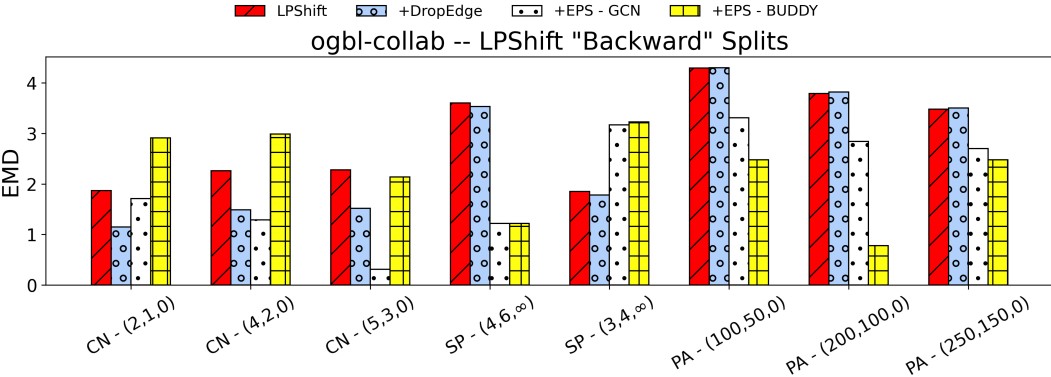

Figure 7: The EMD values calculated between the heuristic scores of training and testing samples on the "Backward" LPShift splits before and after applying structural generalization methods. *Note:* The tested heuristics correspond to their labelled LPShift splits, so as to simulate the dataset splitting.

Table 21: EMD calculations for ogbl-collab on the forward and backward splits. Scores with a distance multiple-times different than the baseline are in **bold**

| Heuristic | Split | Baseline | DropEdge | EPS - GCN | EPS - BUDDY |
|-----------|-------|----------|----------|-----------|-------------|
| CN | (0, 1, 2) | 1.31 | 1.31 | **6.62** | **3.6** |
| | (0, 2, 4) | 1.6 | 1.6 | **4.91** | 2.52 |
| | (0, 3, 5) | 1.45 | 1.65 | **3.82** | 2.22 |
| | (2, 1, 0) | 1.87 | 1.15 | 1.71 | 2.91 |
| | (4, 2, 0) | 2.26 | 1.49 | 1.29 | 2.99 |
| | (5, 3, 0) | 2.28 | 1.52 | **0.314** | 2.14 |
| SP | $(\infty, 6, 4)$ | 5.93 | 5.94 | **0.012** | **0.012** |
| | $(\infty, 4, 3)$ | 5.35 | 5.38 | **0.002** | **0.003** |
| | $(4, 6, \infty)$ | 3.6 | 3.53 | **1.22** | **1.22** |
| | $(3, 4, \infty)$ | 1.85 | 1.78 | 3.17 | 3.23 |
| PA | (0, 50, 100) | 1.87 | 1.89 | 2.94 | 3.42 |
| | (0, 100, 200) | 2.29 | 2.32 | 3.59 | 2.72 |
| | (0, 150, 250) | 2.34 | 2.36 | 3.73 | 3.08 |
| | (100, 50, 0) | 4.29 | 4.3 | 3.31 | 2.48 |
| | (200, 100, 0) | 3.79 | 3.82 | 2.84 | **0.78** |
| | (250, 150, 0) | 3.48 | 3.5 | 2.7 | 2.48 |

Table 22: EMD calculations for ogbl-ppa on the forward and backward splits.

| Heuristic | Split | Baseline | DropEdge | EPS |
|-----------|-------|----------|----------|-----|
| CN | (0, 1, 2) | 2.82 | 2.82 | >24hrs |
| | (0, 2, 4) | 3.13 | 3.13 | >24hrs |
| | (0, 3, 5) | 3.05 | 3.19 | >24hrs |
| | (2, 1, 0) | 3.1 | 2.36 | >24hrs |
| | (4, 2, 0) | 3.3 | 2.55 | >24hrs |
| | (5, 3, 0) | 3.19 | 2.44 | >24hrs |
| SP | $(\infty, 6, 4)$ | 5.81 | 5.84 | >24hrs |
| | $(\infty, 4, 3)$ | 1.36 | 1.4 | >24hrs |
| | $(4, 6, \infty)$ | 2.14 | 2.14 | >24hrs |
| | $(3, 4, \infty)$ | 0.72 | 0.72 | >24hrs |
| PA | (0, 5k, 10k) | 2.55 | 2.55 | >24hrs |
| | (0, 10k, 20k) | 2.76 | 2.76 | >24hrs |
| | (0, 15k, 25k) | 2.78 | 2.78 | >24hrs |
| | (10k, 5k, 0) | 2.96 | 2.96 | >24hrs |
| | (20k, 10k, 0) | 2.68 | 2.68 | >24hrs |
| | (25k, 15k, 0) | 2.48 | 2.48 | >24hrs |

## J  LPShift's effect on HeaRT performance

Table 23: The percent change in Hits@20 for each LPShift split type and respective split direction versus the original HeaRT setting (Li et al., 2024) on ogbl-collab. *Note:* LPFormer was untested within the HeaRT paper and not included in this table.

| Method | "Backward" | | | "Forward" | | |
|---|---|---|---|---|---|---|
| | CN | SP | PA | CN | SP | PA |
| RA | -36.65% | -100% | +300.26% | +327.08% | +330.65% | +281.68% |
| GCN | +9.68% | +33.8% | +267.32% | +219.37% | +186.0% | +233.36% |
| BUDDY | -37.32% | -17.5% | +273.39% | +256.09% | +235.57% | +213.86% |
| NCNC | +42.39% | -86.8% | +291.0% | +216.11% | +255.81% | +162.49% |
| Neo-GNN | -24.82% | -34.67% | +65.69% | +196.32% | +15.81% | +201.32% |
| SEAL | -43.58% | -98.35% | +353.37% | +210.23% | -75.11% | +250.84% |

Table 24: The percent change in Hits@20 for each LPShift split type and respective split direction versus the original HeaRT setting (Li et al., 2024) on ogbl-ppa. *Note:* LPFormer was untested within the HeaRT paper and not included in this table.

| Method | "Backward" | | | "Forward" | | |
|---|---|---|---|---|---|---|
| | CN | SP | PA | CN | SP | PA |
| RA | -97.7% | -100% | -66.3% | -75.85% | +7.45% | -82.6% |
| GCN | -83.18% | -64.76% | -97% | -59.5% | -61.3% | -72.5% |
| BUDDY | -91.0% | -64.7% | -86.1% | -75.4% | -67.2% | -84.48% |
| NCNC | -80.0% | -79.7% | -71.0% | -72.2% | -65.5% | -86.8% |
| Neo-GNN | -99.8% | -85.5% | -96.1% | -69.25% | -72.65% | -71.3% |
| SEAL | -98.5% | -96.6% | -81.9% | -71.12% | -78.0% | -85.0% |

## K  Additional Training Details

This section provides relevant details about training and reproducing results not mentioned in Section 4.1:

Please consult the project README for building the project, loading data, and re-creating results. Tuned model hyperparameters are further detailed within their respective run scripts.

Table 25: Fixed Model Hyperparameters by tested LPShift dataset.

| Model | Dataset | Model Layers | Predictor Layers | Hidden Channels |
|---|---|---|---|---|
| GCN | ogbl-collab | 3 | 3 | 128 |
| | ogbl-ppa | 3 | 3 | 128 |
| BUDDY | ogbl-collab | 3 | 3 | 256 |
| | ogbl-ppa | 3 | 3 | 256 |
| NCNC | ogbl-collab | 3 | 3 | 256 |
| | ogbl-ppa | 3 | 3 | 256 |
| NeoGNN | ogbl-collab | 3 | 3 | 256 |
| | ogbl-ppa | 3 | 3 | 256 |
| LPFormer | ogbl-collab | 3 | 3 | 128 |
| | ogbl-ppa | 3 | 3 | 128 |
| SEAL | ogbl-collab | 3 | 3 | 256 |
| | ogbl-ppa | 3 | 3 | 256 |

- All experiments were conducted with a single A6000 48GB GPU and 1TB of available system RAM.

- NCNC for all datasets and splits, besides the ogbl-ppa PA splits, considers the 'NCNC2' variant of NCNC with an added depth argument of 2 (Wang et al., 2023a). For the ogbl-ppa PA splits, we apply a depth argument of just 1 in order to ensure that a single seeded run does not exceed 24 hour runtime.

- NeoGNN use 2-hop neighborhoods on the LPShift ogbl-collab datasets and 1-hop on the LPShift ogbl-ppa datasets.

- Initial tuning on batch size fixed learning rate at $1e^{-3}$ and dropout at $0.1$. Model performance and memory complexity was then tested for a single run across a space of $\{8, 16, 32, 64, 128, 256, 512, 1024, 2048, 4096, 8192, 16384, 32768, 65536\}$. This was done with the intent to balance computational time with performance for full experiments.

- At runtime, training samples are constrained to the same number as validation samples to prevent overfitting, especially in scenarios where the splitting strategy produces vastly more training samples than valid samples.

## L   WHY USE LPSHIFT? PERSPECTIVES ON DISTRIBUTION SHIFT.

- **Data Perspective:** Link prediction is a task focused on understanding the dynamics when edges form between nodes; requiring models to effectively understand these dynamics to determine whether a link will form (or not) (Liben-Nowell & Kleinberg, 2003). As such, link prediction is more interested in pairwise dynamics than what is necessary for graph and node classification. Due to this distinction, there is limited overlap between graph/node classification and link prediction. As such, distribution shifts that are relevant in graph and node classification do not mean as much in link prediction. For example, we consider the scaffold shift imposed on molecule datasets within the DrugOOD benchmark (Ji et al., 2022). This special type of structural shift groups molecules with similar subgraphs to induce an out-of-distribution scenario. Given that the scaffold shift in DrugOOD is not associated with the formation of links, then the shift loses it's relevancy when applied to models that learn pairwise dynamics necessary for link prediction.

- **Model Perspective:** Structural heuristics that consider pairwise information are important for SOTA GNN4LP models. For example, BUDDY (Chamberlain et al., 2022) integrates RA directly into it's architecture and NCNC (Wang et al., 2023a) elevates CN into a neural architecture. The integration of these structural heuristics allow the link prediction models to achieve SOTA performance. However, as demonstrated in Tables 23 and 24, GNN4LP's reliance on structural heuristics leads to degraded performance when LPShift induces structural shift within the graph dataset. The performance of models less reliant on structure, such as GCN (Kipf & Welling, 2017), do not degrade as significantly. This indicates that future models must balance between learning on graph structure and understanding shifts in link formation to improve performance under LPShift; especially in scenarios analogous to real-life (i.e. Figure 1).

- **Application Perspective:** There is a practical difference between graph or node classification and link prediction as downstream tasks for graph representation learning (Hu et al., 2020). Even though a graph or node classification model could consider pairwise dynamics; it is limiting for a link prediction model to only consider node labels when predicting if edges will form (Yun et al., 2021; Zhang et al., 2021). Pairwise dynamics are distinct in that they determine how a graph forms on a finer scale, whereas node information constitutes what a graph is made of (Adamic & Adar, 2003; Barabási & Albert, 1999). The distinctness of these pairwise dynamics then necessitates dedicated evaluation to edge structures within the graphs. As such, LPShift serves as a measurable means to define a shift in these pairwise dynamics, allowing evaluation in a controlled setting that is relevant for link prediction.

## M    DATASET LICENSES

The dataset splitting strategy proposed in this paper is built using Pytorch Geometric (PyG). As such, this project's software and the PyG datasets are freely-available under the MIT license.

## N    LIMITATIONS

The proposed dataset splitting strategy is restricted to inducing distribution shifts solely with neighborhood heuristics on static graphs. So, it does not directly consider other types of possible distribution shifts for the link prediction task (i.e. spatio-temporal (Zhang et al., 2022) or size (Zhou et al., 2022b) shift). Additionally, since the neighborhood heuristics compute discrete scores produced from an input graph's structural information and effectively training GNN4LP models requires no leakage with validation/testing, it may be difficult to determine the correct thresholds to extract a meaningful number of samples. For Common Neighbors and Preferential-Attachment, this is especially relevant with smaller training graphs, given that larger and/or denser graphs have inherently more edges. Therefore, larger and denser graphs have inherently more possible Common Neighbors and Preferential-Attachment scores. For Shortest-Path, splitting can be exceptionally difficult for denser graphs, as demonstrated with the tiny split sizes for ogbl-ppa in Table 6.

## O    IMPACT STATEMENT

Our proposed dataset splitting strategy mimics the formatting of PyTorch Geometric datasets. This means that our strategy is simple to implement, enabling future work involved with understanding this type of structural shift for link prediction and promoting beginner-friendly practices for artificial intelligence research. Additionally, since the structural shift we propose in this article affects real-life systems, which integrate link prediction models, this research can provide a foundation for the improvement of relevant technologies; which holds positive ramifications for society and future research. No apparent risk is related to the contribution of this work.

