# OpenReview forum: "Towards Understanding Link Predictor Generalizability Under Distribution Shifts"
_ICLR.cc/2025/Conference — Submitted to ICLR 2025_

### Official Review · Reviewer_RYKU · 2024-10-16

**Soundness:** 3
**Presentation:** 3
**Contribution:** 2
**Rating:** 5
**Confidence:** 4

**Summary:**

This paper investigates the distribution shift issues on LP tasks on the graph, where the links in training, validation, and testing sets may be drawn from different distributions. The authors introduce a graph edge splitting method LPShift, which can artificially create a split with distribution shift. Three traditional LP methods are selected as the metrics to perform the LPShift: CN, Shortest-Path, and Preferential Attachment. LPShift is applied to OGB datasets: Collab and PPA to generate a set of new splits. By evaluating the LP methods' performance on these new splits, the authors conclude that current LP methods are sensitive to the distribution shift problem, and advocate for further exploration on this issue.

**Strengths:**

[S1] The paper is easy to read. It has a clear demonstration of the problem setup, methods, experiments, and conclusions.

[S2] Different from previous distribution shift studies, this paper identifies the issue of LP tasks, which is novel.

[S3] The experiment shows that LPShift can effectively cause a distribution shift problem for current SOTA LP methods.

**Weaknesses:**

[W1] My major concern is whether the study has any practical merit. It is unknown if the distribution shift problem for LP tasks (widely) exists in the existing benchmarks. Instead, in this paper, the problem is artificially created and identified by manually creating such a dataset. There is Collab that has such an issue, but it is already reported in [1], with a very similar plot. It would be more convincing if the authors could show and identify more existing datasets that suffer from such an issue.

[W2] The experimental results are confusing. Table 1 uses Hits@20 as the metrics to rank models, but Table 7/8 uses MRR for actual performance. If both metrics have to be applied here, then they should both have the same set of tables (ranking table as of table 1, actual performance table as of table 7) for a clear demonstration.

[W3] The relative comparison among LP methods in Table 1 can be misleading. The authors attempt to conclude that simpler methods with Collab can perform better because GNN4LP struggles to generalize due to the structural shift induced by LPShift. However, the LPShift also uses HeaRT[2] to generate negative samples. And under the HeaRT setting alone, simpler methods like RA can already perform better than GNN4LP. This leads to confusion about whether it is the HeaRT or LPShift that causes GNN4LP to fail to generalize.

Minor issues:

* Authors use "inverse split" and "backward split" interchangeably, which causes confusion.


[1] Neural Common Neighbor with Completion for Link Prediction.

[2] Evaluating Graph Neural Networks for Link Prediction: Current Pitfalls and New Benchmarking.

**Questions:**

[Q1] Link-MoE [3] uses a set of link predictors as experts to perform LP tasks and achieves SOTA performance. Given that the authors show that different methods have different capabilities to handle distribution shift, will a mixture of experts model help with the issue by integrating strengths of different models together?

[Q2] Since CN, SP, and PA are selected as the metrics for LPShift, it would be interesting to include SP and PA and see how they work under the shifting, similar to RA.

[Q3] Follow up on W1, if Collab already shows a distribution shift issue, is it necessary to further manually corrupt the dataset?

[3] Mixture of Link Predictors.

---

> ### Author Response · Authors · 2024-11-21
> **Response to Reviewer RYKU (1/3)**
>
> > My major concern is whether the study has any practical merit. It is unknown if the distribution shift problem for LP tasks (widely) exists in the existing benchmarks. Instead, in this paper, the problem is artificially created and identified by manually creating such a dataset. There is Collab that has such an issue, but it is already reported in [1], with a very similar plot. It would be more convincing if the authors could show and identify more existing datasets that suffer from such an issue.
>
> Within link prediction, the distribution shift problem remains largely unknown. We note that this is the first study to consider benchmarking performance for distribution shifts in link prediction. So, the foundational knowledge provided by our study enhances it's practical merit. Since ogbl-collab has a known distribution shift, while little remains known about **how distribution shift functions and then affects link prediction performance**; then it becomes increasingly important to understand how different and quantifiable distribution shifts affect performance in link prediction.
>
> All other graph out-of-distribution benchmarks [1-4] manually induce distribution shifts in pre-existing datasets to generate datasets that suffer from distribution shift. As such, LPShift functions as a means to induce a distribution shift in order to better understand how distribution shift affects link prediction; a currently under-explored area of research in graph representation learning.
>
> We also note, that the plots included in Figure 5 of our original submission makes an important distinction about the extent of the distribution shift for Common Neighbors **between splits**, whereas Figure 1 in the NCNC paper [5] is interested in the **overall density** of Common Neighbors. As such, our Figure 5 becomes more relevant for a study on distribution shift between different splits.
>
> [1] Gui, Shurui, et al. "GOOD: a graph out-of-distribution benchmark." Proceedings of the 36th International Conference on Neural Information Processing Systems. 2022.
>
> [2] Ji, Yuanfeng, et al. "DrugOOD: Out-of-Distribution (OOD) Dataset Curator and Benchmark for AI-aided Drug Discovery--A Focus on Affinity Prediction Problems with Noise Annotations." arXiv preprint arXiv:2201.09637 (2022).
>
> [3] Koh, Pang Wei, et al. "Wilds: A benchmark of in-the-wild distribution shifts." International conference on machine learning. PMLR, 2021.
>
> [4] Hu, Weihua, et al. "Open graph benchmark: Datasets for machine learning on graphs." Advances in neural information processing systems 33 (2020): 22118-22133.
>
> [5] Wang, Xiyuan, Haotong Yang, and Muhan Zhang. "Neural Common Neighbor with Completion for Link Prediction." The Twelfth International Conference on Learning Representations.
>
> > The experimental results are confusing. Table 1 uses Hits@20 as the metrics to rank models, but Table 7/8 uses MRR for actual performance. If both metrics have to be applied here, then they should both have the same set of tables (ranking table as of table 1, actual performance table as of table 7) for a clear demonstration.
>
> We agree this is important for clarity and have amended the additional results tables showing Hits@20 scores for ogbl-collab and ogbl-ppa to the revised Appendix Section F of our paper. The Tables are numbered: 8, 10, 12, and 14 respectively

---

> ### Author Response · Authors · 2024-11-21
> **Response to Reviewer RYKU (2/3)**
>
> > The relative comparison among LP methods in Table 1 can be misleading. The authors attempt to conclude that simpler methods with Collab can perform better because GNN4LP struggles to generalize due to the structural shift induced by LPShift. However, the LPShift also uses HeaRT[2] to generate negative samples. And under the HeaRT setting alone, simpler methods like RA can already perform better than GNN4LP. This leads to confusion about whether it is the HeaRT or LPShift that causes GNN4LP to fail to generalize.
>
> The traditional link prediction setting relies on random negative samples, such samples are known to be trivial for link prediction models to classify [1]. Given that LPShift is concerned with structural shift within a given dataset split, and therefore amongst the positive samples pulled from that same split, it becomes counter-intuitive to forego structure for the negative samples. Following the intuition provided by HeaRT, random samples extracted for the "Forward" CN split then allow the model to generalize more easily. Following the intuition demosntrate in our original Figure 1, the "Backward" CN split induces a more difficult scenario for the LP model to generalize to as the positive and negative samples are often hard to distinguish (e.g., all have 0 common neighbors).
>
> Our intention is not to be misleading. So, in the Table 1 (shown below), we include MRR results for the ogbl-collab datasets affected by the 'Forward' and 'Backward' CN LPShift with random samples pulled from the original ogbl-collab dataset in order to verify our intuition.
>
> | Collab | CN - 1,2 | CN - 2,4 | CN - 3,5 | CN - 2,1 | CN - 4,2 | CN - 5,3 |
> |:--------:|:-------------:|:------------:|:-------------:|:--------------------:|:------------:|:-------------:|
> | GCN | 70.48 ± 0.76 | 80.36 ± 0.14 | 85.43 ± 0.23 | 27.66 ± 0.30 | 36.44 ± 0.30 | 46.87 ± 0.41 |
> | BUDDY | 88.72 ± 1.77 | 96.23 ± 0.45 | 97.42 ± 0.19 | 22.97 ± 0.67 | 38.53 ± 1.08 | 61.75 ± 0.84 |
> | RA | 95.12 | 98.38 | 98.69 | 0.79 | 25.44 | 63.94 |
> | AA | 95.36 | 98.61 | 98.91 | 0.79	| 25.25	| 63.72 |
> | CN | 94.01 | 98.05 | 98.49 | 0.79	| 24.27	| 60.76 |
>
> Tables 1 (shown above): MRR Results for LPShift's CN Backward and Forward split on ogbl-collab.
>
> We make three observations on Table 1:
> 1. The performance boosts significantly for the both splits versus the LPShift baselines with HeaRT sampling. We note that the forward split follows common LP assumptions; that LP models will generalize much better to testing data with more structural information. We also note that the performance boost for the backward split is not as significant as the performance increase for the forward split.
> 2. Common Neighbors, Adamic-Adar, and Resource-Allocation continue to outperform GCN and BUDDY on all of the 'Foward' splits and the CN - 5,3 split.
> 3. GCN outperforms BUDDY in the CN - 2,1 split, which follows along with the intuition presented in our article's Figure 1. However, BUDDY begins to outperform GCN once more structural information is made available.
>
> Given observations (1), it is apparent that HeaRT leads to a decrease in performance for the initial LPShift baseline tests. However, observations (2) and (3) indicate the powerful impact that a change in structural information has on the performance of LP models. In particular, every tested LP model suffers under the CN - 2,1 split.
>
> [1] Li, Juanhui, et al. "Evaluating graph neural networks for link prediction: Current pitfalls and new benchmarking." Advances in Neural Information Processing Systems 36 (2024).
>
> > Authors use "inverse split" and "backward split" interchangeably, which causes confusion.
>
> We apologize for any confusion caused by a lack of clarity. Our revised paper now clarifies an 'inverse' split as a 'backward' split. We include an example of this below:
>
> *Lines 436-437:* However, Figure 4 indicates that DropEdge only
> has a significant effect on EMD when handling the “backward” ogbl-collab CN split.
>
> >  Link-MoE [3] uses a set of link predictors as experts to perform LP tasks and achieves SOTA performance. Given that the authors show that different methods have different capabilities to handle distribution shift, will a mixture of experts model help with the issue by integrating strengths of different models together?
>
> This is a great suggestion, given that different GNN4LP models integrate structural features into their architectures, like RA into BUDDY or NCNC acts as a neural-version of Common Neighbors, then those models should naturally generalize better in scenarios where a given structual heuristic is increasingly present in the training data. Therefore, leveraging the strengths of different models as experts could enhance performance under distribution-shifted scenarios.
>
> We have contacted the authors, who just recently made the Link-MoE code publicly-available. We will be running experiments in the meantime and hope to report back soon.

---

> ### Author Response · Authors · 2024-11-21
> **Response to Reviewer RYKU (3/3)**
>
> >  Since CN, SP, and PA are selected as the metrics for LPShift, it would be interesting to include SP and PA and see how they work under the shifting, similar to RA.
>
> Thank you for the suggestion, we include available results on PA and SP in Tables 2,3,4, and 5 as shown below. Given the expensive runtime of SP and the large size of the ogbl-collab and ogbl-ppa datasets, SP is still running on the remaining splits.
>
> With the current results, we determine that even though PA and SP are relevant for link prediction, the methods fail to generalize under LPShift scenarios. This is likely due to the high-importance of local information for predicting links within ogbl-collab and ogbl-ppa.
>
> | Collab | CN - 1,2 | CN - 2,4 | CN - 3,5 | SP - 17,26 | SP - 26,36 | PA - 50,100 | PA - 100,200 | PA - 150,250 |
> |:--------:|:-------------:|:------------:|:-------------:|:--------------------:|:------------:|:-------------:|:------------:|:------------:|
> | PA | 3.63 | 3.41 | 3.39 | 3.56 | 3.37 | 2.34 | 2.82 | 2.83 |
> | SP | 4.73 | 20.25 | 2.7 | 5.74 | 5.89 | 7.2 | 3.61 | 3.14 |
>
> Table 2 (shown above): MRR results for PA and SP on the Forward LPShift split of ogbl-collab.
>
> | PPA | CN - 1,2 | CN - 2,4 | CN - 3,5 | SP - 17,26 | SP - 26,36 | PA - 5k,10k | PA - 10k,20k | PA - 15k,25k |
> |:--------:|:-------------:|:------------:|:-------------:|:--------------------:|:------------:|:-------------:|:------------:|:------------:|
> | PA |1.45 | 1.43 | 1.3 | 1.8 | 1.08 | 1.82 | 2.14 | 2.05 |
> | SP | 1.15 | >3 days | >4 days |>4 days | >4 days | >4 days | >4 days | >4 days |
>
> Table 3 (shown above): MRR results for PA and SP on the Forward LPShift split of ogbl-ppa.
>
>
> | Collab | CN - 2,1 | CN - 4,2 | CN - 5,3 | SP - 26,17 | SP - 36,26 | PA - 100,50 | PA - 200,100 | PA - 250,150 |
> |:--------:|:-------------:|:------------:|:-------------:|:--------------------:|:------------:|:-------------:|:------------:|:------------:|
> | PA |1.63 | 1.84 | 2.01 | 1 | 1.29 | 0.66 | 0.74 | 0.81 |
> | SP | 1.08 | 1.98 | 3.02 | 0.68 | 0.98 | >1 day | >1 day | >1 day |
>
> Table 4 (shown above): MRR results for PA and SP on the Backward LPShift split of ogbl-collab.
>
> | PPA | CN - 2,1 | CN - 4,2 | CN - 5,3 | SP - 26,17 | SP - 36,26 | PA - 10k,5k | PA - 20k,10k | PA - 25k,15k |
> |:--------:|:-------------:|:------------:|:-------------:|:--------------------:|:------------:|:-------------:|:------------:|:------------:|
> | PA |0.82 | 0.86 | 0.89 | 0.46 | 0.6 | 0.65 | 0.73 | 0.79 |
> | SP | >4 days | >4 days | >4 days | >4 days | >4 days | >4 days | >4 days | >4 days |
>
> Table 5 (shown above): MRR results for PA and SP on the Backward LPShift split of ogbl-ppa.
>
> > Follow up on W1, if Collab already shows a distribution shift issue, is it necessary to further manually corrupt the dataset?
>
> As shown in Figure 6 of our original submission, the inherent Common Neighbor shift in ogbl-collab indicates that local structural information leads to decreased LP performance. Since LPShift allows manual and fine-grained control over the extent and type of structure used to split a graph dataset. LPShift then makes it possible to parse the **specific structural characteristic** affecting performance of LP models on ogbl-collab. For example, reduced LP performance on the CN - 2,1 split versus the CN - 5,3 split empirically demonstrates the importance of increased local structural information for enhancing performance on ogbl-collab.
>
> On a similar note, given the flexibilty of LPShift as a splitting strategy, it is possible to integrate other structural perspectives, like **global structural information** with SP or high-level concepts such as **preferential-attachment** (PA). If LPShift is used in this manner then it serves as a way to provide a more nuanced perspective for structural analysis on an affected dataset.

---

> ### Author Response · Authors · 2024-11-23
> **Gentle Reminder about Rebuttal**
>
> Dear Reviewer RYKU,
>
> Thank you for taking the time to review our work! Your input greatly improves the quality of our study.
>
> We have posted our rebuttal and hope that it addresses all of your concerns, especially about the practical merit of LPShift. Please consider it in your evaluation and inform us of any additional concerns.
>
> Kind Regards,
>
> Authors

---

> > ### Comment · Reviewer_RYKU · 2024-11-24
> >
> > Thanks for the author's response to my questions. However, my concerns remain:
> >
> > 1. For the first subplot in Figure 5, I still cannot understand how it differs from the one in the NCNC paper. In the NCNC paper, they clearly state that they find a CN distribution shift between training and testing on the Collab dataset. Figure 5 in this paper almost repeats such a finding, adding one extra CN distribution of the validation set. The finding is too subtle to differentiate itself, which hurts the new knowledge brought by this study.
> >
> > 2. I appreciate the authors providing Table 8/10/12/14. However, what makes me confused is in the "Observation 1" in Sec 4.2. The authors claim that RA is robust to LPShift in Table 1, measured by ranking Hits@20. Then it further said RA is best-performing in Table 7, which is measured by MRR. These two different measures make it difficult to understand the context.
> >
> > 3. There should be some typo in Table 12, where NCNC and RA always have the same results. Based on Table 12/14, the ranking of RA in Table 1 with backward PA split is calculated by mistake.
> >
> > 4. I appreciate the experiments conducted by authors to evaluate the LPshift **without** HeaRT. As the results reveal, LPShift can hardly impede the performance of GNN4LPs, only significantly impacting CN - 1,2. In other words, the distribution shift problem identified in this study might be a minor issue for LP tasks. In a comparison, the evaluation setting studied by HeaRT alone can introduce much significant issue for LP methods.
> >
> > Here are my suggestions:
> >
> > a. If HeaRT still needs to be included in LPShift, a ranking table like Table 1 should be made for HeaRT alone. Then a comparison between the ranking of models under HeaRT vs under HeaRT+LPShift should tell how effective LPShift is to introduce an issue to LP models.
> >
> > b. Authors can consider completely getting rid of HeaRT. Then a ranking table like Table 1 alone can tell if it causes problems for GNN4LPs.
> >
> > After all, I am not convinced that the distribution shift introduced by LPShift is a significant problem for LP.
> >
> > 5. Figure 3 still uses "inverse split".

---

> ### Author Response · Authors · 2024-11-25
> **Response #2 to Reviewer RYKU (1/N)**
>
> > For the first subplot in Figure 5, I still cannot understand how it differs from the one in the NCNC paper. In the NCNC paper, they clearly state that they find a CN distribution shift between training and testing on the Collab dataset. Figure 5 in this paper almost repeats such a finding, adding one extra CN distribution of the validation set. The finding is too subtle to differentiate itself, which hurts the new knowledge brought by this study.
>
> The intent of Figure 5 is to demonstrate that LPShift is capable of inducing a meaningful distribution shift. So, we include the first subplot, with the added validation split, to demonstrate the realistic distribution shift for the original ogbl-collab dataset. We then apply the "Backward" LPShift, as indicated from the changes in the distribution observed between the second to third subplot. We apply the 2 KS-test to determine whether the training/validation/testing distributions can be sampled from one another. A p-value of 0 indicating with statistical significance that the distributions are distinct, while a p-value of 1 indicating that the distributions cannot be distinguished from one another. We revise our paper's Section 4.4 to better clarify the contribution of LPShift along with the inspiration from NCNC, shown below in blockquotes:
>
> >> The first subplot in Figure 5 extends the reasoning introduced with NCNC Wang et al. (2023a). As such, this subplot indicates there is a natural shift for CNs within the original ogbl-collab dataset Hu et al. (2020). The p-value of 0, measured across both split permutations, indicates that the training distribution of CNs is shifted from the validation and testing distributions. The second subplot depicts a randomly-split HK graph Holme & Kim (2002), where CN distributions for each split match one another, further indicated by the p-values of 1. The third subplot depicts the HK graph from the second subplot split with LPShift’s CN - 5,3 strategy, resulting in a distinct shift between all dataset splits, as confirmed by the 0 p-values. As such, LPShift induces structural shift that is as measurably dissimilar as the structural shift present in the original ogbl-collab dataset, even when the initial dataset splits are measurably identical. Additionally, the CN - 5,3 split causes the shape of the HK graph’s CN distributions to become more similar to the shift observed within the original ogbl-collab dataset, indicating that the ”Backward” LPShift strategy can function like a real-world distribution shift.
>
> > I appreciate the authors providing Table 8/10/12/14. However, what makes me confused is in the "Observation 1" in Sec 4.2. The authors claim that RA is robust to LPShift in Table 1, measured by ranking Hits@20. Then it further said RA is best-performing in Table 7, which is measured by MRR. These two different measures make it difficult to understand the context.
>
> We have edited Observation 1 in Section 4.2, and made sure to highlight the best- and underline the second-best performing models within Tables 8/10/12/14. The updates indicate that RA is the best-performing model in Table 7 and Table 8, clarifying it's ability to rank and predict information under LPShift with both MRR and Hits@20. We provide the updated Section 4.2 in block quotes below:
>
> >> As shown in Table 1, RA and GCN consistently out-perform or remain competitive with GNN4LP models, with more detailed results included in Appendix G. In Table 7 and 8, RA is overwhelmingly the best-performing, achieving scores at least 6 MRR and 13 Hits@20 higher than the next best model. However the results for the ogbl-ppa forward split, as shown in Table 9, indicate LPFormer as the best-performing model on the PA split and NeoGNN on the CN - 3,5 split, albeit with a much lower average score than those demonstrated within the ogbl-collab forward split.
>
> > There should be some typo in Table 12, where NCNC and RA always have the same results. Based on Table 12/14, the ranking of RA in Table 1 with backward PA split is calculated by mistake.
>
> Thank you for catching this, this was indeed a typo. We have updated Table 12 with the correct scores and double-checked the ranking. Therefore, no changes were necessary within Table 1 to accurately reflect rankings as determined by the results for Tables 12 and 14.

---

> ### Author Response · Authors · 2024-11-25
> **Response #2 to Reviewer RYKU (2/N)**
>
> > a. If HeaRT still needs to be included in LPShift, a ranking table like Table 1 should be made for HeaRT alone. Then a comparison between the ranking of models under HeaRT vs under HeaRT+LPShift should tell how effective LPShift is to introduce an issue to LP models.
>
> Thank you for your suggestions, we want to clarify one consideration about HeaRT and it's relation to LPShift.
>
> * LPShift works as a natural extension of HeaRT; HeaRT was constructed as a sampling technique which solves two key issues posed by random negative sampling: 1) **Non-personalized samples**, where there is no change in the negative samples tested for each positive sample. 2) **Easy Negative Samples**, where negative samples have no relation to the positive sample; making it *trivial* for the link prediction model to distinguish if a link forms within a given sample. From which, HeaRT applies pairwise structural heuristics and then targets relevant edges within positive samples to create better negative samples. Given that LPShfit induces a unique type of structural covariate shift which is also focused on pairwise relations, we consider HeaRT an integral component of the LPShift strategy. The integration of HeaRT into LPShift becomes especially prevalent due to the LPShift+Random results, indicating that the models could be overfitting to structure due to the supervised nature of link prediction and the structural shift exacerbating the triviality of the random negative samples.
>
> However, we agree that this sort of distinction is important to improve understanding of LPShift's effects. We provide Table 6 and 7 as requested in suggestion a.) to allow for comparison of the results across LPShift datasets and their splits to the HeaRT results. Additionally, we add the "Backward" and "Forward" LPShift columns to indicate the average change in Hits@20 between LPShift+HeaRT versus HeaRT for each split type and direction of a respective dataset.
>
> We make these observations with included insights about the effect that LPShift has when combined with HeaRT:
>
> * The rankings in our revised paper's Table 1 do not perfectly align with the HeaRT rankings. We attribute this to RA's reliance on structural information; meaning it drastically loses performance on "Backward" splits.
> * ogbl-ppa has signficantly reduced performance, where the performance declines at least 59.5% but can rise up to 100% loss of performance. The only exception being the "Forward" SP split. We attribute this significant and consistent decrease in performance to the size of ogbl-ppa and it's relative abundance of positive and negative samples.
> * The "Backward" CN and SP split for ogbl-collab typically result in a performance decrease ranging from 17.5% up to 100%. The only exceptions are GCN and NCNC. We attribute GCN's success since it does not rely on structure within it's architecture. We attribute NCNC's initial success on the "Backward" CN split to it's ability to effectively capture local information. However, NCNC loses 86.8% performance on the subsequent "Backward" SP split.
> * The "Backward" PA and all "Forward" splits for ogbl-collab overwhelmingly lead to an increased performance for all models, typically resulting in 2 to 3 times better performance. This serves as measured proof for the prevailing assumption in link prediction; that LP models can generalize better to scenarios with more structural information than what was available in training. The only exception is for SEAL on the "Forward" SP split. We attribute this to the lack of neighborhood overlap, which inhibits the expressive power of SEAL's labelling trick.
>
> Given these observations, it is clear that LPShift has a significant impact on LP performance, especially in the "Backward" the direction. In particular, the effect on performance for the ogbl-ppa dataset indicates how vulnerable GNN4LP models are to structural shift in larger datasets.
>
> ||ogbl-collab|"Backward" - CN  | SP |PA | "Forward" - CN  |SP |PA |
> |:-------:|:---------:|:------:|:------:|:------:|:-------:|:---------:|:------:|
> |  RA     |    1    |-36.65% |-100% |+300.26 |+327.08% |+330.65% |+281.68% |
> |  GCN   |      3    |+9.68% |+33.8% |+267.32% |+219.37% |+186.0% |+233.36% |
> |  BUDDY   |     2   |-37.32% |-17.5% |+273.39% |+256.09% |+235.57% |+213.86% |
> |  NCNC   |     6    |+42.39% |-86.8% |+291.0% |+216.11% |+255.81% |+162.49% |
> |  Neo-GNN   |   5     |-24.82% |-34.67% |+65.69% |+196.32% |+15.81% |+201.32% |
> |  SEAL   |     4     |-43.58% |-98.35% |+353.37% |+210.23% |-75.11% |+250.84% |
>
> **Table 6 (shown above):** The rankings for all models with just HeaRT on ogbl-collab, as determined by the Hits@20 Results from [Table 5 in the HeaRT paper](https://arxiv.org/pdf/2306.10453). The final six columns represent the change in Hits@20 for each LPShift split type and respective split direction versus the original HeaRT *Note:* LPFormer was untested within the HeaRT paper and not included in these rankings.

---

> ### Author Response · Authors · 2024-11-25
> **Response #2 to Reviewer RYKU (3/N)**
>
> ||ogbl-ppa |"Backward" - CN  | SP |PA | "Forward" - CN  |SP |PA |
> |:-------:|:---------:|:------:|:------:|:------:|:-------:|:---------:|:------:|
> |  RA     |    3*   |-97.7% | -100%|-66.3% |-75.85% |+7.45% | -82.6%|
> |  GCN   |      5  | -83.18%|-64.76% |-97% | -59.5%|-61.3% |-72.5%
> |  BUDDY   |   3*    |-91.0% |-64.7% | -86.1%|-75.4% |-67.2% | -84.48%
> |  NCNC   |  1     | -80.0%| -79.7%|-71.0% |-72.2% | -65.5%| -86.8%
> |  Neo-GNN   |   6    |-99.8% | -85.5%| -96.1%|-69.25% |-72.65% | -71.3%
> |  SEAL   |     2   |-98.5% | -96.6%|-81.9% |-71.12% | -78.0%|-85.0% |
>
> **Table 7 (shown above):** The rankings for all models with just HeaRT on ogbl-ppa, as determined by the Hits@20 Results from [Table 5 in the HeaRT paper](https://arxiv.org/pdf/2306.10453). The final six columns represent the change in Hits@20 for each LPShift split type and respective split direction versus the original HeaRT. *Note:* RA and BUDDY (3*) are tied for third best performing model. LPFormer was untested within the HeaRT paper and not included in these rankings.
>
>
> > Figure 3 still uses "inverse split".
>
> Our apologies, Figure 3 has been updated with the correct title indicating 'Backward' Splits.

---

> ### Author Response · Authors · 2024-11-27
> **Gentle Reminder about Rebuttal**
>
> Dear Reviewer RYKU,
>
> Given that today is the last day to revise the paper, we hope to have addressed your concerns about the challenge that LPShift poses.
>
> We have included LPShift's effect on performance (rebuttal Table 6 and 7) into Appendix Section J as Tables 23 and 24. We removed the HeaRT rankings from the paper revision given that it added noise to the percentage values in Tables 23 and 24. That and we believe that the percent change in performance is sufficient to indicate the challenge posed by LPShift as well as it's effect on HeaRT's original performance. We also provide discussion with insights on this effect, shown in blockquotes below:
>
> >> As shown in Appendix Section J, the ”Backward” CN and SP splits on ogbl-collab
> and all ogbl-ppa splits result in a 30−90% performance decrease from the HeaRT standard (Li et al.,
> 2024). This is especially notable given the difficulty HeaRT imposes on the original benchmark
> setting (Hu et al., 2020). We also note that the majority of PA and forward splits for ogbl-collab result
> in a 2-3 times performance increase. This result quantifies the prevailing assumption that GNN4LP
> generalize well in scenarios with increasing structural information (Mao et al., 2024; Wang et al.,
> 2023a). However, LPShift’s ogbl-ppa suffers from the inverse scenario, where the majority of splits
> result in a 60 − 90% decrease in performance. Therefore indicating that LPShift has stronger effects
> across dataset domains, particularly when the dataset is larger.
>
> We hope that this addresses your concerns and are happy to discuss any other concerns or questions.
>
> Best Regards,
>
> Authors

---

> > ### Comment · Reviewer_RYKU · 2024-11-29
> >
> > I sincerely appreciate the authors' comments on my questions.
> >
> > In general, I am still not convinced that LPShift is an effective way to impose a structural distribution shift on the graphs. From what the authors show in the rebuttal above, HeaRT seems to cause a more significant issue for GNN4LP compared to heuristics or basic GNNs, which contradicts the paper's claim that LPShift can interfere with the generalizability of GNN4LP.  For instance, NCNC and BUDDY seem to be much more robust than RA when introduced with LPShift compared to the results shown in the original paper. **Again, when HeaRT is plugged in as part of LPShift, it is very difficult to interpret whether LPShift is a significant method to cause the distribution shift and support the claim that "on LPShift datasets, GNN4LP methods frequently generalize worse than heuristics or basic GNNs".** I encourage the authors to reconsider the presentation of how to interpret the results with LPShift.
> >
> > Besides, if the authors want to keep HeaRT as part of the LPShift method, the ranking/performance with just HeaRT (as shown in the rebuttal above) should be kept, because this is the actual baseline LPShift should be compared. Without it, it can cause a misunderstanding to the readers such that LPShift alone can make RA generalize better than GNN4LP methods. I genuinely believe it is necessary to include them in the paper.
> >
> > Given my confusion about whether LPShift is a significant method w/wo HeaRT, I lower my confidence score and the presentation score accordingly.

---

> ### Author Response · Authors · 2024-12-01
> **Response #3 to Reviewer RYKU (1/N)**
>
> Thank you for your feedback, your perspective improves the quality of our work immensely! We apologize for the confusion and provide justifications below:
>
> * To clarify, Tables 6 and 7 from this rebuttal demonstrate the performance change that LPShift affects on the HeaRT baseline. Given the stark performance decrease seen on all ogbl-ppa splits and most of the "Backward" CN and SP ogbl-collab splits, LPShift serves as a means to induce a significant structural shift on the HeaRT baseline.
>
> * We would like to further clarify why we include HeaRT in LPShift. The random samples used in the original link prediction evaluation bear little correlation to the positive samples [1], making it easy for the supervised LP model to distinguish the positive link from the negative. For example, if we consider just structure, positive link samples with 5 Common Neighbors will be easy to distinguish from negative samples with fewer than 5 Common Neighbors when pulling from the same distribution. This is a problem for real-world scenarios, given that realistic positive and negative link samples are likely to have many features in common. Our use of HeaRT samples resolves this issue by aligning the structural features of the positive and negative samples used within LPShift's valid and testing distributions.
>
> [1] Li, Juanhui, et al. "Evaluating graph neural networks for link prediction: Current pitfalls and new benchmarking." Advances in Neural Information Processing Systems 36 (2024).
>
> We agree that clarity for LPShift's impact on HeaRT performance is important. To demonstrate this change in performance, we include the HeaRT and LPShift+HeaRT  ranks in Tables 8-11 below:
>
> We make the following observations about the ranked results:
> 1. GCN displays resilience to ogbl-ppa's LPShift, as indicated by an average rank increase of 2.875 on the "Forward" split and 1.75 on the "Backward" split. Neo-GNN performs notably well with a respective increase of 2.75 and 1.625.
> 2. NCNC displays resilience to ogbl-collab's LPShift, indicated by an average rank increase of 1.825 on the "Forward" split and 2.875 on the "Backward" split.
> 3. GCN consistently performs 2nd or 1st against GNN4LP models on the ogbl-ppa datasets.
> 4. RA overwhelmingly performs 1st on the ogbl-collab "Forward" splits.
> 5. Where RA fails on the ogbl-collab "Backward" split, GCN will take 1st or 2nd. The only exception being the 'PA - 100,50' split.
>
> Observations 1. and 2. provide clarity on the resilience of GCN, NCNC, Neo-GNN in dataset-dependent scenarios. It is worthwhile to note that NCNC experiences a sharp rank increase on 'CN - 4,2' and 'CN - 5,3' splits, indicating the impact of increased local structural information between training and validation.
>
> Observations 1., 3., 4., and 5. indicate LPShift scenarios where GNN4LP methods lose much of the competitive advantage established on the original HeaRT baseline.
>
>
> |ogbl-ppa|HeaRT|Avg. LPShift Rank|Avg. Rank Δ|  |CN - 1,2|CN - 2,4| CN - 3,5 | PA - 5k,10k|	PA - 10k,20k|PA - 15k,25k | SP - 17,26 | SP - 26, 36|
> |:-------:|:---------:|:------:|:------:|:------:|:-------:|:---------:|:------:|:------:|:------:|:------:|:------:|:------:|
> |  RA     |    3*    |3.875|-0.875 ||6|5|5|3|5|5|1|1|
> |  GCN   |      5    |2.125|+2.875 ||2|1|1|4|2|2|3|2|
> |  BUDDY   |     3*  |4.875|-1.875 ||3|6|6|6|4|6|4|4|
> |  NCNC   |     1    |2.625|-1.625 ||5|2|2|1|3|3|2|3|
> |  Neo-GNN   |  6    |3.25|+2.75 ||4|4|3|2|1|1|6|5|
> |  SEAL   |     2    |4.25|-2.25 ||1|3|4|5|6|4|5|6|
>
> **Table 8 (shown above):** The model rankings for HeaRT and the ogbl-ppa 'Forward' splits. HeaRT Rankings determined by the Hits@20 Results from [Table 5 in the HeaRT paper](https://arxiv.org/pdf/2306.10453). LPFormer is not included in this table since it was not included in HeaRT. (+) = Rank improvement, (-) = Rank decline.
>
> |ogbl-ppa|HeaRT|Avg. LPShift Rank|Avg. Rank Δ||CN - 2,1|CN - 4,2| CN - 5,3 | PA - 10k,5k|	PA - 20k,10k|PA - 25k,15k | SP - 26,17 | SP - 36, 26|
> |:-------:|:---------:|:------:|:------:|:------:|:-------:|:---------:|:------:|:------:|:------:|:------:|:------:|:------:|
> |  RA     |    3*    |3.75|-0.75||6|4|4|2|1|1|6|6|
> |  GCN   |     5     |3.25|+1.75||1|2|2|6|6|6|2|1|
> |  BUDDY   |   3*    |3|0||3|3|3|4|4|4|1|2|
> |  NCNC   |    1     |1.875|-0.875||2|1|1|1|2|2|3|3|
> |  Neo-GNN   |  6    |4.375|+1.625||5|6|6|5|5|5|4|4|
> |  SEAL   |    2     |4.125|-2.125||4|5|5|3|3|3|5|5|
>
> **Table 9 (shown above):** The model rankings for HeaRT and the ogbl-ppa 'Backward' splits. HeaRT Rankings determined by the Hits@20 Results from [Table 5 in the HeaRT paper](https://arxiv.org/pdf/2306.10453). LPFormer is not included in this table since it was not included in HeaRT. (+) = Rank improvement, (-) = Rank decline.

---

> ### Author Response · Authors · 2024-12-01
> **Response #3 to Reviewer RYKU (2/N)**
>
> |ogbl-collab|HeaRT|Avg. LPShift Rank|Avg. Rank Δ||CN - 1,2|CN - 2,4| CN - 3,5 | PA - 50,100|	PA - 100,200|PA - 150,250 | SP - 17,26 | SP - 26, 36|
> |:-------:|:---------:|:------:|:------:|:------:|:-------:|:---------:|:------:|:------:|:------:|:------:|:------:|:------:|
> |  RA     |    1    |1|0||1|1|1|1|1|1|1|1|
> |  GCN   |      3   |3.375|-0.375||3|4|3|3|3|2|4|5|
> |  BUDDY   |     2  |2.75|-0.75||2|2|2|4|4|4|2|2|
> |  NCNC   |     6   |4.125|+1.825||4|5|6|6|6|6|3|3|
> |  Neo-GNN   |   5  |5.125|-0.125||5|6|5|5|5|5|5|5|
> |  SEAL   |     4   |4|0||6|3|4|2|2|3|6|6|
>
> **Table 10 (shown above):** The model rankings for HeaRT and the ogbl-collab 'Forward' splits. HeaRT Rankings determined by the Hits@20 Results from [Table 5 in the HeaRT paper](https://arxiv.org/pdf/2306.10453). LPFormer is not included in this table since it was not included in HeaRT. (+) = Rank improvement, (-) = Rank decline.
>
> |ogbl-collab|HeaRT|Avg. LPShift Rank|Avg. Rank Δ||CN - 2,1|CN - 4,2| CN - 5,3 | PA - 100,50|	PA - 200,100|PA - 250,150 | SP - 26,17 | SP - 36, 26|
> |:-------:|:---------:|:------:|:------:|:------:|:-------:|:---------:|:------:|:------:|:------:|:------:|:------:|:------:|
> |  RA     |    1    |4.125|-3.125||6|6|2|4|1|2|6|6|
> |  GCN   |      3   |2.75|+0.25||1|2|4|3|5|5|1|1|
> |  BUDDY   |     2  |3|-1.0||2|3|6|1|4|4|2|2|
> |  NCNC   |     6   |3.125|+2.875||4|1|1|5|3|3|4|4|
> |  Neo-GNN   |   5  |4.375|+0.625||3|5|3|6|6|6|3|3|
> |  SEAL   |     4   |3.625|+0.375||5|4|5|2|2|1|5|5|
>
> **Table 11 (shown above):** The model rankings for HeaRT and the ogbl-collab 'Backward' splits. HeaRT Rankings determined by the Hits@20 Results from [Table 5 in the HeaRT paper](https://arxiv.org/pdf/2306.10453). LPFormer is not included in this table since it was not included in HeaRT. (+) = Rank improvement, (-) = Rank decline.
>
> We hope this clarifies how LPShift affects HeaRT as a baseline. Once allowed, we will include these new rank tables into the revised article's Appendix. In particular, LPShift's overwhelming effect on HeaRT's ogbl-ppa and many "Backwards" ogbl-collab splits indicate that a significant structural shift occurs from the HeaRT baseline. Please let us know if there are any further concerns, we will be happy to discuss them.

---

> > ### Comment · Reviewer_RYKU · 2024-12-03
> >
> > I appreciate the authors' detailed responses to my questions.
> >
> > First, let me explain again what evidence I am always trying to find in the paper/rebuttal so that it can pass the acceptance threshold: the empirical evidence showing that **LPShift is a significant method to cause the distribution shift and support the claim that "on LPShift datasets, GNN4LP methods frequently generalize worse than heuristics or basic GNNs"**. To me, this is why this paper is interesting to read since it brings new knowledge. Therefore, the paper needs to show that RA/GCN is more resilient than GNN4LP when LPShift is introduced.
> >
> > Because HeaRT is in place, the comparison should be between LPShift+HeaRT and HeaRT alone. Then it comes with tables 8-11 and observations above. Let's keep in mind that we need to show RA/GCN is more resilient than GNN4LP when LPShift is introduced:
> >
> > 1. "GCN displays resilience": In fact, both GCN and Neo-GNN perform fairly well as pointed out by the authors. GCN slightly wins.
> >
> > 3. "GCN consistently performs 2nd or 1st against GNN4LP models": Here, the observation is about absolute ranking in LPShift setting. In fact, NCNC is better on ogbl-ppa than GCN. GNN4LP wins.
> >
> > 4. "RA overwhelmingly performs 1st on the ogbl-collab "Forward" splits.": It does not help to show RA generalizes than GNN4LP since it is already 1st when HeaRT alone.
> >
> > 5. "Where RA fails on the ogbl-collab "Backward" split": On the ogbl-collab "Backward" split, RA drops dramatically when LPShift is introduced. If the conclusion here is that GCN generalized better, then similarly we conclude that RA generalizes worse.
> >
> > In general, the paper/rebuttal cannot convince me that the distribution shift imposed by LPShift makes advanced GNN4LP more vulnerable than basic heuristics or GCN. Therefore, I maintain my score.
> >
> > In fact, I think it can be beneficial to completely remove HeaRT from LPShift. Two reasons: 1. it makes the demonstration of experimental results easier to understand. 2. I suspect HeaRT alone can already introduce some structural distribution shift effect on graphs since it looks at negative testing edges with more Common Neighbors. And its effect is significant, probably more than LPShift. Without HeaRT, LPShift can show a stronger effect on the distribution shift issue.

---

> > > ### Author Response · Authors · 2024-12-03
> > > **Response #4 to Reviewer RYKU (1/N)**
> > >
> > > Thank you Reviewer RYKU, we are grateful for your effort with a continued and engaging discussion. In particular, the clarity of your review brings forth new insights from our results and strengthens link prediction knowledge as a whole.
> > >
> > > >> First, let me explain again what evidence I am always trying to find in the paper/rebuttal so that it can pass the acceptance threshold: the empirical evidence showing that LPShift is a significant method to cause the distribution shift and support the claim that "on LPShift datasets, GNN4LP methods frequently generalize worse than heuristics or basic GNNs". To me, this is why this paper is interesting to read since it brings new knowledge. Therefore, the paper needs to show that RA/GCN is more resilient than GNN4LP when LPShift is introduced.
> > >
> > > We agree that our claim: "on LPShift datasets, GNN4LP methods frequently generalize worse than heuristics or basic GNNs" is too absolute; especially given that our results indicate that the strength of LPShift is largely dataset-dependent. We will be certain to improve the language in our paper to be more specific about the *where* and *how* LPShift causes GNNs and Heuristics to generalize better than GNN4LP. In particular, detailing the difference between "Forward" and "Backward" LPShift splits.
> > >
> > > >> Because HeaRT is in place, the comparison should be between LPShift+HeaRT and HeaRT alone. Then it comes with tables 8-11 and observations above.
> > >
> > > We wish to clarify that all of the claims in this paper were made versus the HeaRT baseline. However, we agree that it would serve the paper well to include language that further clarifies this distinction. So, as not to mislead readers into believing this was conducted on random validation and testing samples. That, and making such a distinction allows us to better link our results to observations and insights developed with this rebuttal's Tables 8-11; allowing us to better break down which models LPShift has the most significant impact on.
> > >
> > > >> Let's keep in mind that we need to show RA/GCN is more resilient than GNN4LP when LPShift is introduced:
> > >
> > > For this key point, we will provide more evidence to support our main claim about LPShift's ability to induce structural shift in points 2-4. We forgeo point 1 since it supports our claim.
> > >
> > > >> 2. "GCN consistently performs 2nd or 1st against GNN4LP models": Here, the observation is about absolute ranking in LPShift setting. In fact, NCNC is better on ogbl-ppa than GCN. GNN4LP wins.
> > >
> > > In many scenarios where local structural information is incrementally more prevalent in validation and testing, such as CN - 4,2 and CN - 5,3 for ogbl-ppa, NCNC will perform the best overall. NCNC will even go so far as to increase it's mean performance incrementally: CN - 4,2 = 18.84 and CN - 5,3 = 22.08. Given that these two CN scenarios are similar to the distribution shift in ogbl-collab (shown in Figure 5 of our paper), it is interesting to note that NCNC actually shows resilience to structural shift if there are samples that have at least a few Common Neighbor present within validation and testing. However, we note that with the CN - 2,1 for both datasets; GCN nearly doubles NCNC's Hits@20 performance on ogbl-ppa and performs well-over 8x better than NCNC on ogbl-collab. Thus, indicating that NCNC is sensitive to distribution shifts that involve incoming links with limited Common Neighbors. A result that was not quantifiably before LPShift.
> > >
> > > >> 3. "RA overwhelmingly performs 1st on the ogbl-collab "Forward" splits.": It does not help to show RA generalizes than GNN4LP since it is already 1st when HeaRT alone.
> > >
> > > It is certainly the case that RA ranks first on HeaRT's ogbl-collab. However, the margins between performance for different models present within the original HeaRT paper is much closer than with LPShift. We include Table 12 below to detail this:
> > >
> > > ||HeaRT |LPShift - CN 1,2 |CN 2,4|CN 3,5|
> > > |:---:|:---:|:---:|:---:|:---:|
> > > |RA|6.29|32.22|29.74|29.86|
> > > | Mean Δ| -0.14|+25.16 |+9.14 |+10.08 |
> > > |---|---|---|---|---|
> > > |SEAL|6.43±0.32|7.06±1.67|20.60±6.23|19.78±1.24|
> > >
> > > **Table 12 (shown above):** MRR results for LPShift's ogbl-collab "Forward" splits on RA and SEAL with the Mean Change between the two model
> > >
> > > From Table 12, we determine that not only does LPShift have a significant enough effect to drastically alter SEAL's performance; but also induces a scenario where the structural shift causes RA to always perform at least 9 MRR better than SEAL. A gap that does not exist with just HeaRT. This occurs regardless of whether more structural information is present in the training distribution within CN - 2,4 and CN - 3,5. Therefore, detailing the effects of a scenario where incoming validation and testing samples do not align with training.

---

> ### Author Response · Authors · 2024-12-03
> **Response #4 to Reviewer RYKU (2/N)**
>
> >> 4. "Where RA fails on the ogbl-collab "Backward" split": On the ogbl-collab "Backward" split, RA drops dramatically when LPShift is introduced. If the conclusion here is that GCN generalized better, then similarly we conclude that RA generalizes worse.
>
> RA ranks whether a link will form based on the RA heuristic. So, this result is a natural occurrence of the "Backwards" splits; where the training distribution has far more structural information than the testing distribution. Therefore, meaning the ranking predictor has limited information to effectively rank links. We agree that the language in the paper should make this distinction about the "Backward" split clear.
>
> >> In fact, I think it can be beneficial to completely remove HeaRT from LPShift. Two reasons: 1. it makes the demonstration of experimental results easier to understand.
>
> Although including HeaRT adds a level of nuance to our results, the random samples would not necessarily make the experimental results easier to understand. Many of the assumptions within our paper are made to simulate how link prediction operates within the real world.
>
> Within the real-world setting, it is not apparent which validation and testing link samples are positive or negative. So, AI practitioners are required to extract a set of potential link candidates and then rank them based on some metric, like Common Neighbors or a Random-Walk, in order to build their positive and negative sets. HeaRT discusses this further in Appendix Section G.1 of their paper [1]. As such, we felt that HeaRT was the perfect addition to simluate the real-world scenario. However, we agree that this distinction is not clear within the paper and we will be certain to include language that clarifies this.
>
>
> >> 2. I suspect HeaRT alone can already introduce some structural distribution shift effect on graphs since it looks at negative testing edges with more Common Neighbors. And its effect is significant, probably more than LPShift. Without HeaRT, LPShift can show a stronger effect on the distribution shift issue.
>
> This is an interesting consideration, and one of the key reasons why we study LPShift. Given that training samples are always random and likely to have 0 Common Neighbors; it then follows that ranking samples based on some heuristic is likely to introduce bias/shift when compared against validation and testing samples that are likely to contain at least a single Common Neighbor.
>
> However, we note that even though LPShift use CNs as the primary heuristic for constructing negative validation and testing samples. HeaRT also avoids shift and bias by applying random-walk and PPR in order to capture global information and unique sample candidates (also detailed in Appendix Section G.1 of the HeaRT paper) [1].
>
> Also, we note that any sampling techniques will draw from the bias or shift that is already present within the initial graph. Given this, HeaRT serves as a natural extension of LPShift since it operates on the same principle as LPShift.
>
> [1] Li, Juanhui, et al. "Evaluating graph neural networks for link prediction: Current pitfalls and new benchmarking." Advances in Neural Information Processing Systems 36 (2024). https://arxiv.org/pdf/2306.10453

---

### Official Review · Reviewer_15W6 · 2024-10-18

**Soundness:** 3
**Presentation:** 3
**Contribution:** 2
**Rating:** 6
**Confidence:** 4

**Summary:**

This paper benchmarks the generalizability of LP models under distribution shifts, which is often neglected in link-level tasks. It introduces a dataset splitting strategy called LPShift, which uses structural properties to induce distribution shifts, enabling the study of LP model performance under such conditions. The authors evaluate various LP models and generalization techniques on 16 LPShift-generated dataset splits. The paper concludes that structural shifts significantly impact LP model performance, advocating for further exploration of generalization techniques.

**Strengths:**

1. The problem of link prediction under distribution shifts is not fully explored, and the paper has the potential to bridge the important gap.
2. The proposed LPShift strategy can be applied to real-world datasets, offering a practical tool for researchers to evaluate the generalization of LP models in realistic scenarios.
3. Many experimental results are provided showing the limitations of current LP-specific generalization methods.

**Weaknesses:**

1. The writing is problematic with many grammar errors and confusing sentences. For example, the CN example in the introduction confuses me without a specific source node and target node context. What is a "red" graph? Line 52: requires $\rightarrow$ require.
2. All splits are based on statistical heuristics without any real-world shift splits, e.g., temporal shift, which produces not enough the significance.
3. Traditional generalization baselines are missing, e.g., IRM, VREx, etc. It is not sure whether not LP specific methods can already solve the LP OOD problem.
4. The hyper-parameter sweeping space is not large enough to produce significant experimental conclusions.
5. As a benchmark, no code is provided.

**Questions:**

1. Why are the proposed splits significant? For example, if a generalization method works well on your split, which real OOD problem can the method solve? Can you provide experiments and discussions to validate that?

---

> ### Author Response · Authors · 2024-11-21
> **Response to Reviewer 15W6 (1/4)**
>
> > The writing is problematic with many grammar errors and confusing sentences. For example, the CN example in the introduction confuses me without a specific source node and target node context. What is a "red" graph? Line 52: requires require.
>
> We apologize for any confusion caused by a lack of clarity in our writing. We make the necessary edits in our revised paper, as highlighted in blue. We include an example of this revision in block quotes below:
>
> >> We apply the 2-sample Kolmgorov-Smirnov (KS) test (Hodges Jr, 1958) to compare training, validation, and testing distributions before and after applying LPShift. As a controlled baseline to test LPShift, we generate a Holme-Kim (HK) graph (Holme & Kim, 2002) with a 40% chance to close a triangle, allowing the HK graph to contain numerous Common Neighbors without
> becoming fully-connected.
>
> Additionally, we update the writing with a better description for Figure 1 in the Introduction section. We include this revision in block quotes below:
>
> >> Consider the three graphs in Figure 1 as samples pulled from a larger graph dataset. In order of appearance, each of the three graphs represent samples from the: training, validation, and testing splits of our dataset. Our task is to predict whether new links will form between existing nodes. These predicted links are shown as dotted lines with colors: “green” = training, “blue” = validation, “red” = testing. An optimal LP model trained on the first graph sample effectively learns a representation understanding source and target nodes with 2 Common Neighbors Zhang & Chen (2018), making the model effective at predicting new links in structure-rich scenarios. However, such a model is likely to fail when predicting links on the subsequent validation and testing samples; the second and third graph in Figure 1 with 1 CN and 0 CNs, respectively. The learned representation necessary to capture the differing pairwise relations in the second and third graphs requires generalizing to a scenario with fewer CNs, and therefore less structural information. As such, the previously-mentioned scenario represents a case of structural covariate shift, where the training distribution cannot effectively model the testing distribution, defined mathematically as P^{Train}(X)̸ = P^{Test}(Y). LPShift, our proposed splitting strategy, provides a controllable means to induce the scenario shown in Figure 1 (labelled in experiments as the CN - 2,1 split), as well as fifteen other tested scenarios. More details on structural shift are included in Section 3.2.
>
> > All splits are based on statistical heuristics without any real-world shift splits, e.g., temporal shift, which produces not enough the significance.
>
> We would like to clarify core contributions of LPShift to demonstrate it's significance:
>
> * The application of LPShift is novel. LPShift uses structural heuristics in a flexible splitting strategy, where a user controls the: magnitude, type, and direction of the structural shift in their desired dataset. Since the user is aware of the magnitude, type, and direction of shift that they applied to their dataset; any further analysis on the updated dataset enables understanding distribution shift in link prediction in a way that has never been previously quantified.
>
> * Figure 5 within our submission indicates that ogbl-collab, a dataset with temporal shift also experiences a structural shift with common neighbors. Given that a change in graph structure functions as a change over time, then temporal shift and structural shift can be considered synonymous with one another. For example, the CN - 2,1 split of LPShift (as shown in our updated Figure 1) causes models to learn on a well-connected training split and then generalize on disconnected testing data, just like temporal shift. Additionally, link prediction heuristics are integrated into high-performing GNN4LP models, such as RA in BUDDY and CN into NCNC, indicating their effectiveness as tools for enhancing link-prediction models and relevance to the link prediction problem.

---

> ### Author Response · Authors · 2024-11-21
> **Response to Reviewer 15W6 (2/4)**
>
> > Traditional generalization baselines are missing, e.g., IRM, VREx, etc. It is not sure whether not LP specific methods can already solve the LP OOD problem.
>
> Our initial study could not consider traditional generalization methods given that the traditional methods rely on invariance between training environments to generalize in OOD scenarios. LPShift considers structural covariate shift, $P^{Train}(X) ≠ P^{Test}(Y)$, within a single environment, making evaluation of the traditional generalization baselines impractical.
>
> To overcome this barrier, we construct an updated test setting which applies a given link prediction heuristic on each LPShift split to build training environment subsets with distinctive structural characteristics. Given these distinctive structural characteristcs, the environmental subsets are well-suited for testing how well the traditional generalization methods work under LPShift.
>
> Following the standard set in the GOOD paper [1], we tuned the parameters for IRM across {0.1, 1, 10} and VREx across {10, 100, 100}.
>
> We include results which apply IRM and VREx to BUDDY and GCN for all tested splits in the table detailed below:
>
> | Collab | CN - 1,2 | CN - 2,4 | CN - 3,5 | SP - 17,26 | SP - 26,36 | PA - 50,100 | PA - 100,200 | PA - 150,250 |
> |:--------:|:-------------:|:------------:|:-------------:|:--------------------:|:------------:|:-------------:|:------------:|:------------:|
> | BUDDY | 17.48 ± 1.19 | 15.47 ± 0.57 | 16.60 ± 0.89 | 16.20 ± 1.40 | 16.42 ± 2.30 | 21.27 ± 0.74 | 14.04 ± 0.76 | 13.06 ± 0.53 |
> | +IRM | 1.94 ± 0.16 | 13.42 ± 2.15 | 14.18 ± 0.97 | 4.92 ± 2.55 | 5.05 ± 0.91 | 18.17 ± 3.87 | 9.40 ± 2.22 | 9.40 ± 1.92 |
> | +VREx | 14.76 ± 0.58 | **29.81 ± 0.06** | 16.32 ± 0.51 | 11.36 ± 0.92 | 17.56 ± 0.71 | 22.36 ± 0.48 | 14.30 ± 0.26 | 13.30 ± 0.27 |
> | -------- | -------- | -------- | -------- | -------- | -------- | -------- | -------- | -------- |
> | GCN | 12.92 ± 0.31 | 15.20 ± 0.16 | 17.54 ± 0.19 | 10.29 ± 0.52 | 12.94 ± 0.59 | 20.78 ± 0.25 | 14.66 ± 0.20 | 14.03 ± 0.15 |
> | +IRM | 9.29 ± 0.25 | 12.15 ± 0.38  | 12.00 ± 0.61   | 9.93 ± 0.48 | 10.15 ± 0.72 | 18.77 ± 0.52 | 13.75 ± 0.13 | 12.08 ± 0.74 |
> | +VREx | **14.03 ± 0.428**  | **17.58 ± 0.39**  | 17.56 ± 0.16  | **11.47 ± 0.45**  | **13.96 ± 0.76**  | **21.13 ± 0.35**  | 14.68 ± 0.08  | 13.81 ± 0.07  |
>
> **Table 1 (shown above):** MRR Results for VREx and IRM applied to GCN and BUDDY on the "Forward" LPShift splits of ogbl-collab.
>
> | Collab | CN - 2,1 | CN - 4,2 | CN - 5,3 | SP - 26,17 | SP - 36,26 | PA - 100,50 | PA - 200,100 | PA - 250,150 |
> |:--------:|:-------------:|:------------:|:-------------:|:--------------------:|:------------:|:-------------:|:------------:|:------------:|
> | BUDDY | 3.70 ± 0.13 | 3.55 ± 0.09 | 5.40 ± 0.04 | 6.73 ± 0.32 | 3.71 ± 0.39 | 24.95 ± 0.92 | 15.52 ± 0.73 | 13.36 ± 0.83 |
> | +IRM | 1.81 ± 0.29 | 2.63 ± 0.16 | 3.82 ± 0.17 | 4.93 ± 3.73 | 1.72 ± 0.21 | 16.99 ± 3.53 | 15.76 ± 3.08 | 12.29 ± 4.12 |
> | +VREx | 2.65 ± 0.08 | 3.00 ± 0.11 | 4.99 ± 0.13 | 5.10 ± 1.01 | 18.12 ± 1.41 | 22.51 ± 0.76 | 16.01 ± 1.10 | **17.01 ± 0.65** |
> | -------- | -------- | -------- | -------- | -------- | -------- | -------- | -------- | -------- |
> | GCN | 5.78 ± 0.19 | 5.91 ± 0.05 | 7.07 ± 0.11 | 8.69 ± 0.47 | 6.47 ± 0.37 | 21.38 ± 0.27 | 13.36 ± 0.24 | 11.95 ± 0.35 |
> | +IRM | 4.54 ± 0.54 | 4.87 ± 0.20 | 5.18 ± 0.17 | 6.41 ± 0.35 | 4.65 ± 0.96 | 15.15 ± 0.69 | 10.01 ± 0.57 | 8.63 ± 0.57 |
> | +VREx | 5.84 ± 0.31 | **6.04 ± 0.08** | **7.22 ± 0.14** | 8.55 ± 0.42 | 6.49 ± 0.19 | 21.30 ± 0.31 | 13.28 ± 0.28 | 11.94 ± 0.26 |
>
> **Table 2 (shown above):** MRR Results for VREx and IRM applied to GCN and BUDDY on the "Backward" LPShift splits of ogbl-collab.
>
> From our tests we make three observations:
> 1. VREx is the only method that increases performance under LPShift. The CN - 1,2 split for ogbl-collab is notable example, given that it nearly doubles BUDDY's original performance.
> 2. IRM fails to improve performance across all splits and even reduces performance by 33% in certain scenarios; such as the "Backward" splits for SP on ogbl-collab.
> 3. Since IRM and VREx rely on multiple environmental subsets in the training data, training all of these methods is significantly slower following our current methodology of dividing invariant subsets.

---

> ### Author Response · Authors · 2024-11-21
> **Response to Reviewer 15W6 (3/4)**
>
> > The hyper-parameter sweeping space is not large enough to produce significant experimental conclusions.
>
> Our initial study analyzed 32 unique dataset splits. From which we measured the performance over 5 different seeds, and tuned over a grid search-space of {dropout: 0.1, 0.3} and {learning rate: 1e-3, 1e-2}. When we consider the 7 tested models, this results in a total of 4,480 independent runs to conduct just the baseline experiments for this paper. Additional experiments were necessary for the analysis.
>
> Initial parameter tuning within our study included tuning parameters such as batch size and L2-regularization. However, tuning these two parameters provided limited effect on performance. Given the massive computation load for conducting baseline experiments, we prioritized conducting additional analysis over including more parameters in our grid search.
>
> Additionally, it is quite common for graph benchmark papers to limit fine-tuning of tested models and prioritize testing more models or datasets. We include a list detailing tuning practices for other graph-related benchmarks:
>
> * GOOD [1] -- Fixed the baseline model parameters and focused on tuning single hyperparameters in generalization algorithms.
> * DrugOOD [2] -- Grid search over learning rate and batch size, average performance over 3 random seeds.
> * WILDS [3] -- Grid search over learning rate and L2-regularization strengths on publicly-available pre-trained models.
> * OGB [4] -- Fixed baseline model parameters, tuned on dropout ratio between 0.0 and 0.5.
>
> [1] Gui, Shurui, et al. "GOOD: a graph out-of-distribution benchmark." Proceedings of the 36th International Conference on Neural Information Processing Systems. 2022.
> [2] Ji, Yuanfeng, et al. "DrugOOD: Out-of-Distribution (OOD) Dataset Curator and Benchmark for AI-aided Drug Discovery--A Focus on Affinity Prediction Problems with Noise Annotations." arXiv preprint arXiv:2201.09637 (2022).
> [3] Koh, Pang Wei, et al. "Wilds: A benchmark of in-the-wild distribution shifts." International conference on machine learning. PMLR, 2021.
> [4] Hu, Weihua, et al. "Open graph benchmark: Datasets for machine learning on graphs." Advances in neural information processing systems 33 (2020): 22118-22133.
>
> As such, our hyperparameter grid-search across 6 variants of learning rate and dropout along with our initial batch size investigations reaches the standard set by other graph benchmark papers. This is especially true given that any remaining hyperparameters were fixed following guidance provided in the original paper for a given model and other graph benchmarks measure performance across fewer seeds than what LPShift is tested on.
>
> We conduct grid-search tuning on dropout: {0.1, 0.3} and learning rate: {1e-4, 5e-3} for GCN across all splits on ogbl-collab and ogbl-ppa to demonstrate empirically the robustness of our model performance. With the best results for MRR detailed below:
>
> | Collab | CN - 1,2 | CN - 2,4 | CN - 3,5 | SP - 17,26 | SP - 26,36 | PA - 50,100 | PA - 100,200 | PA - 150,250 | CN - 2,1 | CN - 4,2 | CN - 5,3 | SP - 26,17 | SP - 36,26 | PA - 100,50 | PA - 200, 100 | PA - 250,150 |
> |:--------:|:-------------:|:------------:|:-------------:|:--------------------:|:------------:|:-------------:|:------------:|:------------:|:------------:|:------------:|:------------:|:------------:|:------------:|:------------:|:------------:|:------------:|
> | GCN | 12.51 ± 0.90 | **18.50 ± 0.28** | **18.05 ± 0.22** | 10.62 ± 0.55 | 11.81 ± 0.88 | 21.32 ± 0.30 | 15.34 ± 0.12 | 13.01 ± 0.06 | 5.55 ± 0.11 | 5.95 ± 0.07 | 7.15 ± 0.10 |8.62 ± 1.04 | 6.55 ± 0.22 |19.19 ± 0.65 |11.81 ± 0.34 |10.90 ± 0.40|
>
> **Table 2 (shown above) :** MRR Results for GCN on LPShift's ogbl-collab with additional tuning on dropout and learning rate.
>
> | PPA | CN - 1,2 | CN - 2,4 | CN - 3,5 | SP - 17,26 | SP - 26,36 | PA - 5k,10k | PA - 10k,20k | PA - 15k,25k | CN - 2,1 | CN - 4,2 | CN - 5,3 | SP - 26,17 | SP - 36,26 | PA - 10k,5k | PA - 20k, 10k | PA - 25k,15k |
> |:--------:|:-------------:|:------------:|:-------------:|:--------------------:|:------------:|:-------------:|:------------:|:------------:|:------------:|:------------:|:------------:|:------------:|:------------:|:------------:|:------------:|:------------:|
> | GCN |8.37 ± 0.17 | 8.13 ± 0.72 |8.28 ± 0.56|4.98 ± 0.38|5.37 ± 0.12|  4.32 ± 0.13| 5.46 ± 0.25| 6.13 ± 0.53 |3.50 ± 0.10|3.12 ± 0.09| 3.06 ± 0.05| 10.07 ± 0.22 |3.33 ± 0.11| 1.79 ± 0.36|1.49 ± 0.02|1.49 ± 0.01|
>
> **Table 3 (shown above) :** MRR Results for GCN on LPShift's ogbl-ppa with additional tuning on dropout and learning rate.

---

> ### Author Response · Authors · 2024-11-21
> **Response to Reviewer 15W6 (4/4)**
>
> Given Table 2 and 3, we make three observations:
> 1. Even with additional tuning on learning rate and dropout, only two of the thirty-two possible results receive a performance boost large enough to eliminate overlap between the score distributions from our original GCN results.
> 2. The only scenarios where further tuning increases performance is the CN - 2,4 and CN - 3,5 split on ogbl-collab. We note that this follows general assumptions with link prediction, where a model can be expected to generalize more effectively on validation and testing data with more structural information than training.
> 3. None of the "Backward" splits for both datasets receive any noticeable change in their originally-reported performance. This follows along with the initial motivation provided by Figure 1 in our revised paper. As such, experimental conclusions drawn about the extent that LPShift impacts performance holds. So, this result actually serves to bolster any experimental conclusions about the impact of LPShift, given the principle that all of the "Backward" LPShift splits subsequently lose structural information from training to validation and then testing.
>
> > As a benchmark, no code is provided.
>
> We are sorry the code was not available within our main paper submission. We include the link from Appendix Section C here: [Anonymous GitHub Link](https://anonymous.4open.science/r/LP_OOD-75D7/README.md)
>
> > Why are the proposed splits significant? For example, if a generalization method works well on your split, which real OOD problem can the method solve? Can you provide experiments and discussions to validate that?
>
> Subplot 1 in Figure 5 of our original submission details that the unaltered ogbl-collab dataset suffers from a structural shift in Common Neighbors. From there, Figure 6 demonstrates that the CN distribution shift induced by LPShift can behave like the temporal shift in ogbl-collab. Given that Figure 6 and 1 are both concerned with CN - 2,1 split, we conclude that the decreased structural information contained within the validation and testing simulates a scenario where LP models must generalize to new nodes with relatively few edge make connections to an existing dataset; like how new papers cite previous work but have no citations themselves.

---

> > ### Comment · Reviewer_15W6 · 2024-11-21
> >
> > Thank you for the authors' reply. The first and second concerns are well addressed. However,
> >
> > 1. Compared to the previous benchmark like GOOD the authors mentioned. The traditional OOD baselines are still not enough. I appreciate the new results on IRM and VREx, where the authors found good performances on VREx. I believe the authors have already realized that, in this case, it is necessary to evaluate traditional OOD baselines comprehensively. Please refer to the GitHub Repo of DomainBed [1] for more typical baselines.
> > 2. As a benchmark, it is important to be transparent about all the details. Instead of claiming using the recommended hyperparameter searching spaces, please list them explicitly. I have seen many works saying that using recommended hyperparameters without really doing a full evaluation.
> > 3. The quality of the code provided is not high enough without minimal instructions, documentation, and proper coding structures, which is not usable for most of the readers. I highly encourage the authors to align their work with previous benchmarks.
> >
> > [1] Gulrajani, Ishaan, and David Lopez-Paz. "In search of lost domain generalization." arXiv preprint arXiv:2007.01434 (2020).

---

> ### Author Response · Authors · 2024-11-23
> **Response to Reviewer 15W6 Comment #1 (1/2)**
>
> > Compared to the previous benchmark like GOOD the authors mentioned. The traditional OOD baselines are still not enough. I appreciate the new results on IRM and VREx, where the authors found good performances on VREx. I believe the authors have already realized that, in this case, it is necessary to evaluate traditional OOD baselines comprehensively. Please refer to the GitHub Repo of DomainBed [1] for more typical baselines.
>
> Thank you for your prompt response! We agree additional evaluation is important to understand the scope of the problem posed by LPShift. To demonstrate this, we include current results for GroupDRO, DANN, and Deep CORAL in Table 4. The generalization methods were tuned over the same parameter space as GOOD:
> * GroupDRO: {0.1, 0.01, 0.001}
> * DANN: {0.01, 0.1, 1.0}
> * Deep CORAL: {0.01, 0.1, 1.0}
>
> We note that our 5 newly-tested generalization methods are comprehensive in the breadth of their functionality for improving performance under distribution shift:
> * IRM determines invariant correlations across training environments [1].
> * VREx is specially designed for covariate shift and extrapolating to variance within training environments [2].
> * GroupDRO regularizes weights for the worst training environments to interpolate towards better performance [3].
> * DANN applies additional layers to generate adversarial predictions to then enhance domain adaptation [4].
> * Deep CORAL computes a covariance and applies it in order to better align training and testing distributions [5].
>
> | Collab | CN - 1,2 | CN - 2,4 | CN - 3,5 | SP - 17,26 | SP - 26,36 | PA - 50, 100 | PA - 100, 200 | PA - 150, 250 | CN - 2,1 | CN - 4,2 | CN - 5,3   | PA - 100, 50 | PA - 200, 100 | PA - 250, 150 | SP - 26,17 | SP - 36,26 |
> |:--------:|:-------------:|:------------:|:-------------:|:--------------------:|:------------:|:------------:|:------------:|:------------:|:------------:|:------------:|:------------:|:------------:|:------------:|:------------:|:------------:|:------------:|
> | GroupDRO |5.59 ± 0.52 | 5.03 ± 0.34|5.68 ± 0.32 |5.74 ± 0.45 | 6.61 ± 0.65| 9.33 ± 1.28|7.46 ± 0.95 |6.10 ± 0.77 | 2.49 ± 0.07 |2.93 ± 0.18 | 3.04 ± 0.08|4.04 ± 1.24 |2.93 ± 1.06 | 2.63 ± 0.73| 2.60 ± 0.32 | 2.35 ± 0.26|
> | DANN | 9.91 ± 0.24 | 10.43 ± 0.39 | 12.50 ± 0.37 | **10.69 ± 0.48** | 11.21 ± 0.85| **21.51 ± 0.15** | **14.88 ± 0.18** | 12.97 ± 0.15 | 6.01 ± 0.15 | 5.95 ± 0.14 | 6.49 ± 0.07 | 14.68 ± 0.38 | 9.38 ± 0.20 | 8.62 ± 0.24 | 8.73 ± 0.54 | 6.67 ± 0.19|
> | Deep CORAL | 9.52 ± 0.36 | 10.08 ± 0.62 | 12.40 ± 0.24 | **10.62 ± 0.51**| 10.99 ± 0.25 | **21.50 ± 0.11** | 14.65 ± 0.22 | 12.80 ± 0.24 | 5.98 ± 0.12 | 6.01 ± 0.15 | 6.48 ± 0.16 | 14.73 ± 0.44 | 9.42 ± 0.09 | 8.42 ± 0.45 | 8.57 ± 0.37 | 6.54 ± 0.17 |
>
> **Table 4 (shown above):** MRR results for GroupDRO with GCN on LPShift's ogbl-collab splits.
>
> We make three additional observations based on Table 2 and Table 4:
> 1. DANN and Deep CORAL improve performance on just the SP - 17,26 and PA - 50, 100, and PA - 100, 200 splits.
> 2. GroupDRO does not succeed at improving performance.
> 3. All "Backward" splits do not experience a significant increase in performance.
>
> We attribute all observations to the graph structure in the training environments. There are two explanations:
> * The CN - 1,2 split contains limited structure within the training environment (i.e. zero Common Neighbors), meaning that invariance across each training environment has limited usefulness for improving downstream performance.
> * The "Backward" splits have plenty of structural information within their training environments (i.e. CN - 5,3 contains 5 Common Neighbors at minimum) but must generalize to validation and testing splits with increasingly limited structural information. Therefore, eliminating any advantage provided by our 5 tested generalization methods.
>
> [1] Arjovsky, Martin, et al. "Invariant risk minimization." arXiv preprint arXiv:1907.02893 (2019).
>
> [2] Krueger, David, et al. "Out-of-distribution generalization via risk extrapolation (rex)." International conference on machine learning. PMLR, 2021
>
> [3] Sagawa, Shiori, et al. "Distributionally robust neural networks for group shifts: On the importance of regularization for worst-case generalization." arXiv preprint arXiv:1911.08731 (2019).
>
> [4] Ganin, Yaroslav, et al. "Domain-adversarial training of neural networks." Journal of machine learning research 17.59 (2016): 1-35.
>
> [5] Sun, Baochen, and Kate Saenko. "Deep coral: Correlation alignment for deep domain adaptation." Computer Vision–ECCV 2016 Workshops: Amsterdam, The Netherlands, October 8-10 and 15-16, 2016, Proceedings, Part III 14. Springer International Publishing, 2016.

---

> ### Author Response · Authors · 2024-11-23
> **Response to Reviewer 15W6 Comment #1 (2/2)**
>
> In order to better align with GOOD [6], we attempted to apply Mixup for LPShift. However, the original Mixup does not function for graph data [7]. There is an implementation of Mixup designed for node- and graph-classification [8], but has no readily-apparent link-prediction equivalent.
>
> [6] Gui, Shurui, et al. "Good: A graph out-of-distribution benchmark." Advances in Neural Information Processing Systems 35 (2022): 2059-2073.
>
> [7] Zhang, Hongyi. "mixup: Beyond empirical risk minimization." arXiv preprint arXiv:1710.09412 (2017).
>
> [8] Wang, Yiwei, et al. "Mixup for node and graph classification." Proceedings of the Web Conference 2021. 2021.
>
> > As a benchmark, it is important to be transparent about all the details. Instead of claiming using the recommended hyperparameter searching spaces, please list them explicitly. I have seen many works saying that using recommended hyperparameters without really doing a full evaluation.
>
> We agree that transparency on model parameters is critical for effective AI research, especially benchmarking work. We have revised our paper to include fixed hyperparameters in Appendix Section D - Table 21 of our revised paper (also shown below):
>
> |Model|Dataset|Model Layers | Predictor Layers | Hidden Channels |
> |:-----:|:-----:|:-----:|:-----:|:-----:|
> |GCN     |ogbl-collab | 3| 3|128 |
> | |ogbl-ppa | 3| 3| 128 |
> |-----|-----|-----|-----|-----|
> |BUDDY   |ogbl-collab | 3| 3| 256 |
> | |ogbl-ppa | 3| 3|256 |
> |-----|-----|-----|-----|-----|
> |NCNC    |ogbl-collab | 3| 3|256 |
> | |ogbl-ppa | 3| 3|256 |
> |-----|-----|-----|-----|-----|
> |NeoGNN  |ogbl-collab | 3| 3|256 |
> | |ogbl-ppa | 3| 3|256 |
> |-----|-----|-----|-----|-----|
> |LPFormer|ogbl-collab | 3| 3|128 |
> | |ogbl-ppa | 3| 3|128 |
> |-----|-----|-----|-----|-----|
> |SEAL    |ogbl-collab | 3| 3|256 |
> | |ogbl-ppa | 3| 3|256 |
>
> **Table 5 (shown above):** Fixed model hyperparameters for each tested model across respective LPShift datasets.
>
> We also provide additional details in Appendix Section D on parameters that are not set by default within our updated code repository, shown here in block quotes:
>
> >> • NCNC for all datasets and splits, besides the ogbl-ppa PA splits, considers the ’NCNC2’
> variant of NCNC with an added depth argument of 2 Wang et al. (2023a). For the ogbl-ppa
> PA splits, we apply a depth argument of just 1 in order to ensure that a single seeded run
> does not exceed 24 hour runtime.
> • NeoGNN use 2-hop neighborhoods on the LPShift ogbl-collab datasets and 1-hop on the
> LPShift ogbl-ppa datasets.
> • Initial investigations on batch size fixed learning rate at 1e−3 and dropout at
> 0.1, model performance was then tested for a single run across a space of
> {8, 16, 32, 64, 128, 256, 512, 1024, 2048, 4096, 8192, 16384, 32768, 65536}. This was done with the intent to balance computational time with performance when conducting full experiments.
>
> > The quality of the code provided is not high enough without minimal instructions, documentation, and proper coding structures, which is not usable for most of the readers. I highly encourage the authors to align their work with previous benchmarks.
>
> The replication of results is important for us as well. To better align with benchmarks like HeaRT and GOOD, we have updated our README, gen_synth.sh, and run/ script files. All files are available at the following link: [Updated Anonymous Repository](https://anonymous.4open.science/r/LP_OOD-75D7/README.md). *Note:* All hyperparameters in their respective run/{model_name}.sh files are fixed to recreate results within our paper.

---

> ### Author Response · Authors · 2024-11-25
> **Upcoming Rebuttal Deadline**
>
> Dear Reviewer 15W6,
>
> Thank you for taking the time to review our paper!
>
> As the discussion deadline is rapidly approaching, we are eager to hear your response on our rebuttal as soon as possible. We hope that all aspects of our response addresses your concerns and await additional questions.
>
> Regards,
>
> Authors

---

> > ### Comment · Reviewer_15W6 · 2024-11-26
> >
> > I appreciate the authors' extensive efforts, but the results seem not updated in the revision yet. I will consider raising my score once the authors finish revising the paper and improve the Repo further.

---

> ### Author Response · Authors · 2024-11-26
> **Reply to Reviewer 15W6**
>
> Thank you, we appreciate your dedication to help us improve our work!
>
> We include the results for the traditional OOD generalization method in our revised paper's Appendix Section H as Tables 19 and 20. We include a reference to the Tables including appropriate citations in Section 4.4 under the subsection 'Does GNN4LP generalize and do generalization methods work?'. Shown in block quotes below:
>
> >> Traditional generalization methods (Arjovsky et al., 2019; Sagawa et al., 2019; Ganin et al., 2016; Sun & Saenko, 2016), such as VREx (Krueger et al., 2021), achieve some promising results. However, these gains are typically marginal, especially on the ”Backward” ogbl-collab split; as indicated in Appendix Section H
>
> We note that the anonymous repository is the *functional* code for this paper and includes a detailed **README** and **shell scripts** to enable replication of baseline results reported in this paper.
>
> Given the technical debt within the repository, it will likely take longer than the review period to truly improve the code's standard. However, we acknowledge that the code repository is unwieldy and provide a plan detailed below:
>
> **Within the next few days:**
>  * Flatten the folder structure to provide easier access to code modules.
> * Re-organize modules for easier results replication.
>
> **Within the next few months:**
> * Build a pip package or integrate LPShift into PyTorch Geometric.
> * Archive the current working repository with a link to a re-factored project repository.
>
> We hope this addresses your concerns, please let use know if there are any specific pain points within the code repository that need addressing.

---

> > ### Comment · Reviewer_15W6 · 2024-11-27
> >
> > The revised manuscript is much better regarding the presentation and experiments than the initial version. The future maintenance plan for the benchmark seems satisfactory. Thank you for the authors' tremendous efforts. I have raised my score from 3 to 6.

---

> > > ### Author Response · Authors · 2024-11-27
> > > **Reply to Reviewer 15W6**
> > >
> > > Thank you Reviewer 15W6, your input was exceptionally valuable to enhancing the quality of this work. We are happy to address any further questions.
> > >
> > > Kind Regards,
> > >
> > > Authors

---

### Official Review · Reviewer_xF2u · 2024-11-04

**Soundness:** 2
**Presentation:** 3
**Contribution:** 2
**Rating:** 6
**Confidence:** 3

**Summary:**

This paper proposes a new problem related to data distribution shift in link prediction. To address this, this paper introduces a novel splitting strategy called LPShift to create benchmark data for distribution shift in link prediction. The proposed dataset is used to evaluate current link prediction methods and analyze why these models exhibit weaknesses.

**Strengths:**

The paper proposes a method to introduce distribution shift in link prediction.

By evaluating various methods across multiple benchmarks, it demonstrates that existing models are vulnerable to distribution shifts in link prediction.

Through several analyses, the authors aim to show the effectiveness of the proposed method for inducing distribution shift and to explain why there is a drop in model performance.

**Weaknesses:**

1. The most critical issue is the lack of clarity on why distribution shift is necessary in link prediction and what realistic scenarios justify this need. Can’t we consider changes in edge connections as changes at the graph level? How does graph-level distribution shift differ from this? It would be beneficial to provide specific scenarios that demonstrate the importance of this problem. Without this, it might seem like the issue is being created without a clear purpose.

2. It’s unclear whether this problem is genuinely challenging. In Table 2 for DropEdge and TC (ogbl-ppa), the mean change in performance is marginal, ranging from -1% to +4%. Additionally, in Tables 1 and 7, simple RA and GCN show strong performance.
While these results may highlight limitations of LP methods, could you further explain why this new problem is challenging and why there is a need to advance methods to address it?

3. As a minor suggestion, placing citations within parentheses would enhance readability.

**Questions:**

Please refer to W1 and W2.

1. The most critical issue is the lack of clarity on why distribution shift is necessary in link prediction and what realistic scenarios justify this need. --> Regarding this, please refer to following questions to better justify the importance and novelty of the proposed work:
(1) Provide concrete examples of real-world scenarios where link prediction models face distribution shifts.
(2) Clarify how the proposed distribution shift differs from or relates to graph-level changes.
(3) Explain more explicitly why existing graph-level distribution shift benchmarks are insufficient for link prediction tasks.

2. It’s unclear whether this problem is genuinely challenging. In Table 2 for DropEdge and TC (ogbl-ppa), the mean change in performance is marginal, ranging from -1% to +4%. Additionally, in Tables 1 and 7, both Simple RA and GCN show strong performance.
While these results may highlight limitations of LP methods, could you further explain why this new problem is challenging and why there is a need to advance methods to address it?

---

> ### Author Response · Authors · 2024-11-21
> **Response to Reviewer xF2u (1/3)**
>
> > The most critical issue is the lack of clarity on why distribution shift is necessary in link prediction and what realistic scenarios justify this need.
> >(1) Provide concrete examples of real-world scenarios where link prediction models face distribution shifts.
>
> Our apologies for the lack of clarity on this distinction. We have updated the Introduction section and Figure 1 to provide a more intuitive understanding of this problem and how it relates to covariate shift. We include this revision below in block quotes:
>
> >>Consider the three graphs in Figure 1 as samples pulled from a larger graph dataset. In order of appearance, each of the three graphs represent samples from the: training, validation, and testing splits of our dataset. Our task is to predict whether new links will form between existing nodes. These predicted links are shown as dotted lines with colors: “green” = training, “blue” = validation, “red” = testing. An optimal LP model trained on the first graph sample effectively learns a representation understanding source and target nodes with 2 Common Neighbors Zhang & Chen (2018), making the model effective at predicting new links in structure-rich scenarios. However, such a model is likely to fail when predicting links on the subsequent validation and testing samples; the second and third graph in Figure 1 with 1 CN and 0 CNs, respectively. The learned representation necessary to capture the differing pairwise relations in the second and third graphs requires generalizing to a scenario with fewer CNs, and therefore less structural information. As such, the previously-mentioned
> scenario represents a case of structural covariate shift, where the training distribution cannot effectively model the testing distribution, defined mathematically as P^{Train}(X) \neq P^{Test}(Y). LPShift, our proposed splitting strategy, provides a controllable means to induce the scenario shown in Figure 1 (labelled in experiments as the CN - 2,1 split), as well as fifteen other tested scenarios. More details on structural shift are included in Section 3.2.
>
> Additionally, we acknowledge that realistic examples are important for the practical applicability of our work, and amend these examples into a new Appendix Section. The examples are as follows:
>
> * Adversarial Recommender Systems = Shortest-Path: A company may want to understand which products to *avoid* showing to a potential customer without need to hear the user's preferences directly. In this scenario, global information, as captured by Shortest-Path, becomes the most valuable for the specific use-case.
> * Social Recommender Systems = Preferential-Attachment: A video-streaming platform working with independent content creators may wish to understand what drives users to engage with the platform's content creators, so that engagement can increase for less-popular creators. As a starting point, the content creators with the most followers may have different characteristics that increase engagement versus less-popular creators. So, the streaming platform may wish to tune their dataset to determine if their recommendation system can generalize from more to less-popular creators.
> * Movie Recommender Systems = Common-Neighbors: A movie-streaming platform wants to provide suggestions to users that are the most relevant to the user's current interests. So, the movie platform sorts possible movie recommendations by how much overlap the movies share with one another. In order to enhance exposure to new movies that overlap with a user's interests, the streaming platform can apply the Common Neighbor LPShift to their dataset to force the algorithm to generalize in a scenario where movies may not fully-overlap.
>
> >(2) Clarify how the proposed distribution shift differs from or relates to graph-level changes + Can’t we consider changes in edge connections as changes at the graph level?
>
> Since link prediction considers interactions between source and target nodes, it operates at a finer-scale than what is relevant for graph-classification. Even though enough link-level edits can result in the entire graph structure changing, the addition of a few nodes with sparse edges to a social network is unlikely to induce a graph-level shift. Whether another link will form between these new nodes is likely to remain difficult for link prediction models to predict, even if node or graph classification models succeed at distinguishing the update graphs/nodes. As such LPShift applies pairwise structural heuristics as a means to target link-level shifts within graph datasets.

---

> ### Author Response · Authors · 2024-11-21
> **Response to Reviewer xF2u (2/3)**
>
> > While these results may highlight limitations of LP methods, could you further explain why this new problem is challenging and why there is a need to advance methods to address it?
> >(3) Explain more explicitly why existing graph-level distribution shift benchmarks are insufficient for link prediction tasks.
>
> Existing graph benchmark datasets are concerned entirely with graph- and node-classification, although structural dynamics are important for building effective classifiers with these datasets, (i.e. classifiers should be aware of the differences in molecule structures). These datasets are more concerned how the features define **the way the graph is represented**. For example, OGB and DrugOOD select similar (or dissimilar) molecule scaffolds within a given molecule dataset to group molecules with backbone structures in the same splits. Link prediction, and by extension LPShift, is concerned with modelling finer-scale connections of **links forming**.
>
> Since link prediction is a supervised task, LP models must generalize to scenarios where two nodes with dissimilar features will form a link but two nodes with similar features may not. For example, children in the same grade may take all of their classes together but never become friends due to conflicting interests and vice versa. To effectively model such a scenario, LPShift applies pairwise heuristics to control for the extent that edge structure may influence link prediction. If we were too apply the scaffold split used within OGB or DrugOOD then there is no guarantee that there is a structural shift for the molecule edges at a scale small enough to be applicable for link prediction.
>
> There is a practical difference between graph- or node-classification and link prediction as downstream tasks for graph representation learning. Even though a node-classificaiton model could consider pairwise dynamics; it is impractical for a link prediction model to consider just node labels in order to predict if a link will form between said nodes. Pairwise dynamics are distinct in that they determine how a graph forms on a finer scale, whereas node information constitutes what a graph is made of. The distinctness of these pairwise dynamics then necessitates dedicated evaluation to edge structures within the graphs. As such, LPShift provides a mean to measurable define a shift in these pairwise dynamics to allow evaluation that is relevant for link prediction.
>
> There has been no foundational work that studies how distribution shift affects link prediction performance. As such, this study quantifies what effect structural shift has on LP performance.
>
> There are also no graph benchmarks that are specifically catered towards link prediction, meaning that the problem itself is so under-studied that even knowing where to start is difficult. This study aims to provide a starting point for building new methods.
>
> Even though ogbl-collab is a link-prediction dataset with a known distribution shift, there is no currently practical means to define a metric on the extent that ogbl-collab experiences a distribution shift. So, LPShift provides a benchmark that determines how changes in the direction and intensity of structural shift impacts (and typically decreases) link prediction model performance.
>
> > It’s unclear whether this problem is genuinely challenging. In Table 2 for DropEdge and TC (ogbl-ppa), the mean change in performance is marginal, ranging from -1% to +4%.
>
> We would like to clarify that Table 2 indicates results for ogbl-collab and ogbl-ppa. The application of DropEdge across all LPShift splits of ogbl-collab indicate only a **slight increase** of 4% and 2% on the forward and backward split, respectively. On ogbl-ppa, DropEdge **fails to increase the performance when applied to the backward split** and even **reduces performance by 0.5% when applied to the forward split**.
>
> **This implies that DropEdge doesn't work to improve performance on our datasets**, especially in the backward split scenario. As demonstrated in Figure 1 and 6 of our original submission, the backward split scenario is analogous to realistic distribution shifts.
>
> > Additionally, in Tables 1 and 7, simple RA and GCN show strong performance.
>
> We consider the strong performance of RA and GCN versus GNN4LP models a symptom of the larger problem caused by distribution shift in link prediction. Why bother training advanced GNN4LP models when structural heuristics can achieve similar, if not better, results in a small fraction of the time and computations? Such a result suggests that new methods need to be created in order to tackle distribution shifts for link prediction. This becomes especially pertinent when advanced GNN4LP models, which are designed to understand pairwise dynamics in graphs, fail to generalize under LPShift since they are more sensitive to structural changes in the dataset versus simpler baselines.

---

> ### Author Response · Authors · 2024-11-21
> **Response to Reviewer xF2u (3/3)**
>
> > As a minor suggestion, placing citations within parentheses would enhance readability.
>
> Thank you for the suggestion, readability is a priority of ours. However, our submission uses the template and style provided by the [ICLR Conference](https://iclr.cc/Conferences/2025/CallForPapers). So, it may violate ICLR policy to implement this suggestion.

---

> ### Author Response · Authors · 2024-11-23
> **Gentle Reminder about Rebuttal**
>
> Dear Reviewer xF2u,
>
> Thank you for taking the time to review our work, your input greatly improves the quality of our study!
>
> We have posted our rebuttal and hope that it addresses all of your concerns, especially about LPShift's practicality when compared to realistic distribution shift scenarios. Please consider it in your evaluation and inform us of any additional concerns.
>
> Kind Regards,
>
> Authors

---

> ### Author Response · Authors · 2024-11-25
> **Upcoming Rebuttal Deadline**
>
> Dear Reviewer xF2u,
>
> Thank you for taking the time to review our paper!
>
> As the discussion deadline is rapidly approaching, we are eager to hear your response on our rebuttal as soon as possible. We hope that all aspects of our response addresses your concerns and await additional questions.
>
> Regards,
>
> Authors

---

> > ### Comment · Reviewer_xF2u · 2024-11-26
> >
> > Thank you for the author response.
> > While I still find the explanation regarding the challenging problem related to DropEdge somewhat unclear, I appreciate the effort in addressing my concerns.
> > As some of my concerns have been resolved, I will adjust my score accordingly.
> >
> > Additionally, for your reference, using \citep can help present citations within parentheses while complying with the ICLR submission policy.

---

> > > ### Author Response · Authors · 2024-11-26
> > > **Reply to Reviewer xF2u**
> > >
> > > Thank you for your reponse and review! We applied the \citep to our revised paper and agree that it's much easier to read.
> > >
> > > To improve the clarity on why DropEdge works in a limited capacity under LPShift, we fixed the grammar in the "Observation 1" and "Observation 2" subsections of Section 4.3 in our newly-revised paper. Shown below in block quotes:
> > >
> > > **Observation 1:**
> > >
> > > >> As demonstrated in Table 2, the two generalization methods specific to LP: TC (Wang et al., 2023b) and EPS (Singh et al., 2021) fail to improve performance under LPShift. EPS always results in a decrease of performance from our baseline, indicating a failure to adjust for structural changes induced by LPShift. To validate this, we calculate Earth Mover’s Distance (EMD) (Rubner et al., 1998) between the heuristic scores of the training and testing splits before and after applying the generalization methods. EPS injects CNs into the training adjacency matrix, significantly altering the training and testing distributions. This drastic change is indicated in Figures 4 and 7 with the difference between ’EPS’ and ’LPShift’ EMD scores. Such a change in EMD and the Table 2 results, demonstrate that generalizing under LPShift requires more than simply updating the training graph’s structure.
> > >
> > > **Observation 2:**
> > >
> > > >> As demonstrated in Table 2, DropEdge (Rong et al., 2020) consistently improves performance on LPShift’s ogbl-collab; with a small detrimental effect on ogbl-ppa. Figure 4 indicates that DropEdge causes little change in EMD between training and testing samples. Given that EPS significantly affects EMD scores, DropEdge improving performance is due to minor structural changes in the training adjacency matrix and limited effect on sample distributions. Additional EMD results are provided in Appendix Section I.
> > >
> > > We hope this helps demonstrate why DropEdge works on ogbl-collab. We appreciate your input and are happy to address any further questions or concerns.

---

### Official Review · Reviewer_dNUs · 2024-11-04

**Soundness:** 3
**Presentation:** 3
**Contribution:** 3
**Rating:** 6
**Confidence:** 3

**Summary:**

The paper introduces a dataset-splitting strategy to assess the generalizability of the link prediction models under distribution shifts. LPShift utilizes three structural metrics—Common Neighbors, Shortest Path, and Preferential Attachment—to induce shifts, creating training, validation, and testing sets with distinctive structural properties. The empirical results reveal that GNN4LP models perform poorly under LPShift, with simple heuristics sometimes outperforming complex models. The paper posits LPShift as a framework for studying distribution shifts in LP tasks and highlights the need for generalization methods.

**Strengths:**

1. This paper tackles a compelling and timely research problem: handling complex distribution shifts in graph data for link prediction tasks, a topic of significant recent interest.
2. The splitting strategy is straightforward and well-explained, which is easy to understand.
3. The observations are inspiring to show the author’s motivations.

**Weaknesses:**

1. The novelty is somewhat concerning to me since there is no comprehensive approach proposed.
2. The paper lacks detailed theoretical analyses to support their main claims.
3. I think this paper seems to be a technical report/benchmark paper instead of a research paper. The authors might address this concern by presenting a formal method rather than intuitive explanations to answer the presented questions.

**Questions:**

I am curious what is the main difference among node-level graph OOD method and link-level graph OOD method?

---

> ### Author Response · Authors · 2024-11-21
> **Response to Reviewer dNUs (1/2)**
>
> > The novelty is somewhat concerning to me since there is no comprehensive approach proposed.
>
> Thank you for the suggestion, this was a strong consideration in our initial investigation of this study. Since distribution shift in link prediction is less-studied than distribution shift in graph and node classification, we consider LPShift as a means to understand independent structural mechanisms that affect the formation of links within networks. As such, we build LPShift to serve as a tool which induces distribution shift in current link prediction datasets and benchmark those results.
>
> Since there is limited foundational work on distribution shift in link prediction, we submitted this paper to the datasets and benchmarks track. Therefore, the goal of this paper is to construct a foundational understanding on structural shift in link prediction. This foundational understanding can then foster more research towards building models that can comprehensively handle distribution shift relevant for link prediction.
>
> > The paper lacks detailed theoretical analyses to support their main claims.
>
> Given that this paper is intended as a datasets & benchmarks submission and meant to empirically understand how distribution shift can occur in link prediction datasets, we have prioritized results over theory. In the future, we believe that a paper in similar style to [1] as a means to provide a theoretical perspective.
>
> However, we revise Section 1 in our paper to provide mathematical definitions on how LPShift is a special case of structural covariate shift and relates to more general covariate shift. We include these revisions below as well:
>
> **Section 1:**
>
> >> Consider the three graphs in Figure 1 as samples pulled from a larger graph dataset. In order of appearance, each of the three graphs represent samples from the: training, validation, and testing splits
> of our dataset. Our task is to predict whether new links will form between existing nodes. These predicted links are shown as dotted lines with colors: “green” = training, “blue” = validation, “red” = testing. An optimal LP model trained on the first graph sample effectively learns a representation understanding source and target nodes with 2 Common Neighbors Zhang & Chen (2018), making the model effective at predicting new links in structure-rich scenarios. However, such a model is likely to fail when predicting links on the subsequent validation and testing samples; the second and third graph in Figure 1 with 1 CN and 0 CNs, respectively. The learned representation necessary to capture the differing pairwise relations in the second and third graphs requires generalizing to a scenario with fewer CNs, and therefore less structural information. As such, the previously-mentioned scenario represents a case of structural covariate shift, where the training distribution cannot effectively model the testing distribution, defined mathematically as P^{Train}(X) \neq P^{Test}(Y). LPShift, our proposed splitting strategy, provides a controllable means to induce the scenario shown in Figure 1 (labelled in experiments as the CN - 2,1 split), as well as fifteen other tested scenarios.
>
> [1] Mao, Haitao, et al. "Demystifying structural disparity in graph neural networks: Can one size fit all?." Advances in neural information processing systems 36 (2024).

---

> ### Author Response · Authors · 2024-11-21
> **Response to Reviewer dNUs (2/2)**
>
> > I think this paper seems to be a technical report/benchmark paper instead of a research paper. The authors might address this concern by presenting a formal method rather than intuitive explanations to answer the presented questions.
>
> We clarify that this paper was submitted to the dataset & benchmarks track for ICLR. However, we acknowledge that further insights are important to enhance pracical applicability of this research. As such, we updated analysis within Section 4.2 and the Conclusion to provide more practical insights:
>
> **Section 4.2:**
>
> >> Observation 3: Impacts on Performance Vary by Model. Note: In order to save space, all raw scores are stored in Appendix Section F within Tables 7 to 14. Common Neighbors: Most models fail to generalize on the “Backward” CN splits. However, once more Common Neighbors are made available in the CN - 4,2 and CN - 5,3 splits; NCNC performs 2 to 3 times better than other
> GNN4LP models. Therefore, indicating that it is possible to generalize with limited local information. Shortest-Path: GNN4LP Models which rely more on local structural information (i.e. NCNC, LPFormer, and SEAL) typically suffer worse under the “Backward” SP splits, resulting in the models consistently under-performing 2x to 4x worse versus BUDDY or GCN. Therefore, indicating the necessity for models to extrapolate in scenarios in an absence of local structural information. Preferential-Attachment: There is limited consistency for trends between model performance for the PA splits. However, it is important to note that performance on the PA split often exceeds the original ogbl-collab but reduces drastically with LPShift’s ogbl-ppa. Therefore, indicating the impact that structural shift incurs on larger datasets.
>
> **Conclusion:**
>
> >> Further analysis with EMD calculations and CN distributions indicate that current generalization methods do not improve performance under structural shift. As such, LPShift provides a challenging problem requiring new considerations about how structure functions within link prediction architectures.
>
> > I am curious what is the main difference among node-level graph OOD method and link-level graph OOD method?
>
> Thank you for the question, link prediction (LP) is unique from graph- and node-classification since LP is concerned with determining if a link forms between two nodes for a given sample. Advanced link prediction models take advantage of this princple to consider pairwise interactions between nodes and improve model performance. As such, LPShift leverages LP-relevant structural heuristics to create scenarios where this pairwise information is shifted from the original dataset. Therefore, creating a unique type of structural covariate shift affiliated with pairwise information in graph data. Since the node- and graph-classification tasks are not concerned with structural information encoded within edges, understanding these pairwise dynamics for graph- and node-level shifts is irrelevant. For example, understanding a change in graph size means that the graph represents a new molecule is more important for classifying the graph than understanding that a link will form between two atoms in a given molecule.

---

> ### Author Response · Authors · 2024-11-23
> **Gentle Reminder about Rebuttal**
>
> Dear Reviewer dNUs,
>
> Thank you for taking the time to review our work! Your input greatly improves the quality of our study.
>
> We have posted our rebuttal and hope that it addresses all of your concerns, especially about the theory and analysis contained within our study. Please consider it in your evaluation and inform us of any additional concerns.
>
> Kind Regards,
>
> Authors

---

> ### Author Response · Authors · 2024-11-25
> **Upcoming Rebuttal Deadline**
>
> Dear Reviewer dNUs,
>
> Thank you for taking the time to review our paper!
>
> As the discussion deadline is rapidly approaching, we are eager to hear your response on our rebuttal as soon as possible. We hope that all aspects of our response addresses your concerns and await additional questions.
>
> Regards,
>
> Authors

---

> > ### Comment · Reviewer_dNUs · 2024-11-25
> > **Thanks for rebuttal**
> >
> > Dear authors,
> >
> > Thank you for the detailed response! After reading the response and the other reviews, I am willing to maintain my positive score.

---

> ### Author Response · Authors · 2024-11-25
> **Thank You for your Review**
>
> Dear Reviewer dNUs,
>
> Thank you for your review, we appreciate the input since it improves our work! If you have any further concerns or questions please feel free to leave a comment.
>
> Best Regards,
>
> Authors

---

### Official Review · Reviewer_QSyV · 2024-11-05

**Soundness:** 2
**Presentation:** 2
**Contribution:** 2
**Rating:** 5
**Confidence:** 4

**Summary:**

The paper investigates the generalizability of link prediction models under distribution shifts, a crucial problem given the structural differences often encountered between training and real-world data. To address this, the authors introduce LPShift, a dataset-splitting strategy that induces distribution shifts based on structural properties, allowing for a controlled study of model performance under these conditions. They evaluate several state-of-the-art Graph Neural Network for Link Prediction models and generalization methods across multiple split scenarios.

**Strengths:**

1. The focus on distribution shifts in link prediction is valuable, as models often fail to generalize well when such shifts occur in real-world settings, like social networks or recommender systems. This paper directly addresses this issue by benchmarking models in varied shift scenarios.

2. The paper provides comprehensive baseline comparisons under the distribution shift setting.

**Weaknesses:**

1. The motivation behind distribution shifts in link prediction is not effectively illustrated in the introduction. A clearer example is needed to demonstrate why link prediction models might struggle with distribution shifts:
Figure 1 is confusing. Models should easily predict links with more common neighbors (blue and green links) compared to red links, aligning with typical link prediction assumptions. However, this is not explained well, which weakens the introductory motivation.

2. The definition of distribution shift, based solely on heuristics, feels arbitrary. It would strengthen the study if the importance of these heuristics in link prediction were demonstrated upfront, clarifying their role in creating meaningful shifts.
a) Previous studies (e.g., [1]) emphasize the significance of global structure information, which seems not fully considered in the LPShift splitting strategy.
b)  The use of temporal datasets like Collab may not suit this split strategy, as it is sensitive to time. As a benchmark paper, two datasets seem limited.

3. The paper’s terminology needs refinement for clarity. For example, using consistent names for terms like “inverse”/“backward” splits would reduce confusion. “’repair’” should be replaced with “repair” for consistency and correctness.

4. As a benchmark study, the insights provided are minimal. It would be more valuable to include a deeper analysis, such as how different splitting strategies impact performance or specific guidance on choosing GNN4LP models based on observed patterns in the results.

[1] Revisiting Link Prediction: A Data Perspective. ICLR 2024.

**Questions:**

Why is it important to focus on distribution shifts in link prediction? While distribution shifts can impact GNN performance, is it useful to have a benchmark dedicated to evaluating these shifts independently? In many cases, performance on the original dataset splits remains a key concern. For instance, using the time-based splits in OGB-Collab might offer a more practical perspective. I would consider raising the score if the authors provide a promising rebuttal for my questions and concerns.

---

> ### Author Response · Authors · 2024-11-21
> **Response to Reviewer QSyV (1/7)**
>
> > The motivation behind distribution shifts in link prediction is not effectively illustrated in the introduction. A clearer example is needed to demonstrate why link prediction models might struggle with distribution shifts: Figure 1 is confusing. Models should easily predict links with more common neighbors (blue and green links) compared to red links, aligning with typical link prediction assumptions. However, this is not explained well, which weakens the introductory motivation.
>
> Thank you for bringing this to our attention. We have updated Figure 1 to demonstrate the CN - 2,1 LPShift scenario instead of the originally depicted CN - 1,2 LPShift scenario. The figure now follows prevailing assumptions in link prediction, detailing a scenario where a link prediction model trains on 2 Common Neighbors and then must generalize on testing samples with 0 Common Neighbors. Our empirical tests on the "Backward" CN splits align with this assumption, as indicated by the loss in performance shown between Table 7 (the 'Forward' ogbl-collab split) and Table 11 (the "Backward" ogbl-collab split) in the Appendix. We include a representative example of results both tables in Table 1, as shown below:
>
> | Collab | CN - 1,2 - "Forward"     | CN - 2,4       | CN - 3,5         | CN - 2,1 - "Backward"            | CN - 4,2      | CN - 5,3    |
> |:--------:|:-------------:|:------------:|:-------------:|:--------------------:|:------------:|:------------:|
> | RA | 32.22 | 29.74 | 29.86 | 0.6 | 4.79 | **15.9** |
> GCN  | 12.92 ± 0.31 | 15.20 ± 0.16 | 17.54 ± 0.19 | 5.78 ± 0.19 | 5.91 ± 0.05 | 7.07 ± 0.11 |
> BUDDY| 17.48 ± 1.19 | 15.47 ± 0.57 | 16.60 ± 0.89 | 3.70 ± 0.13 | 3.55 ± 0.09 | 5.40 ± 0.04 |
> NCNC | 9.00 ± 1.02 | 13.99 ± 1.35 | 15.04 ± 1.25 | 1.89 ± 1.27 | **16.48 ± 1.30** | **19.69 ± 1.52** |
> LPFormer | 4.27 ± 1.17 | 13.7 ± 1.48 | 25.36 ± 2.04 | 2.01 ± 0.95 | 3.87 ± 0.74 | 8.36 ± 0.63 |
> NeoGNN | 5.76 ± 1.69 | 16.10 ± 0.82 | 18.22 ± 0.61 | 2.14 ± 0.05 | 3.48 ± 0.10 | 9.44 ± 0.38 |
> SEAL | 7.06 ± 1.67 | 20.60 ± 6.23 | 19.78 ± 1.24 | 1.01 ± 0.02 | 3.38 ± 0.62 | 8.08 ± 1.96 |
>
> **Table 1 (shown above):** An example of the CN Results combined from Tables 7 and 11 in Appendix Section F of the revised paper. Note: the significant decrease in performance for most models between the "Backward" splits in the final three columns versus the "Forward" splits in the first three columns. Notably, RA and NCNC performance improves signficantly once more Common Neighbors are made available in the validation and testing splits, highlighted in bold (i.e. CN - 4,2 and CN - 5,3).

---

> ### Author Response · Authors · 2024-11-21
> **Response to Reviewer QSyV (2/7)**
>
> > The definition of distribution shift, based solely on heuristics, feels arbitrary. It would strengthen the study if the importance of these heuristics in link prediction were demonstrated upfront, clarifying their role in creating meaningful shifts.
>
> Thank you for the suggestion. We've revised our paper in the following ways:
>
> * **We relate LPShift to the mathematical definition of covariate shift**. This can be found in the updated introduction section, highlighted in blue indicating that LPShift induces a special case of structural covariate shift. We also provide a more detailed perspective on how LPShift can create meaningful dataset shifts. Detailed below:
>
> >> Consider the three graphs in Figure 1 as samples pulled from a larger graph dataset. In order of appearance, each of the three graphs represent samples from the: training, validation, and testing splits of our dataset. Our task is to predict whether new links will form between existing nodes. These predicted links are shown as dotted lines with colors: “green” = training, “blue” = validation, “red” = testing. An optimal LP model trained on the first graph sample effectively learns a representation understanding source and target nodes with 2 Common Neighbors Zhang & Chen (2018), making the model effective at predicting new links in structure-rich scenarios. However, such a model is likely to fail when predicting links on the subsequent validation and testing samples; the second and third graph in Figure 1 with 1 CN and 0 CNs, respectively. The learned representation necessary to capture the differing pairwise relations in the second and third graphs requires generalizing to a scenario with fewer CNs, and therefore less structural information. As such, the previously-mentioned scenario represents a case of structural covariate shift, where the training distribution cannot effectively model the testing distribution, defined mathematically as $P^{Train}(X) \neq P^{Test}(Y)$. LPShift, our proposed splitting strategy, provides a controllable means to induce the scenario shown in Figure 1 (labelled in experiments as the CN - 2,1 split), as well as fifteen other tested scenarios. More details on structural shift are included in Section 3.2.
>
> * **We include a discussion of the importance that the structural heuristics play in link prediction and the types of structural shifts they can induce**. This is in Section 3.1, highlighted in blue.
>
> >> Furthermore even on complex real-world datasets, CNs achieves competitive performance against more advanced neural models Hu et al. (2020). To control for the effect of CNs on shifted performance, the relevant splits will consider thresholds which include more CNs.
>
> > a) Previous studies (e.g., [1]) emphasize the significance of global structure information, which seems not fully considered in the LPShift splitting strategy.
>
> We would like to clarify that we consider global structure information in Section 3.1 of our original submission, including a citation by Mao et. al [1]. Following this consideration, we apply the Shortest-Path heuristic with LPShift for it's ability to capture global structure information; with further details provided in part (2) of Section 3.1, detailed below in quotes:
>
> >> (2) Shortest Path (SP): SP captures a graph’s global structural information, thanks to the shortest-path between a given target and source node representing the most efficient path for reaching the target Russell & Norvig (2009). The shift in global structure caused by splitting data with SP can induce a scenario where a model must learn how two dissimilar nodes form a link with one another Evtushenko & Kleinberg (2021), which is comparable to the real-world scenario where two opponents choose to co-operate with one another Schelling (1978); Granovetter (1978).
>
> [1] Haitao Mao, Juanhui Li, Harry Shomer, Bingheng Li, Wenqi Fan, Yao Ma, Tong Zhao, Neil Shah, and Jiliang Tang. Revisiting link prediction: A data perspective. ICLR, 2024.
>
> > b) The use of temporal datasets like Collab may not suit this split strategy, as it is sensitive to time.
>
> We agree that the distinction amongst splits types is important for learning effective graph representations. However, LPShift functions as a means to induce a tunable shift within the structure of a given link prediction dataset. Even though the original ogbl-collab dataset is split temporally based on the node years, the distribution of Common Neighbors within the graph samples inherently capture the temporal nature of this split. We visualize this natural shift in edges within Subplot 1 of Figure 5.

---

> ### Author Response · Authors · 2024-11-21
> **Response to Reviewer QSyV (3/7)**
>
> We can then consider how the graph structure within ogbl-collab changes over time and how that relates to LPShift. Ogbl-collab's nodes from more recent years are likely to have fewer edges than older nodes, a shifted scenario due to the fact that older nodes in ogbl-collab have progressively more time to accumulate more edges/citations. Given that LPShift functions on the adjacency matrix as a whole, regardless of dataset split, the CN - 2,1 split can be seen as means to test a similar scenario caused by the original ogbl-collab. As such, the original temporal split remains captured within datasets altered by LPShift.
>
> > As a benchmark paper, two datasets seem limited.
>
> We originally made this decision since the ogbl datasets are relatively large compared to other datasets, such as cora and citeseer. Therefore, the evaluation of LPShift more closely follows what is expected in real-world settings with larger and difficult datasets. However, we acknowledge that additional datasets will provide more solid empirical evidence for our claims. In Table 2, shown below, we include additional empirical results for cora, citeseer, and pubmed.
>
> We note the following observation from the results shown in Table 1:
> 1. The general trend that we discuss in our original paper remains: a change in the structure of the training adjacency matrix results in a change in performance. Thus, indicating the importance of structural information for LP model performance.
> 2. RA performs significantly worse than all tested LP models.
> 3. Regardless of the 'forward' and 'backward' splits, there is an increase in performance for all GNN4LP and LP models.
> 4. Also following along the trend of our original submission, GCN continues to perform competitively against GNN4LP models. However, there are scenarios, such as Pubmed - CN 3,5, where GNN4LP models can outrank GCN.
>
> * The limited effectiveness of RA along with the reduced competitiveness of GCN is attributed to the relative lack of structural information contained within the LPShift versions of Cora, Citeseer, and Pubmed. Since Cora, Citeseer, and Pubmed are a fraction of the size of ogbl-collab and ogbl-ppa, there is simply less relevant links captured by CN, SP, and PA. As such, RA has fewer training links to reliably rank testing links, causing it's performance to plummet.
> * In regards to GCN, since the raw number of relevant CN, SP, PA links in each split is markedly lower than the LPShift versions of ogbl-collab and ogbl-ppa; this means that the structural information captured in the training adjacency matrix is more important. Therefore, resulting in more competitive performance from the GNN4LP models versus GCN as well as the increase in perforamnce regardless of 'forward' or 'backward' split.

---

> ### Author Response · Authors · 2024-11-21
> **Response to Reviewer QSyV (4/7)**
>
> **Table 2 (shown below)**: Cora, citeseer, pubmed results for all models tested with LPShift.
>
> |          | Cora - CN     |              |               | Cora - SP            |              | Cora - PA     |              |              |
> |:--------:|:-------------:|:------------:|:-------------:|:--------------------:|:------------:|:-------------:|:------------:|:------------:|
> | Method   |               |              |               |                      |              |               |              |              |
> |          | 1,2           | 2,4          | 3,5           | 17,26                | 26,36        | 50,100        | 100,200      | 150,250      |
> | GCN      | 9.90 ± 0.64   | 29.91 ± 1.72 | 32.73 ± 1.17  | 3.78 ± 0.63          | 5.36 ± 0.87  | 12.12 ± 0.99  | 12.30 ± 2.03 | 12.34 ± 2.29 |
> | NCNC     | 12.82 ± 0.44  | 19.57 ± 1.14 | 31.14 ± 4.01  | 10.41 ± 1.20         | 9.97 ± 0.33  | 11.86 ± 0.90  | 10.06 ± 3.07 | 10.67 ± 2.20 |
> | BUDDY    | 11.24 ± 0.40  | 26.34 ± 1.20 | 28.59 ± 0.72  | 7.46 ± 1.37          | 10.01 ± 0.35 | 14.38 ± 0.72  | 11.23 ± 0.78 | 8.20 ± 0.45  |
> | RA       | 0.76          | 23.49        | 29.98         | 0.79                 | 0.78         | 6.6           | 7.78         | 9.61         |
> | NeoGNN   | 15.06 ± 3.97  | 31.67 ± 2.17 | 26.67 ± 0.98  | 9.44 ± 0.84          | 9.90 ± 0.50  | 12.09 ± 0.50  | 10.93 ± 1.07 | 10.17 ± 1.34 |
> | SEAL     | 7.24 ± 1.36   | 15.51 ± 8.65 | 13.90 ± 10.33 | 1.83 ± 0.83          | 1.93 ± 0.40  | 7.33 ± 0.68   | 5.40 ± 0.70  | 8.95 ± 2.38  |
> | LPFormer | 12.84 ± 0.90  | 23.42 ± 2.49 | 22.80 ± 3.94  | 1.96 ± 0.18          | 2.08 ± 0.17  | 9.15 ± 0.67   | 12.91 ± 2.49 | 8.28 ± 1.16  |
> |          | 2,1           | 4,2          | 5,3           | 26,17                | 36,26        | 100,50        | 200,100      | 250,150      |
> | GCN      | 7.22 ± 0.87   | 20.96 ± 3.65 | 23.03 ± 3.45  | 4.08 ± 0.42          | 5.26 ± 0.55  | 8.28 ± 0.62   | 6.81 ± 0.52  | 7.47 ± 0.71  |
> | NCNC     | 11.70 ± 1.83  | 30.31 ± 3.76 | 30.65 ± 4.00  | 6.04 ± 0.70          | 6.46 ± 0.62  | 8.40 ± 0.65   | 7.01 ± 1.12  | 7.03 ± 0.77  |
> | BUDDY    | 9.11 ± 1.36   | 18.09 ± 2.26 | 34.97 ± 5.36  | 2.93 ± 0.62          | 4.86 ± 0.39  | 4.16 ± 0.18   | 3.93 ± 0.47  | 4.70 ± 0.16  |
> | RA       | 0.76          | 15.55        | 28.43         | 0.76                 | 0.76         | 5.55          | 3.19         | 3.23         |
> | NeoGNN   | 9.24 ± 1.04   | 28.60 ± 3.06 | 37.74 ± 2.71  | 4.12 ± 0.31          | 5.15 ± 0.65  | 8.43 ± 0.25   | 6.46 ± 0.36  | 8.02 ± 0.49  |
> | SEAL     | 3.82 ± 0.94   | 10.30 ± 5.94 | 20.32 ± 14.79 | 1.95 ± 0.79          | 1.43 ± 0.20  | 3.77 ± 1.24   | 1.74 ± 0.60  | 2.46 ± 0.18  |
> | LPFormer | 4.36 ± 1.53   | 13.67 ± 2.67 | 31.19 ± 1.94  | 1.44 ± 0.22          | 2.25 ± 0.16  | 6.28 ± 2.49   | 4.48 ± 1.34  | 4.84 ± 1.04  |
> |          |               |              |               |                      |              |               |              |              |
> |          | Citeseer - CN |              |               | Citeseer - SP        |              | Citeseer - PA |              |              |
> |          | 1,2           | 2,4          | 3,5           | 17,26                | 26,36        | 50,100        | 100,200      | 150,250      |
> | GCN      | 13.64 ± 1.60  | 21.82 ± 2.04 | 20.06 ± 1.46  | 2.92 ± 0.81          | 7.56 ± 0.58  | 13.92 ± 2.07  | 13.09 ± 4.23 | 15.85 ± 7.01 |
> | NCNC     | 19.56 ± 4.74  | 20.46 ± 2.12 | 21.64 ± 3.73  | 16.93 ± 8.33         | 19.03 ± 8.17 | 14.72 ± 3.63  | 10.49 ± 2.40 | 14.73 ± 3.55 |
> | BUDDY    | 19.79 ± 0.57  | 25.77 ± 0.60 | 23.57 ± 0.59  | 21.31 ± 1.01         | 13.31 ± 1.36 | 15.62 ± 0.63  | 18.19 ± 0.80 | 7.89 ± 1.12  |
> | RA       | 0.76          | 11.83        | 14.32         | 0.79                 | 0.79         | 3.78          | 3.69         | 4.74         |
> | NeoGNN   | 29.65 ± 2.78  | 21.24 ± 3.26 | 19.56 ± 1.99  | 12.83 ± 2.71         | 17.87 ± 3.64 | 12.84 ± 1.21  | 10.81 ± 1.83 | 7.92 ± 0.98  |
> | SEAL     | 12.65 ± 1.98  | 14.22 ± 2.36 | 15.92 ± 3.04  | 1.56 ± 0.76          | 1.41 ± 0.40  | 4.13 ± 0.55   | 6.71 ± 0.86  | 5.76 ± 0.75  |
> | LPFormer | 18.52 ± 1.89  | 20.08 ± 0.65 | 16.02 ± 1.31  | 1.90 ± 0.30          | 1.90 ± 0.05  | 11.26 ± 1.53  | 6.49 ± 1.08  | 10.79 ± 3.72 |

---

> ### Author Response · Authors · 2024-11-21
> **Response to Reviewer QSyV (5/7)**
>
> |          | Citeseer - CN |              |               | Citeseer - SP        |              | Citeseer - PA |              |              |
> |:--------:|:-------------:|:------------:|:-------------:|:--------------------:|:------------:|:-------------:|:------------:|:------------:|
> |          | 2,1           | 4,2          | 5,3           | 26,17                | 36,26        | 100,50        | 200,100      | 250,150      |
> | GCN      | 9.56 ± 2.06   | 9.23 ± 1.60  | 7.29 ± 1.32   | 2.27 ± 0.45          | 4.68 ± 0.85  | 9.58 ± 0.67   | 3.11 ± 0.24  | 4.55 ± 0.66  |
> | NCNC     | 12.62 ± 4.74  | 10.43 ± 2.13 | 9.31 ± 1.15   | 5.88 ± 2.80          | 10.02 ± 1.83 | 11.99 ± 2.10  | 3.62 ± 0.30  | 3.38 ± 0.28  |
> | BUDDY    | 13.08 ± 0.22  | 10.55 ± 0.35 | 7.48 ± 1.74   | 8.10 ± 0.60          | 11.73 ± 1.47 | 9.19 ± 0.72   | 4.00 ± 0.27  | 5.09 ± 0.12  |
> | RA       | 0.76          | 9.67         | 8.15          | 0.76                 | 0.77         | 6.99          | 3.95         | 3.37         |
> | NeoGNN   | 8.30 ± 0.91   | 13.05 ± 1.06 | 8.41 ± 0.18   | 5.52 ± 3.67          | 7.19 ± 2.60  | 10.33 ± 1.33  | 4.15 ± 0.36  | 4.70 ± 1.10  |
> | SEAL     | 10.09 ± 2.94  | 7.85 ± 1.25  | 10.20 ± 2.52  | 1.17 ± 0.40          | 2.27 ± 0.72  | 5.95 ± 1.27   | 4.87 ± 1.24  | 2.66 ± 0.46  |
> | LPFormer | 5.52 ± 1.18   | 12.90 ± 4.20 | 13.50 ± 3.78  | 1.18 ± 0.28          | 3.01 ± 0.52  | 11.78 ± 2.96  | 6.52 ± 2.81  | 4.93 ± 1.59  |
> |          |               |              |               |                      |              |               |              |              |
> |          | Pubmed - CN   |              |               | Pubmed - SP          |              | Pubmed - PA   |              |              |
> |          | 1,2           | 2,4          | 3,5           | 17,26                | 26,36        | 50,100        | 100,200      | 150,250      |
> | GCN      | 16.81 ± 0.44  | 13.85 ± 0.16 | 7.81 ± 0.42   | 19.24 ± 0.48         | 20.26 ± 0.61 | 26.68 ± 2.42  | 21.59 ± 0.85 | 19.06 ± 0.55 |
> | NCNC     | 19.19 ± 1.40  | 13.51 ± 0.58 | 7.51 ± 0.31   | 24.78 ± 0.50         | 22.60 ± 1.09 | 31.02 ± 0.89  | 24.05 ± 0.89 | 20.58 ± 0.82 |
> | BUDDY    | 3.48 ± 0.74   | 6.72 ± 0.18  | 5.90 ± 0.17   | 14.53 ± 1.15         | 20.57 ± 0.58 | 26.25 ± 0.54  | 11.45 ± 1.61 | 15.75 ± 0.75 |
> | RA       | 0.71          | 6.69         | 6.27          | 0.79                 | 0.78         | 3.4           | 3.89         | 4.02         |
> | NeoGNN   | 37.43 ± 9.70  | 14.81 ± 0.41 | 9.38 ± 0.31   | 27.10 ± 0.37         | 23.76 ± 1.02 | 26.00 ± 1.37  | 20.59 ± 0.52 | 17.99 ± 0.49 |
> | SEAL     | 16.09 ± 8.41  | 13.67 ± 1.15 |   10.76 ± 0.64   | 3.25 ± 1.16          | 2.70 ± 0.28  | 5.32 ± 1.09   | 9.69 ± 0.94  | 10.68 ± 0.84 |
> | LPFormer | 17.45 ± 3.24  | 11.75 ± 1.12 | 11.53 ± 4.06  | 2.05 ± 0.13          | 2.91 ± 0.20  | 5.76 ± 0.50   | 8.11 ± 1.53  | 9.54 ± 1.48  |
> |          | 2,1           | 4,2          | 5,3           | 26,17                | 36,26        | 100,50        | 200,100      | 250,150      |
> | GCN      | 6.07 ± 0.17   | 8.46 ± 0.18  | 10.52 ± 0.21  | 4.70 ± 0.19          | 5.50 ± 0.25  | 5.50 ± 0.19   | 4.46 ± 0.07  | 4.34 ± 0.18  |
> | NCNC     | 5.54 ± 0.78   | 9.16 ± 0.45  | 10.38 ± 0.69  | 4.85 ± 0.80          | 5.54 ± 0.72  | 3.86 ± 0.36   | 3.91 ± 0.32  | 6.02 ± 0.89  |
> | BUDDY    | 3.52 ± 0.17   | 6.38 ± 0.27  | 8.40 ± 0.12   | 4.99 ± 0.28          | 2.86 ± 0.28  | 3.65 ± 0.14   | 3.38 ± 0.10  | 3.38 ± 0.12  |
> | RA       | 0.69          | 3.41         | 5.46          | 0.73                 | 0.71         | 1.92          | 1.71         | 1.75         |
> | NeoGNN   | 5.19 ± 0.25   | 8.05 ± 0.24  | 9.05 ± 0.30   | 5.09 ± 0.26          | 4.60 ± 0.15  | 5.05 ± 0.21   | 4.21 ± 0.15  | 4.12 ± 0.14  |
> | SEAL     | 4.53 ± 0.60   | 7.94 ± 1.20  | 10.72 ± 1.26  | 0.98 ± 0.37          | 1.49 ± 0.32  | 7.03 ± 1.13   | 3.73 ± 0.86  | 3.73 ± 0.81 |
> | LPFormer | 2.95 ± 0.49   | 7.59 ± 0.88  | 10.39 ± 1.51  | 0.82 ± 0.09          | 1.90 ± 0.28  | 5.50 ± 0.28   | 4.32 ± 0.37  | 3.24 ± 0.25  |

---

> ### Author Response · Authors · 2024-11-21
> **Response to Reviewer QSyV (6/7)**
>
> > The paper’s terminology needs refinement for clarity. For example, using consistent names for terms like “inverse”/“backward” splits would reduce confusion. “’repair’” should be replaced with “repair” for consistency and correctness.
>
> We are sorry for any lack of clarity within the writing. We've revised the paper submission to make it consistent. We highlight edits in blue, including Table captions in the Appendix. We include example sentences of this correction, as detailed below:
>
> *Lines 433-434:* However, Figure 4 indicates that DropEdge only has a significant effect on EMD when handling the “backward” ogbl-collab CN split...
> *Lines 361-364:*  On the “backward” split, a stark increase is seen across most splits, which indicates that increasingly-available structural information in the training adjacency matrix yields improved LP performance (Wang et al., 2023a).
>
> > As a benchmark study, the insights provided are minimal. It would be more valuable to include a deeper analysis, such as how different splitting strategies impact performance or specific guidance on choosing GNN4LP models based on observed patterns in the results.
>
> Our revised Discussion section 4.2 includes a new 'Observation 3', highlighted in blue, that provide more detailed analysis on how different splitting strategies affect LP model performance and relevant insights about structural dynamics that can aid choosing effective GNN4LP models. Shown below in block quotes:
>
> >> Common Neighbors: Most models fail to generalize on the “Backward” CN splits. However, once more Common Neighbors are made available in the CN - 4,2 and CN - 5,3 splits; NCNC performs 2 to 3 times better than other GNN4LP models. Therefore, indicating that it is possible to generalize with limited local information. Shortest-Path: GNN4LP Models which rely more on local structural information (i.e. NCNC, LPFormer, and SEAL) typically suffer worse under the “Backward” SP splits, resulting in the models consistently under-performing 2x to 4x worse versus BUDDY or GCN. Therefore, indicating the necessity for models to extrapolate in scenarios in an absence of local structural information. Preferential-Attachment: There is limited consistency for trends between model performance for the PA splits. However, it is important to note that performance on the PA split often exceeds the original ogbl-collab but reduces drastically with LPShift’s ogbl-ppa. Therefore, indicating the impact that structural shift incurs on larger datasets.
>
> > Why is it important to focus on distribution shifts in link prediction?
> > While distribution shifts can impact GNN performance, is it useful to have a benchmark dedicated to evaluating these shifts independently?
>
> The importance of benchmarking distribution shifts in link prediction is threefold:
>
> **1) Data Perspective:** To clarify, link prediction is a task focused on understanding the dynamics present when edges form between nodes; requiring models to effectively understand these dynamics to determine whether a link will form (or not). As such, link prediction is more interested in pairwise dynamics than what is necessary for graph and node classification. Due to this distinction, there is limited overlap between graph/node classification and link prediction. As such, distribution shifts that are relevant in graph and node classification do not mean as much in link prediction. For example, we consider the scaffold shift imposed on molecule datasets within the DrugOOD benchmark. This special type of structural shift groups molecules with similar subgraphs in a way to induce an out-of-distribution scenario. Given that DrugOOD scaffold shift is not associated with the formation of links, then it loses it's relevancy when applied to models that learn pairwise dynamics necessary for link prediction.
>
> **2) Model Perspective:** Structural heuristics that consider pairwise information are important for SOTA link prediction models. For example, BUDDY integrates RA directly into it's architecture and NCNC elevates CN into a neural architecture. SOTA node classification models can be concerned with how graph structure may enhance node features. For example, GLEM is a node classification model that merges text-attributed graphs architectures with graph structure to produce SOTA performance on [ogbn-products](https://ogb.stanford.edu/docs/leader_nodeprop/). However, instead of considering pairwise information, GLEM leverages the information of neighboring nodes to produce better node classifications; completing ignoring how altering edges may affect the model.

---

> ### Author Response · Authors · 2024-11-21
> **Response to Reviewer QSyV (7/7)**
>
> **3) Application Perspective:** There is a practical difference between graph- or node-classification and link prediction as downstream tasks for graph representation learning. Even though a node-classificaiton model could consider pairwise dynamics; it is impractical for a link prediction model to consider just node labels in order to predict if a link will form between said nodes. Pairwise dynamics are distinct in that they determine **how a graph forms on a finer scale**, whereas node information constitutes **what a graph is made of**. The distinctness of these pairwise dynamics then necessitates dedicated evaluation to edge structures within the graphs. As such, LPShift provides a mean to measurable define a shift in these pairwise dynamics to allow evaluation that is relevant for link prediction.
>
> [1] Zhao, Jianan, et al. "Learning on Large-scale Text-attributed Graphs via Variational Inference." The Eleventh International Conference on Learning Representations.
>
> > In many cases, performance on the original dataset splits remains a key concern. For instance, using the time-based splits in OGB-Collab might offer a more practical perspective.
>
> Benchmarks on link prediction models have begun to reach near-perfect performance. However, structural shift causes a noticeable decrease in performance, even for datasets in the same domain. For example, GraphGPT(d1n30) reaches an MRR score of 93.05±0.2 on the [ogbl-citation2 dataset](https://ogb.stanford.edu/docs/leader_linkprop/#ogbl-citation2) whereas [ogbl-collab](https://ogb.stanford.edu/docs/leader_linkprop/#ogbl-collab) lags behind with a Hits@50 score of 71.29±0.18. As demonstrated in our original submission's Figure 5, there is a noticeable shift in the Common Neighbors distribution between the training and validation/testing splits of the original ogbl-collab. Since both ogbl-collab and ogbl-citation2 are split temporally, then it becomes less promising to induce time-based shifts. This is due to the fact that the graph structure in citation networks is not likely to change over time, meaning that a quantifiable temporal shift which decreases performance is difficult to induce. LPShift handles this problem by using structural heuristics to create an easily-defined threshold; allowing researchers to determine the exact value that contributes to a decrease in performance.

---

> ### Author Response · Authors · 2024-11-23
> **Gentle Reminder about Rebuttal**
>
> Dear Reviewer QSyV,
>
> Thank you for taking the time and effort to review our work! We appreciate your input, it greatly improves the quality of our study.
>
> We have posted our rebuttal and hope that it addresses all of your concerns, especially about the importance of studying distribution shifts in link prediction. Please consider it in your evaluation and inform us of any additional concerns.
>
> Kind Regards,
>
> Authors

---

> ### Author Response · Authors · 2024-11-25
> **Upcoming Rebuttal Deadline**
>
> Dear Reviewer QSyV,
>
> Thank you for taking the time to review our paper!
>
> As the discussion deadline is rapidly approaching, we are eager to hear your response on our rebuttal as soon as possible. We hope that all aspects of our response addresses your concerns and await additional questions.
>
> Regards,
>
> Authors

---

> ### Author Response · Authors · 2024-11-27
> **Gentle Reminder about Rebuttal**
>
> Dear Reviewer QSyV,
>
> Thank you for taking the effort to review our paper! We appreciate your input as it improves our work.
>
> Given that today is the last day to revise the paper, we hope we have addressed your concerns. Please let us know if you have additional questions or concerns.
>
> Best Regards,
>
> Authors

---

> > ### Author Response · Authors · 2024-11-28
> > **Added Paper Revision**
> >
> > Dear Reviewer QSyV,
> >
> > Thank you for your review, we have appended Appendix Section L with the appropriate citations in order to provide a threefold perspective as to why distribution shift in link prediction, and therefore LPShift is important. We hope to have addressed all of your concerns and are happy to answer any further questions. We include the edits below in block quotes:
> >
> > >> Data Perspective: Link prediction is a task focused on understanding the dynamics when
> > edges form between nodes; requiring models to effectively understand these dynamics to
> > determine whether a link will form (or not) (Liben-Nowell & Kleinberg, 2003). As such,
> > link prediction is more interested in pairwise dynamics than what is necessary for graph
> > and node classification. Due to this distinction, there is limited overlap between graph/node
> > classification and link prediction. As such, distribution shifts that are relevant in graph and
> > node classification do not mean as much in link prediction. For example, we consider
> > the scaffold shift imposed on molecule datasets within the DrugOOD benchmark (Ji et al.,
> > 2022). This special type of structural shift groups molecules with similar subgraphs to
> > induce an out-of-distribution scenario. Given that the scaffold shift in DrugOOD is not
> > associated with the formation of links, then the shift loses it’s relevancy when applied to
> > models that learn pairwise dynamics necessary for link prediction.
> >
> > >> • Model Perspective: Structural heuristics that consider pairwise information are important
> > for SOTA GNN4LP models. For example, BUDDY (Chamberlain et al., 2022) integrates
> > RA directly into it’s architecture and NCNC (Wang et al., 2023a) elevates CN into a neural
> > architecture. The integration of these structural heuristics allow the link prediction models
> > to achieve SOTA performance. However, as demonstrated in Tables 23 and 24, GNN4LP’s
> > reliance on structural heuristics leads to degraded performance when LPShift induces struc-
> > tural shift within the graph dataset. The performance of models less reliant on structure,
> > such as GCN (Kipf & Welling, 2017), do not degrade as significantly. This indicates that
> > future models must balance between learning on graph structure and understanding shifts
> > in link formation to improve performance under LPShift; especially in scenarios analogous
> > to real-life (i.e. Figure 1).
> >
> > >> • Application Perspective: There is a practical difference between graph or node classifica-
> > tion and link prediction as downstream tasks for graph representation learning (Hu et al.,
> > 2020). Even though a graph or node classification model could consider pairwise dynam-
> > ics; it is limiting for a link prediction model to only consider node labels when predicting if
> > edges will form (Yun et al., 2021; Zhang et al., 2021). Pairwise dynamics are distinct in that
> > they determine how a graph forms on a finer scale, whereas node information constitutes
> > what a graph is made of (Adamic & Adar, 2003; Barab´asi & Albert, 1999). The distinct-
> > ness of these pairwise dynamics then necessitates dedicated evaluation to edge structures
> > within the graphs. As such, LPShift serves as a measurable means to define a shift in
> > these pairwise dynamics, allowing evaluation in a controlled setting that is relevant for link
> > prediction

---

> > > ### Comment · Reviewer_QSyV · 2024-12-01
> > >
> > > Thank you for providing the detailed rebuttal. While some of my concerns have been addressed, I still have reservations about the dataset selection and the depth of insights provided. I will adjust my score to 5 and will continue to engage in discussions with the other reviewers.

---

> ### Author Response · Authors · 2024-12-01
> **Response to Reviewer QSyV**
>
> Thank you for your review, given the extended discussion period we are happy to address any further concerns.
>
> Our paper considers:
> * 5 datasets, with 16 LPShift splits and 7 tested models. (3 new datasets -- 560 model experiments)
> * 5 traditional OOD generalization methods (new)
> * 2 LP-Specific and 1 graph-data generalization method with included analysis (updated) on the effect that these methods have on dataset structure.
> * In particular, the ogbl-ppa and ogbl-collab datasets are much larger than most available benchmark datasets and consist of two distinct domains; allowing insights from this paper to better translate to real-world settings.
>
> Given that this paper is for the datasets & benchmarks track, our insight is focused on LPShift's effect for current model and generalization method performance. This way, we intend to provide a foundation from which future research can develop methods to handle distribution shift in link prediction. We hope further concerns about the insights are addressed clearly within our revised paper. Please let us know if there any specific pain points about the datasets or analysis insights.

---

### Author Response · Authors · 2024-12-03
**Rebuttal Summary (2/2)**

* *Heuristic Type*
    * Since there is no currently quantifiable means to induce distribution shift relevant for link prediction, LPShift leverages well-known link prediction heuristics in order to ensure that pairwise information is properly considered within the distribution shift. As such, LPShift is modular since all it needs is a heuristic to generate dataset splits based on threshold magnitudes and direction. Our study focuses on Common Neighbors, Shortest-Path, and Preferential-Attachment since these heuristics allow targeting for local, global, and node-degree information, respectively.
* *Time Efficiency*
    * Given LPShift's use of heuristics, applying LPShift as a splitting strategy is far more time-efficient than gathering a new dataset manually. As such, researchers wishing to target local structural information within a real-world dataset can apply various magnitudes and directions for a given heuristic in an LPShift split to generate multiple different perspectives on the original dataset.

Our study on LPShift provides a foundation to quantify the previously-mentioned factors. From which, we observe interesting patterns in the behavior of GNN4LP and basic LP methods:
* Resource Allocation (RA) overwhelmingly performs the best on LPShift's "Forward" ogbl-collab splits. However, RA performs the worst on the "Backward" SP and CN - 2,1 splits, whereas GCN performs the best. This result is especially interesting, since a prevailing assumption in link prediction is that **GNN4LP generalize better in scenarios where local structural information is more available between training and validation. LPShift induces this exact scenario with it's "Forward" splits.**
* NCNC performs worse than GCN on the CN - 2,1 split for both ogbl-collab and ogbl-ppa. However, NCNC rises to become the best-performing model on the subsequent CN - 4,2 and CN - 5,3 splits. Therefore indicating **NCNC's sensitivity to shifts within local information**, but a capability to generalize well once more local structural information is made available.
* LPFormer and SEAL perform competitively on many of the PA splits for ogbl-collab and ogbl-ppa, with a special case for NCNC on the "Backwards" ogbl-ppa PA split. In this scenario, the three models frequently out-performs other GNN4LP models by margins of 10 MRR. Therefore, indicating their ability to generalize given an abundance of highly-connected nodes within the training distribution. This is especially interesting given that Neo-GNN frequently performs the worst on these splits, indicating that **the neighborhood information Neo-GNN requires is lost within the "Backwards" PA splits**.
* BUDDY demonstrates an interesting resilience to the "Backwards" SP splits for both ogbl-ppa and ogbl-collab. From which, BUDDY always out-performs other GNN4LP methods. In this same scenario, SEAL's lower performance indicates that **the ability to distinguish subgraph features is more important than neighboorhood information when considering a shift in long-range structural information**.

Given the previously-mentioned observations, **LPShift allows analysis to adopt a fine-grained perspective which eliminates concerns about whether a GNN4LP model can leverage an unfair advantage by exploiting structural quirks within given datasets**. Such a level of analysis would be tedious with real-world disitribution shifts, requiring identification and then expensive computations in order to find targeted graph substructures.  Given that **distribution shift is not just dataset-dependent but can also be task-specific**, there is no one-size-fits-all solution. However, as long as a structural heuristic (or any combination of heuristics) is relevant to a given researcher's interest, then **LPShift serves as a means to test any link prediction models under targeted structural shifts and gain insights about model weakpoints**.


[1] Liben-Nowell, David, and Jon Kleinberg. "The link prediction problem for social networks." Proceedings of the twelfth international conference on Information and knowledge management. 2003.

[2] Zhao, Jianan, et al. "Learning on large-scale text-attributed graphs via variational inference." arXiv preprint arXiv:2210.14709 (2022).

[3] Gui, Shurui, et al. "Good: A graph out-of-distribution benchmark." Advances in Neural Information Processing Systems 35 (2022): 2059-2073.

[4] Ji, Yuanfeng, et al. "DrugOOD: Out-of-Distribution (OOD) Dataset Curator and Benchmark for AI-aided Drug Discovery--A Focus on Affinity Prediction Problems with Noise Annotations." arXiv preprint arXiv:2201.09637 (2022).

[5] Zou, Deyu, et al. "GeSS: Benchmarking Geometric Deep Learning under Scientific Applications with Distribution Shifts." The Thirty-eight Conference on Neural Information Processing Systems Datasets and Benchmarks Track.

[6] Mao, Haitao, et al. "Revisiting link prediction: A data perspective." arXiv preprint arXiv:2310.00793 (2023).

---

### Author Response · Authors · 2024-12-03
**Rebuttal Summary (1/2)**

Our deepest thanks to all reviewers, your varied perspectives and feedback improved the quality of our paper significantly. Since we are approaching the end of the rebuttal period, we summarize how we've adressed all concerns below:

* Numerous grammatical edits for clarity on motivations and insights provided by the paper. In particular, we detail why distribution shift matters for link prediction.
* Additional observations, examples, and updated figures/tables on how LPShift affects link prediction performance and relates to realistic scenarios.
* Updated repository for replication of baseline results and generating LPShift splits with included code re-factoring plan.
* 5 new tests on traditional generalization methods: VRex, IRM, GroupDRO, DANN, Deep CORAL. From which, we were able to determine that LPShift is resistant to performance improvements from traditional generalization methods.
* 3 additional datasets: Cora, Citeseer, and PubMed for quantifying the effect of LPShift on smaller benchmark datasets.

Additionally, we wish to detail the core motivation behind LPShift, shown below:

The core motivation for building LPShift is it's practical utility. Not only is there a lack of OOD benchmark datasets in link prediction, but also **a lack of study understanding distribution shift and it's mechanisms within link prediction**. We attribute this lack of study to the difference in task-dependent mechanisms between graph/node classification and link prediction. Pairwise and neighborhood structural heuristics are critical for link prediction performance [1]. As a self-supervised task, link prediction requires understanding structure within samples to determine if target links will form. Whereas node/graph classification do not require understanding pairwise structural dynamics to significantly boost performance [2].

Given link prediction's reliance on pairwise information, **all tested GNN4LP architectures integrate structural information for powerful performance**. The methods for integrating structural characteristics into GNN4LP models are distinct; NCNC integrates Common Neighbors (CN), BUDDY integrates subgraph features and RA, Neo-GNN applies neighborhood characteristics, SEAL elevates structural expressiveness with the labelling trick, LPFormer applies pairwise attention. But their ability to generalize under distribution shift was not deeply studied until LPShift.

Real-world distribution shifts, like molecular scaffold shift, can be seen as a mixture of heuristic structural shifts. So, applying current GNN4LP models on **complicated real-world shifts make it unnecessarily difficult to control how GNN4LP and their distinct structural methods generalize under distribution shift**. Therefore, we apply LPShift not just for building datasets with distribution shift, but to also **control** the shift introduced into the dataset [3,4, 5]. **LPShift's fine-grained control makes it simple to analyze how specific structures and their attributes affect GNN4LP performance**. From here, LPShift integrates factors such as the magnitude, direction, heuristic type, and time efficiency in order to define a structural shift that was previously un-quantifiable. We break the dynamics behind these factors further:

* *Magnitude*
    * We determine the amount of structural information contained within each sample distribution with hard-boundary threshold values. For example, CN - 1,2 has a valid threshold at 1 and a testing threshold at 2; meaning that training has zero Common Neighbors, validation has a single Common Neighbors, and testing contains 2 or more Common Neighbors. The magnitude of structural information within these splits can then be easily adjusted to accomodate more or less information (i.e. CN - 2,4 with zero and 1 Common Neighbors in training, 2 and 3 Common Neighbors in validation, and 4 or more Common Neighbors in testing).
* *Direction*
    * Given a certain magnitude between training, validation, and testing; we then build a "Forward" and "Backward" split of LPShift for each dataset. The "Forward" split represents a scenario where the magnitude of structural information available subsequently increases between splits (i.e. CN - 1,2). The "Backward" split represents the scenario, were the magnitude of structural information available within splits subsequently decreases between splits. From example, CN - 2,1 will have training samples with 2 or greater Common Neighbors but then validates on samples with 1 Common Neighbors and eventually test on samples with no Common Neighbors.

---

### Meta-Review · Area_Chair_CEfn · 2024-12-19

**Metareview:**

This work tackles the important task of evaluating link prediction models' generalizability under distribution shifts. The work proposes LPShift, a dataset-splitting strategy designed to induce distribution shifts based on structural properties, enabling a controlled examination of model performance under these conditions. Through comprehensive experiments on multiple datasets and state-of-the-art Graph Neural Network (GNN) models, the paper shows that existing link prediction methods are vulnerable to distribution shifts, often resulting in poor performance compared to their in-distribution versions.

However, despite the importance of the problem and the potential impact of the proposed benchmark, the reviewers have raised several concerns regarding the paper's writing quality and clarity. More significantly, the paper lacks a rigorous theoretical and empirical justification for the proposed LPShift benchmark. As reviewer QSyV astutely points out in the rebuttal, the very few foundational works on link prediction under distribution shifts does not excuse not having a solid theoretical grounding and real-world justification for the benchmark itself.

Upon closer examination of the paper, it becomes apparent to me that many of the reviewer concerns stem from the authors' failure to provide a clear and well-motivated causal model for the distribution shift. Specifically, the paper could benefit from: (1) a more explicit causal model that underlies the proposed shift, (2) a thorough justification of this causal model through real-world mechanisms or theoretical arguments, (3) a proof or empirical demonstration that the proposed shift indeed produces the correct structure under the assumed causal model, and (4) deeper insights into the problem, including an analysis of the implications of the distribution shift on the link prediction task.

By addressing these concerns and providing a more comprehensive and theoretically sound foundation for the LPShift benchmark, the authors could significantly strengthen their contribution and provide a more convincing case for the importance and relevance of their work. In the current form, I feel the paper is not ready for publication, unfortunately.

**Additional Comments On Reviewer Discussion:**

There was a good discussion. The authors did a decent job in the rebuttal but the paper needs more work.

---

### Decision · Program_Chairs · 2025-01-22

Reject